# Continuous-time edge modelling using non-parametric point processes

**Xuhui Fan**[1], **Bin Li**[2], **Feng Zhou**[3], and **Scott A. Sisson**[1]

[1]UNSW Data Science Hub, and School of Mathematics & Statistics, University of New South Wales
[2]Shanghai Key Lab of IIP, School of Computer Science, Fudan University
[3]Department of Computer Science & Technology, Tsinghua University,
*{xuhui.fan, scott.sisson}@unsw.edu.au; libin@fudan.edu.cn; zhoufeng6288@tsinghua.edu.cn*

## Abstract

The mutually-exciting Hawkes process (ME-HP) is a natural choice to model *reciprocity*, which is an important attribute of continuous-time edge (dyadic) data. However, existing ways of implementing the ME-HP for such data are either inflexible, as the exogenous (background) rate functions are typically constant and the endogenous (excitation) rate functions are specified parametrically, or inefficient, as inference usually relies on Markov chain Monte Carlo methods with high computational costs. To address these limitations, we discuss various approaches to model design, and develop three variants of non-parametric point processes for continuous-time edge modelling (CTEM). The resulting models are highly adaptable as they generate intensity functions through sigmoidal Gaussian processes, and so provide greater modelling flexibility than parametric forms. The models are implemented via a fast variational inference method enabled by a novel edge modelling construction. The superior performance of the proposed CTEM models is demonstrated through extensive experimental evaluations on four real-world continuous-time edge data sets.

## 1 Introduction

Continuous-time edge (or dyadic) data, which comprise information on edges (between two individuals or *nodes*) and their times of occurrence, arise naturally in many applications, such as sending and receiving messages, posting and answering comments on social networks, and purchasing groceries in online stores. Reciprocity is an important attribute of edge data, in that many edges are induced as responses to previous oppositely-directed edges (e.g. a reply email is written in response to an initial message between two individuals). As a result, mutually-exciting Hawkes processes (ME-HP) [13, 12], which regard the directed edges as events, are commonly used to analyse edge data [3, 8, 25, 29, 26, 20]. For example, [3] uses the ME-HP to model the reciprocity of group-wise edges, and [20] uses sparse exchangeable random measures [4, 28] to model the ME-HPs exogenous values.

However, these existing methods are either limited in modelling flexibility, or inefficient to implement. For the former, the exogenous (background rate) functions in the ME-HP are typically assumed to be static and the endogenous (excitation) functions are specified parametrically, which may be inadequate for modelling complex real-world systems. E.g., the exogenous intensities may exhibit daily or weekly periodicity, and the endogenous functions might not be strictly decreasing over time. For the latter, Markov chain Monte Carlo (MCMC) methods are usually adopted for model inference, which usually require high computational costs and large output storage. Without proper scalable inference methods, it is difficult to deploy these models to larger-scale datasets.

35th Conference on Neural Information Processing Systems (NeurIPS 2021).

In this work, to address the above limitations, we develop continuous-time edge models (CTEM) based on non-parametric point processes. The CTEM models also use point processes to model data specific to nodal pairs. CTEMs have two major differences compared to existing approaches: Firstly, they place sigmoidal Gaussian process priors on the individual exogenous functions and endogenous functions. As a result, the exogenous functions are able to evolve over time and account for gradual changes and variations in the background environment (such as, e.g., trends or periodicity). Similarly, non-parametric modelling of the endogenous functions enables more behaviour choices (e.g., the rate of decrease in the excitation functions may vary over time, or there might be increases in mutual-excitation effects). Secondly, we develop an efficient variational inference method for model inference, which is more suited for large-scale data modelling than MCMC methods.

By specifying $K$ exogenous functions and $K$ endogenous functions, we propose a family of three non-parametric CTEM models on $N$ nodes, in which the directed edges from node $i$ to node $j$ are regarded as events for the $(i, j)$-th point process:

- **Nodal pair-specific Poisson process (NP-PP):** For each directed nodal pair $(i, j)$, $i, j \in \{1, \ldots, N\}$, the intensity function for their Poisson process is constructed from the linear combination of $K$ exogenous functions and nodes $i, j$'s latent exogenous features.
- **Nodal pair-specific mutually-exciting Hawkes process (NP-HP):** We additionally include the combination of $K$ endogenous functions and nodes $i, j$'s latent endogenous features as the endogenous functions for the $(i, j)$-th ME-HP.
- **Mutually-exciting Hawkes process with edge clustering (NP-HP-C):** Rather than using nodal pair-specific endogenous functions, we associate each edge with a latent label, and assume that those edges with the same label are allocated to the same endogenous function.

The last approach may be used to model the phenomenon that many edges are often associated with certain behavioural or other patterns. For instance, messages may be grouped into particular labelled events, or edges associated with different patterns may generate different triggering effects.

Inspired by inference techniques [5, 31] which adopt the Polya-Gamma trick to form conjugate models for sigmoidal Gaussian process-modulated Poisson processes and Hawkes processes, we develop mean-field variational algorithms to enable fast inference for the proposed models. In addition we also construct a fast variational inference solution to the Dirichlet-Hawkes process related models [7, 19, 24], which typically heavily rely on Sequential Monte Carlo [6] methods.

In summary, the merits of the proposed CTEM family include: (1) **Modelling flexibility:** The proposed models are highly flexible as they are able to describe both exogenous and endogenous functions using non-parametric sigmoidal Gaussian processes. The experimental results on four real-world data sets show clear periodic patterns in both exogenous and endogenous functions. (2) **Inferential efficiency:** To the best of our knowledge, this is the first work to develop fast variational inference for the continuous-time edge modelling problem, making it suitable for large-scale relational modelling settings; (3) **Modelling choice exploration:** The CTEM family permits different structural choices in modelling continuous-time edge data, including nodal pair-specific intensity functions and latent label determined endogenous functions.

## 2 Background on mutually-exciting Hawkes processes for edge data

The mutually-exciting Hawkes process [12, 13] is a special case of the temporal point process (TPP). TPPs are realised through sets of events in a time interval $[0, T]$, and are parameterised by an intensity function $\lambda(t)$, which states that the probability of an event occurring in any infinitesimal time interval $[t, t + dt)$ is $\lambda(t)dt$. TPPs which are independent of the history of the process (e.g. a Poisson point process) are not well-suited to describe edge data (which comprise directed events from one node to another), as the occurrence of edges may be the result of responses to previous oppositely-directed edges. For example, receiving a message from a person would likely increase the probability of sending a response message to that person over the following time period.

The ME-HP provides a natural way to model the reciprocity of edge data. Its intensity functions $\lambda_{ij}(t)$, which describes edge generation from node $i$ to node $j$ at time $t$, depend on the historical edges from node $j$ to node $i$. We use the triplet $(i_m, j_m, t_m)$ to denote an edge, meaning that among all edges, the $m$-th one is initiated from node $i_m$ to node $j_m$ and occurred at time $t_m$. Using $f_{ij}(t)$

and $g_{ij}(t)$ as the exogenous function and endogenous function from node $i$ to node $j$ respectively, the intensity function $\lambda_{ij}(t)$ and the likelihood $\mathcal{L}(\cdot)$ of the ME-HP over all $M$ edges is

$$\lambda_{ij}(t) = f_{ij}(t) + \sum_{m':i_{m'}=j \cap j_{m'}=i} g_{ij}(t - t_{m'}), \tag{1}$$

$$\mathcal{L}(\{\lambda_{ij}(t)\}_{(i,j):i\neq j}|\{(i_m, j_m, t_m)\}_{m=1}^M|) = \prod_{(i,j):i\neq j} \left[ e^{-\int_0^T \lambda_{ij}(t)dt} \prod_{m':i_{m'}=i \cap j_{m'}=j} \lambda_{ij}(t_{m'}). \right] \tag{2}$$

**ME-HP with block models:** Stochastic block models [22, 21, 14] are a classical way to describe static edge data, via partitioned nodal groups and group-wise edges. Several methods [3, 8, 25, 29, 26, 20] have incorporated stochastic block models within the ME-HP to describe edge data reciprocity. Among these, the HPGP-IRM [25] is the most similar model to ours, with the generative process

$$\nu \sim \text{CRP}(1:N), \quad \lambda_{ij}(t) = \gamma_{\nu(i)\nu(j)} + \sum_{m':i_{m'}=j \cap j_{m'}=i} \beta_{ij}(F_i, F_j) g_{ij}(t - t_{m'}) \tag{3}$$

where $\nu$ is a Chinese Restaurant Process (CRP) [23] group partition on all $N$ nodes, $\gamma_{\nu(i)\nu(j)}$ is the exogenous intensity from person $i$'s group $\nu(i)$ to person $j$'s group $\nu(j)$, $\beta_{ij}(F_i, F_j)$ is the endogenous scalar measuring the feature similarity between person $i$ and person $j$ through their features $F_i, F_j$, and the occurrence times $\{t_m\}_m$ follow a Hawkes process (HP) with intensity function $\lambda_{ij}(t)$. Similar to other approaches, the HPGP-IRM specifies parametric functions for both exogenous and endogenous functions, and its MCMC-based approach to inference is not practical in large-scale settings. Further, the absence of edge clustering capability means that intrinsic structure among persons, especially the sharing of endogenous functions, can not be explored.

## 3 Non-parametric point processes for continuous-time edge modelling

The family of Continuous-Time Edge Modelling (CTEM) methods is inspired by three motivations: constructing highly flexible models to adequately describe the complexities of real world edge data; enabling fast variational inference; and exploring different modelling choices for studying continuous-time edge data. The following sections introduce three non-parametric point processes, in which the intensity functions are generated from sigmoidal Gaussian processes.

### 3.1 Nodal pair-specific non-parametric Poisson processes (NP-PP)

The first approach (NP-PP) is to use nodal pair-specific Poisson processes to model the continuous-time edge data. This involves $K$ exogenous functions $\sigma(f_k(t))$, where $f_k(t)$ is a random function generated from a Gaussian process and $\sigma(x) = 1/(1 + \exp(-x))$ is the sigmoidal function to ensure the positiveness of the intensity values. Each node $i$ is assumed to have $K$-length positive latent exogenous features $\boldsymbol{\pi}_i \in [\mathbb{R}^+]^{1 \times K}$, in which the $k$-th entry $\pi_{ik}$ represents node $i$'s affiliation with the $k$-th exogenous feature. The intensity function of the Poisson process for generating edges from node $i$ to node $j$ is thus a linear combination of $\{\sigma(f_k(t))\}_k$ weighted by $\boldsymbol{\pi}_i, \boldsymbol{\pi}_j$, with the corresponding generative process described as follows:

1. Generate $f_k(t) \sim \text{GP}(0, \kappa_k^{(f)})$ for group features $k = 1, \ldots, K$, where $\text{GP}(0, \kappa)$ denotes a Gaussian process with mean 0 and covariance function $\kappa(\cdot, \cdot)$.

2. Generate $\{\pi_{ik}\}_{i=1}^N \sim \text{Gam}(a_\pi, b_\pi)$ for group features $k = 1, \ldots, K$, where $\text{Gam}(a, b)$ denotes a Gamma distribution with mean $a/b$ and variance $a/b^2$.

3. Generate the time stamps of edges from node $i$ to node $j$ as: $\{(i, j, t_{m'})\}_{m'} \sim \text{PoissonProcess}(\lambda_{ij}(t))$, with intensity function $\lambda_{ij}(t) = \sum_k \pi_{ik} \pi_{jk} \cdot \sigma(f_k(t))$.

Under this setting, the intensity functions between any nodal pair are symmetric $\lambda_{ij}(t) = \lambda_{ji}(t)$. Together, $\pi_{ik}$ and $\pi_{jk}$ scale the $k$-th exogenous intensity onto the interval $(0, \pi_{ik}\pi_{jk})$. Nodes with similar feature-affiliation values are more likely to generate edges. The $L_1$-norm of the latent features of a node $\boldsymbol{\pi}_i$ indicates its edge activity, since a larger value means it is more likely to induce edges.

This non-parametric setting provides greater modelling flexibility than those using static or piecewise-constant exogenous functions. I.e., the NP-PP allows each exogenous function to change over time, while most existing approaches [3, 8, 25, 29, 26, 20] use a time-constant exogenous intensity value, even though real-world settings are known to vary in this manner. [8] also used Poisson processes to model the edges, however, their intensity functions are piecewise-constant with the changepoints and intensty values almost pre-fixed, which is substantially less flexible than our approach.

However, without incorporating endogenous functions, the NP-PP cannot describe the *reciprocity* property of continuous-time edges. A direct consequence might be overestimated exogenous intensities, as they need to explain all edges, and consequently lower model performance.

## 3.2 Nodal pair-specific non-parametric Hawkes process (NP-HP)

Our second approach is to use mutually-exciting Hawkes processes (ME-HP) to model the continuous-time edge data. In addition to the non-parametric exogenous functions in the first approach, the NP-HP includes nodal pair-specific endogenous functions to model the reciprocity of edge data. In particular, the endogenous function for the $(i,j)$-th nodal pair is structured as $\sum_k \rho_{ik}\rho_{jk} \cdot \sigma(g_k(t))$, where $\{\sigma(g_k(t))\}_k$ are random functions from a sigmoidal Gaussian process and $\boldsymbol{\rho}_i \in [\mathbb{R}^+]^{1 \times K}$ is node $i$'s $K$-length latent endogenous features. As a result, the occurrence of an edge at time $t_m$ is either instantiated by the exogenous effect (i.e. $\sum_k \pi_{ik}\pi_{jk} \cdot \sigma(f_k(t_m))$) or triggered by one of its historical oppositely-directed edges (through the endogenous effect $\sum_k \rho_{ik}\rho_{jk} \cdot \sigma(g_k(t_m - t_{m'}))$, where $m' \in H_{ij}(t_m)$ and $H_{ij}(t) = \{m' : (t_{m'} < t) \cap (i_{m'}, j_{m'}) = (j, i)\}$ denotes the set of historical (i.e. occurring before time stamp $t$) oppositely-directed (i.e. from node $j$ to node $i$) edges). Since the occurrence of an edge increases the intensity function value of its future oppositely-directed edges, the NP-HP can describe reciprocity well. It's generative process is described as follows:

1. Generate $f_k(t) \sim \text{GP}(0, \kappa_k^{(f)})$, $\quad g_k(t) \sim \text{GP}(0, \kappa_k^{(g)})$, $\quad k = 1, \ldots, K$.
2. Generate $\pi_{ik} \sim \text{Gam}(a_\pi, b_\pi)$, $\quad \rho_{ik} \sim \text{Gam}(a_\rho, b_\rho)$, $\quad i = 1, \ldots, N, k = 1, \ldots, K$.
3. Simultaneously generate two sets of triplets $\{(i, j, t_{m'})\}_{m'}$ and $\{(j, i, t_{m'})\}_{m'}$, which are edges initiated from node $i$ to node $j$, and from node $j$ to node $i$ respectively, from Hawkes processes with the following intensity functions:

$$\lambda_{ij}(t) = \sum_k \pi_{ik}\pi_{jk} \cdot \sigma(f_k(t)) + \sum_k \sum_{m':m' \in H_{ij}(t)} \rho_{ik}\rho_{jk} \cdot \sigma(g_k(t - t_{m'})), \quad (4)$$

$$\lambda_{ji}(t) = \sum_k \pi_{jk}\pi_{ik} \cdot \sigma(f_k(t)) + \sum_k \sum_{m':m' \in H_{ji}(t)} \rho_{jk}\rho_{ik} \cdot \sigma(g_k(t - t_{m'})). \quad (5)$$

Since the functions $\{\sigma(g_k(t))\}_k$ are also drawn from sigmoidal Gaussian process, the NP-HP's endogenous functions possess greater flexibility than previous approaches, in which the endogenous functions are typically specified as exponential functions. Incorporating the product of the nodes' latent endogenous features ensures that these endogenous functions are nodal pair specific, as for the exogenous functions. However, NP-HP assumes that each nodal pair would use same endogenous functions for all its edges. This might be unable to describe the changes in the edges, such as emails with different topics between two persons might generate distinct endogenous effects.

## 3.3 Non-parametric Hawkes process with edge-clustering effects (NP-HP-C)

The third approach introduces an edge clustering mechanism for the mutually-exciting Hawkes process with nodal pair-specific exogenous functions. Here each generated edge is assigned a latent group label. The NP-HP-C generative process for edges in the time period $[0, T]$ is then as follows:

1. Generate $f_k(t) \sim \text{GP}(0, \kappa_k^{(f)})$, $\quad g_k(t) \sim \text{GP}(0, \kappa_k^{(g)})$, $\quad k = 1, \ldots, K$.
2. Generate $\pi_{ik} \sim \text{Gam}(a_\pi, b_\pi)$, $\quad k = 1, \ldots, K, i = 1, \ldots, N$;
3. Generate $v_k \sim \text{Gam}(a_v, b_v), k = 1, \ldots, K$, where $v_k$ is the scaling variable for the $k$-th endogenous function $\sigma(g_k(t))$.
4. For each pair of nodes $\{(i, j) : 1 \leq i < j \leq N\}$, generate triplets $\{(i, j, t_{ij}^{(m')}), b_{ij}^{(m')}, z_{ij}^{(m')}\}_{m'}$ and $\{(j, i, t_{ji}^{(m'')}), b_{ji}^{(m'')}, z_{ji}^{(m'')}\}_{m''}$ for the two directed edges

according to the following procedure, until $t_{M'} \leq T < t_{M'+1}, t_{M''} \leq T < t_{M''+1}$.
Initialise the current time stamp at $t_* = 0$ and set $M_{ij} = M_{ji} = 0$. Then:

(a) Starting from $t_*$, obtain two candidate event times $\bar{t}_{ij} = \min(\{\bar{t}_{ij}^{(l)}\}_l | \bar{t}_{ij}^{(l)} > t_*), \bar{t}_{ji} = \min(\{\bar{t}_{ji}^{(l)}\}_l | \bar{t}_{ji}^{(l)} > t_*)$, where the two time sequences $\{\bar{t}_{ij}^{(l)}\}_l, \{\bar{t}_{ji}^{(l)}\}_l$ are generated from:

$$\{\bar{t}_{ij}^{(l)}\}_l \sim \text{PoissonProcess}(\sum_{k=1}^{K} \pi_{ik}\pi_{jk}\sigma(f_k(t)) + \sum_{m'=1}^{M_{ji}} v_{z_{ji}^{(m')}} \cdot \sigma(g_{z_{ji}^{(m')}}(t - t_{ji}^{(m')}))), \quad (6)$$

$$\{\bar{t}_{ji}^{(l)}\}_l \sim \text{PoissonProcess}(\sum_{k=1}^{K} \pi_{ik}\pi_{jk}\sigma(f_k(t)) + \sum_{m'=1}^{M_{ij}} v_{z_{ij}^{(m')}} \cdot \sigma(g_{z_{ij}^{(m')}}(t - t_{ij}^{(m')}))). \quad (7)$$

(Note that the second term in Eqs (6) and (7) is not involved when $M_{ij} = M_{ji} = 0$). If $\bar{t}_{ij} > T \cap \bar{t}_{ji} > T$, break for this nodal pair and start generating edges for other pairs.

(b) If $\bar{t}_{ij} < \bar{t}_{ji}$, set $M_{ij} = M_{ij} + 1, t_* = t_{ij}^{(M_{ij})} = \bar{t}_{ij}$ and generate $b_{ij}^{(M_{ij})}, z_{ij}^{(M_{ij})}$ from $(m' = 1, \ldots, M_{ji})$

$$P(b_{ij}^{(M_{ij})}, z_{ij}^{(M_{ij})}) \propto \begin{cases} \pi_{ik}\pi_{jk}\sigma(f_k(t_*)), & \text{if } b_{ij}^{(M_{ij})} = 0, z_{ij}^{(M_{ij})} = k; \\ v_{z_{ji}^{(m')}}\sigma(g_{z_{ji}^{(m')}}(t_* - t_{ji}^{(m')})), & \text{if } b_{ij}^{(M_{ij})} = m', z_{ij}^{(M_{ij})} = z_{ji}^{(m')}, \end{cases}$$

else, set $M_{ji} = M_{ji} + 1, t_* = t_{ji}^{(M_{ji})} = \bar{t}_{ji}$ and generate $b_{ji}^{(M_{ji})}, z_{ji}^{(M_{ji})}$ from $(m' = 1, \ldots, M_{ij})$

$$P(b_{ji}^{(M_{ji})}, z_{ji}^{(M_{ji})}) \propto \begin{cases} \pi_{ik}\pi_{jk}\sigma(f_k(t_*)), & \text{if } b_{ji}^{(M_{ji})} = 0, z_{ji}^{(M_{ji})} = k; \\ v_{z_{ij}^{(m')}}\sigma(g_{z_{ij}^{(m')}}(t_* - t_{ij}^{(m')})), & \text{if } b_{ji}^{(M_{ji})} = m', z_{ji}^{(M_{ji})} = z_{ij}^{(m')}. \end{cases}$$

where $M_{ij}$ is the number of edges from node $i$ to node $j$, and where $t_{ij}^{(m)}, b_{ij}^{(m)} \in \{\{0\} \cup H_{ij}(t_{ij}^{(m)})\}$, and $z_{ij}^{(m)} \in \{1, \ldots, K\}$ denote the time, "source" factor (i.e., exogenous rate or endogenous rates) and latent label of the $m$-th edge from node $i$ to node $j$ respectively. The above generative process produces observations between nodal pairs in the form $\{\{(i, j, t_{ij}^{(m')}), b_{ij}^{(m')}, z_{ij}^{(m')}\}_{m'=1}^{M_{ij}}\}_{i \neq j}$. For notational convenience, we reformat these as $\{(i_m, j_m, t_m), b_m, z_m\}_{m=1}^{M}$, with $m$ denotes the edge number among all the edges, for the following discussion.

In this generative process, steps 1 & 2 generate the exogenous function random bases and the nodes' latent features, in the same way as for the NP-HP. The exogenous function in the NP-HP-C $\sum_k \pi_{ik}\pi_{jk} \cdot \sigma(f_k(t))$ is also nodal-pair specific. Step 3 generates the scaling variables for $K$ endogenous functions. Step 4.(a) generates two directed edges between each nodal pair. Starting from any time stamp $t_*$, the Poisson processes with the current intensity functions (Eqs (6) and (7)) are used to generate the candidate edge occurrence time stamps $\bar{t}_{ij}, \bar{t}_{ji}$ for each edge direction. If both $\bar{t}_{ij}, \bar{t}_{ji}$ are larger than $T$, which means that the two Poisson processes do not generate edges in $(t_*, T]$, we stop generating edges for this nodal pair. Otherwise, we use the earlier candidate time stamp (e.g. we take $\bar{t}_{ij}$ if $\bar{t}_{ij} < \bar{t}_{ji}$) as the actually occurring edge time, and add its triggering effect to the oppositely-directed intensity function (from the actual occurrence time).

Step 4.(b) uses the idea of branching structure [17, 32] to describe the "source" factor of the edge triplet $(i_m, j_m, t_m)$. That is, $b_m = 0$ if the triplet $(i_m, j_m, t_m)$ is instantiated by an exogenous intensity and $b_m = m'(m' \in H_{i_m j_m}(t_m))$ if the triplet $(i_m, j_m, t_m)$ is triggered by its historical oppositely-directed edge triplet $(j_m, i_m, t_{m'})$. At the same time, $z_m \in \{1, \ldots, K\}$ is the latent label of $(i_m, j_m, t_m)$ which determines the pattern group to which $(i_m, j_m, t_m)$'s triggering effect belongs. The edges with the same label use the same endogenous function. In this way, the endogenous functions are shared among the edges of all nodal pairs. Those edges triggered by the same labelled edges (through the same labelled endogenous function) or instantiated by the same labelled exogenous component are associated with the same endogenous function. This process is visualised in Figure 1.

As previously mentioned, NP-HP-C's clustering mechanism for the endogenous functions of edges can help modelling practical phenomena, such as urgent emails would usually result in fast response, regardless of the nodal pair. What is more, NP-HP-C uses a much smaller number ($K$) of endogenous functions than NP-HP. This small number might be able to help NP-HP-C avoid overfitting issues.

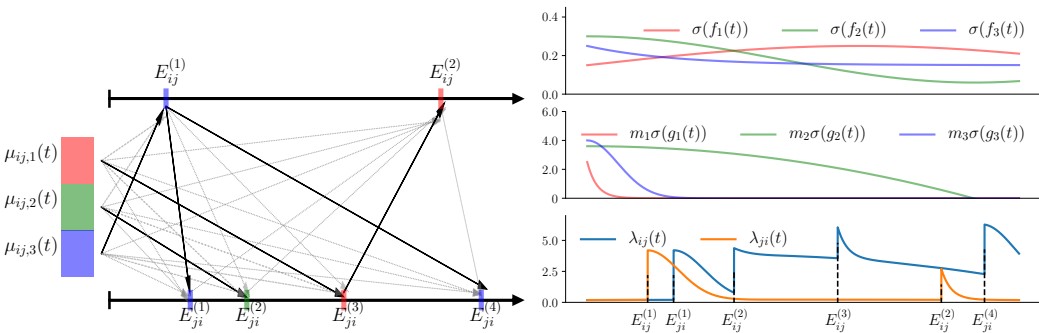

Figure 1: (Left panel) Visualisation of edge generation between nodes $i$ and $j$ in the NP-HP-C. Different edge colours (red, green, blue) represent different edge-group labels. Arrows (grey-dotted and black solid arrows) link the exogenous functions and past oppositely-directed edges to current edges, and they together represent all potential triggering relationships between mutually-excited edges. Black solid arrows denote the actual triggering relationships. If an edge is triggered by one of its past oppositely-directed edges, their colours (i.e labels) are the same. (Right panel) From top to bottom: the exogenous functions $\{\sigma(f_k(t))\}_{k=1}^3$, the endogenous functions $\{v_k \cdot \sigma(g_k(t))\}_{k=1}^3$, and the corresponding intensity functions $\lambda_{ij}(t), \lambda_{ji}(t)$ between nodes $i$ and $j$.

### 3.4 Mean-field variational inference

To implement faster posterior inference than a MCMC implementation, we use the strategy of [5, 31] to develop mean-field variational posterior approximation [2]. In particular, [5] integrated a Poly-Gamma trick into a variational approximation for the sigmoidal Gaussian process-modulated Poisson process, and [31] applied similar techniques in the simple Hawkes point process setting. For e.g. the NP-HP-C model, we aim to approximate the joint posterior distribution of sigmoidal Gaussian process modulated random functions $\{f_k(t), g_k(t)\}_k$, the endogenous function scaling values $\{v_k\}_k$, the latent features $\{\pi_{ik}\}_{i,k}$, and the branching variables $\{b_m\}_m$ and latent labels $\{z_m\}_m$ for all edges. Full details of the variational model and updates are provided in the Supplementary Material.

**Variational distributions for** $b_m, z_m$**:** Since $b_m, z_m$ are discrete random variables, we can adopt categorical distributions as their variational distributions. It is noted that when calculating the likelihood for $z_m$, which denotes the latent label of the $m$-th edge $(i_m, j_m, t_m)$, we need to include all the edges triggered by the $m$-th edge (i.e. $(j_{m'}, i_{m'}, t_{m'}) : b_{m'} = m \cap (i_m, j_m) = (j_{m'}, i_{m'})$).

In order to reduce the computational cost of the Gaussian process regression, we use the sparse Gaussian process approximation [27] to calculate $\mathbb{E}_q[f_k(t)], \mathbb{E}_q[f_k^2(t)], \mathbb{E}_q[g_k(t)]$ and $\mathbb{E}_q[g_k^2(t)]$, making it scale to the number of edges [5].

**Computational complexity:** The CTEM models provide an efficient solution to large-scale continuous-time edge data modelling. Let $N_E$ denote the number of observed edges, $N_{ET} = \sum_m \#\{F_{i_m j_m}(t_m)\}$ denote the number of potential triggering phenomena between all oppositely-directed edges, $N_{IN}$ denote the number of inducing points for sparse Gaussian process approximation and $K$ denote the number of latent features. The computational cost of the CTEM models then scales with $\mathcal{O}((N_{ET} + N_E)K + N_E N_{IN}^2 K)$, which scales linearly to the number of edges and would be suitable for large-scale continuous-time edge datasets.

## 4    Related work

Of the various approaches which use the ME-HP to model continuous-time edge data, the Hawkes Infinite Relational Model (Hawkes-IRM) [3] was the first to apply the ME-HP on the edges between groups formed by the nodes partition. By using the stochastic block model [22, 21, 14], the Hawkes-IRM model adopted static exogenous rates. The Hawkes-Dual Latent Space Embedding model (Hawkes-DLS) [29] created latent space vectors for nodes and used the products over these vectors to construct the exogenous and endogenous functions in the ME-HP. [26] adopted the nested-CRP [11, 1] to construct a hierarchical structure in the exogenous function when using the ME-HP

Table 1: Continuous-time edge dataset information. $N$ is the number of nodes, $N_E$ is the number of unique edges, $N_{EC}$ is the number of edges (including their counts), and $T$ is the number of days.

| Dataset | $N$ | $N_E$ | $N_{EC}$ | $T$ | Dataset | $N$ | $N_E$ | $N_{EC}$ | $T$ |
|---|---|---|---|---|---|---|---|---|---|
| Email | 793 | 1 032 | 81 371 | 17.34 | College | 1 417 | 15 329 | 41 371 | 21.37 |
| Overflow | 7 274 | 21 291 | 41 710 | 133.60 | Ubuntu | 71 8576 | 24 062 | 47 298 | 147.35 |

for continuous-time edges. [20] applied the same ME-HP in the framework of sparse exchangeable random measures [4, 28], and obtained continuous-time sparse exchangeable random measures. [30] adopted the Hierarchical Gamma Process to construct static exogenous function rates.

We note that the marginal probability of $z_m$, obtained by summing over the values of $b_m$ in their joint probability function $P(z_m, b_m)$ in NP-HP-C, is similar to the one used in the Dirichlet-Hawkes process (DHP) and its variants [7, 19, 24]. The NP-HP-C model has three major differences with these DHP models: 1) These DHP models consider the particular latent feature in triggering the current event. In contrast, the NP-HP-C model simultaneously specifies the particular latent feature (through $z_m$) and the particular historical oppositely-directed edges (through $b_m$); 2) We are able to develop a fast variational inference algorithm for this model (Section 3.4), whereas inference for DHP models relies on Sequential Monte Carlo methods; 3) The NP-HP-C model describes continuous-time edge data, whereas DHP models are mainly used for streaming text data.

Instead of using point processes to model the continuous-time edges, the approaches of [9, 10] studied time-discretised networks and used Bernoulli emission distributions to model all the binary-valued edges at each observed timestamp. They adopted Gaussian processes/Nested Gaussian processes to generate continuous-time features for each node at the observed timestamps. The approach of [18] also considered continuous-time edges. It extended the Poisson IRM by using histogram or kernel approximations to model the time-varying exogenous rate function, whereas our NP-PP uses Sigmoidal Gaussian Processes approximation.

# 5    Experiments

We perform experiments on four continuous-time edge datasets (Table 1) collected from the Stanford Large Social Network Dataset [16]. In each dataset the occurrence time of an edge is recorded to the nearest second. We scale them onto the interval $[0, T]$ by dividing the recorded time by $24 \times 3600$, so that $T$ is the total number of observation days. The first two datasets (*Email*, *College*) comprise instant response-typed edges (replying to emails and sending messages), while the last two (*Overflow*, *Ubuntu*) comprise edges recording relatively slow replies (writing answers to questions or commenting on others' answers).

**Experimental setup:** We use the Gaussian kernel functions, $\kappa_k^{(f)}(t_i, t_j) = \theta_k^{(f)} e^{-0.5(t_i - t_j)^2/(\delta_k^{(f)})^2}$ and $\kappa_k^{(g)}(t_i, t_j) = \theta_k^{(g)} e^{-0.5(t_i - t_j)^2/(\delta_k^{(g)})^2}$, in the Gaussian processes for generating random functions $\{f_k(t)\}_{k=1}^K$ and $\{g_k(t)\}_{k=1}^K$ respectively, where $\theta_k^{(f)}, \theta_k^{(g)}$ are scaling parameters and $\delta_k^{(f)}, \delta_k^{(g)}$ are bandwidth parameters. These parameters are optimized by applying the ADAM algorithm [15] on the Evidence Lower Bound of the variational distributions. The parameters $a_v, b_v$ of the endogenous function scaling values $v_k$ and the parameters $a_\pi, b_\pi$ of the latent feature $\pi_{ik}$ are given a prior distribution of $\text{Gam}(0.1, 0.1)$ and optimized through the variational inference. We usually set 400 iterations for the MF-VI algorithm. Without particular specifications, we set the number of latent features as $K = 5$.

**Comparison methods:** The CTEM models are compared to the following benchmark methods: (1) the Poisson-Infinite Relational Model (Poisson-IRM) [3]; (2) the Hawkes-Infinite Relational Model (Hawkes-IRM) [3]; (3) the Hawkes Dual Latent Space model (Hawkes-DLS) [29]; and (4) Hawkes process with overlapping communities (HawkesNetOC) [20]. We use the exponential function as the endogenous functions and implement Markov chain Monte Carlo (MCMC) methods for the Poisson-IRM and the Hawkes-IRM models to the best of our ability. Inference for the Hawkes-DLS model and Hawkes-NetOC is executed by using the authors' provided code. The default settings of Hawkes-NetOC and and Hawkes-DLS are used in the comparison. We use the previously developed CTM model constructions, giving the non-parametric Poisson process (NP-PP),

Table 2: Testing log-likelihood and Area Under ROC Curve (AUC).

| Models | Testing Log-Likelihood | | | | AUC | | | |
|---|---|---|---|---|---|---|---|---|
| | *Email* | *College* | *Overflow* | *Ubuntu* | *Email* | *College* | *Overflow* | *Ubuntu* |
| Poisson-IRM | $-62\,320$ | $-58\,552$ | $-90\,786$ | $-96\,630$ | 77.46 | 79.18 | 83.14 | 83.20 |
| Hawkes-IRM | $-58\,341$ | $-59\,461$ | $-88\,746$ | $-90\,435$ | 79.42 | 80.14 | 85.16 | 84.14 |
| Hawkes-DLS | $-59\,810$ | $-56\,237$ | $-97\,505$ | $-93\,485$ | 80.15 | 80.16 | 81.63 | 80.19 |
| HawkesNetOC | $-57\,483$ | $-57\,706$ | $\mathbf{-88\,193}$ | $\mathbf{-89\,850}$ | 83.17 | 81.38 | 83.48 | 82.24 |
| NP-PP | $-68\,475$ | $-71\,154$ | $-133\,839$ | $-136\,830$ | 78.16 | 79.71 | 89.15 | 86.77 |
| NP-HP | $-53\,121$ | $-55\,729$ | $-114\,277$ | $-120\,419$ | 81.78 | 81.57 | 89.58 | 86.78 |
| NP-HP-C | $\mathbf{-50\,126}$ | $\mathbf{-53\,958}$ | $-125\,387$ | $-134\,027$ | **85.29** | **82.17** | **89.91** | **88.02** |

non-parametric Hawkes process (NP-HP), and non-parametric Hawkes process with edge clustering (NP-HP-C).

## 5.1 Testing Log-likelihood and link-prediction performance

The performance of each model is evaluated through its model fitting performance to the Testing edges and its capability to predict future edges. The edges of each dataset are sorted according to their occurrence times, and the dataset split into a training set (the first 70% of edges) and a testing set (the remaining 30%). The comparisons between the CTEM and other models are based on two criteria: the log-likelihood for the edges in the testing time period, and the AUC values on randomly sampled sub-interval in the testing time period. We use Eq. (2) to calculate the log-likelihood, in which the intensity function $\lambda_{ij}(t)$ is either the exogenous function (for Poisson processes) or a combination of the exogenous and endogenous functions (for Hawkes processes).

For the AUC calculation, we first uniformly sample 100 time stamps $t_i$ in the time interval of the test dataset, and then evaluate the probability of an edge appearing between all nodal pairs in the time window $[t_i, t_i + \delta)$. We compute AUC values for each time window $[t_i, t_i + \delta)$. The final link-prediction performance score is the mean of these 100 AUC values on all time windows.

The results are displayed in Table 2. Due to the space limit, we have provided the standard error in the Supplementary Material. The CTEM models perform differently for the two types of datasets. For the instant-response datasets, the NP-HP and NP-HP-C produces better performance in both testing log-likelihood and AUC. This may indicate that our exogenous functions and endogenous functions are fitted better than the comparison methods. However, for the slow-response datasets, the CTEM models performs worse in the testing log-likelihoods, although it still produces better prediction performance. A possible reason is that these two datasets have relatively simple patterns (here low-response activities) which can be well captured by simple rate functions. For instance, the exogenous functions for these slow-response datasets did not show clear periodicities (see the right two panels in the top row of Figure 2). Simple rate functions like constant functions may well describe the exogenous effects. For the endogenous effects, they are dominated by one single endogenous function in both Overflow and Ubuntu (see the right two panels in the bottom row of Figure 2). One single exponential decay function may also capture these simple patterns.

## 5.2 Visualisations for exogenous functions and endogenous functions

The top row in Figure 2 presents the mean exogenous function values $\sum_{(i,j):i \neq j} \sum_k \pi_{ik} \pi_{jk} \cdot \sigma(f_k(t))$ for the NP-PP, NP-HP and NP-HP-C models. With the vertical dotted lines separating the time lines into weeks, daily periodicity is quite clear for the instant-response datasets (i.e. *Email*, *College*), and approximate weekly periodicity for *Overflow* and *Ubuntu*. The weekend effect is also easily identifiable in *Email*, a gradual increasing trend in location and scale for *College*, and occasional weeks of peak high and low activity for the slow response datasets. For the latter, the large activity spikes could be explained by sudden interest in popular topics being discussed on the message boards.

In terms of performance difference within the CTEM models, the exogenous values of the NP-PP model (blue lines) are always higher than those of the NP-HP and NP-HP-C models, and the NP-HP (green lines) usually produces the smallest values. The former may occur as the NP-PP model is using the exogenous functions to explain all edges as it does not have endogenous functions. The latter may occur as the NP-HP models less edges than the NP-HP-C. For the slow-response datasets,

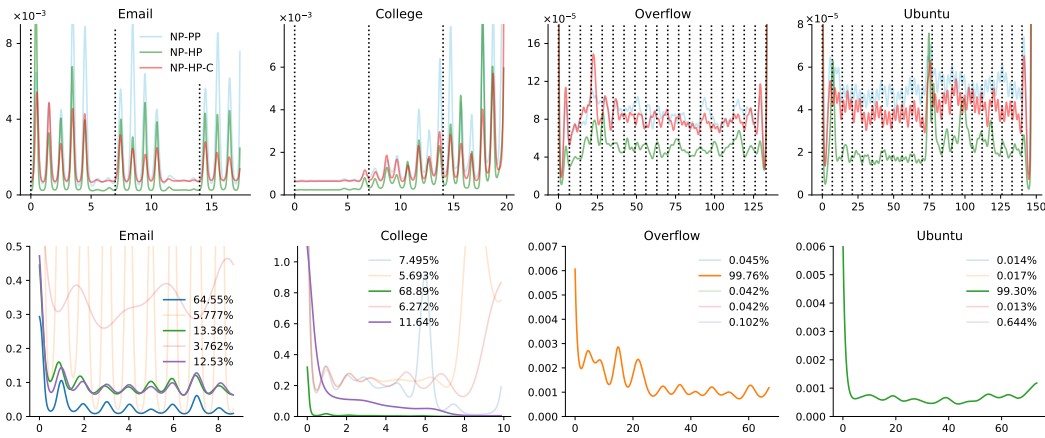

Figure 2: Top row: mean exogenous functions $\frac{1}{N(N-1)} \sum_{i \neq j} \sum_k \pi_{ik} \pi_{jk} \cdot \sigma(f_k(t))$ for the NP-PP, NP-HP and NP-HP-C models on each dataset. The axes show the the exogenous rates ($y$-axis) and the number of days ($x$-axis). Vertical dotted lines in each panel indicate weekly intervals. Bottom row: fitted endogenous functions $\{v_k \cdot \sigma(g_k(t))\}_k$ for NP-HP-C on each dataset. Panel legends indicate the proportion of triggered edges explained by each function. In order to improve visualisation, exogenous functions with weight under 0.1 are not shown.

the average values are similar for the NP-PP and NP-HP-C models. This may be explained as the endogenous effects may be more important to the instant-response data than the slow-response data.

The bottom row in Figure 2 illustrates the most influential endogenous functions of the NP-HP-C model. Within our expectation that endogenous effects would become weaker along with the time, the values of all main endogenous functions have sharp decreases in the beginning period. Three of the datasets also exhibit (daily or weekly) periodicity, which indicates that for these data there has been an increase in triggered responses above and beyond the increase in events attributable to background (exogenous) rates. This is less obviously the case for the *College* dataset, where, despite some initial periodicity, many of the endogenous functions become constant after a certain time. We are unaware of any related discussions of periodic endogenous functions for these data in the literature.

We have provided visualisations for other variables in the Supplementary Material.

### 5.3 Intensity functions $\lambda_{ij}(t)$ and influence of the number of latent features

The left two panels of Figure 3 show the intensity functions $\lambda_{ij}(t)$ of the NP-HP-C for randomly selected nodal pairs on an instant-response (*Email*) and a slow-response (*Overflow*) dataset. The daily periodicity is again strongly apparent in $\lambda_{ij}(t)$ for *Email*, and more muted weekly periodicity for $\lambda_{ij}(t)$ in *Overflow*. We note that most of the observed edges (points on the $x$-axis) are located in regions where the intensity increases, indicating correct performance.

The right two panels of Figure 3 show the influence of different number of latent features $K$. Predictive performance appears less sensitive to $K$ for the instant-response datasets, which perhaps have simpler structures, than the slow-response datasets, as the latter have improved performance with increasing $K$. Consistent with the complexity analysis, running times are linear in the number of edges. As the NP-HP-C with $K = 5$ requires $\sim 60$ seconds for one iteration with $80\,000$ edges, we need $\sim 6$ hours of for posterior inference on these edges (with $400$ iterations).

## 6 Conclusion

In this work, we have explored three non-parametric point process models for continuous-time edge data. The sigmoidal Gaussian process priors for the exogenous and endogenous functions enable flexible modelling, and strong predictive performance. The developed mean-field variational

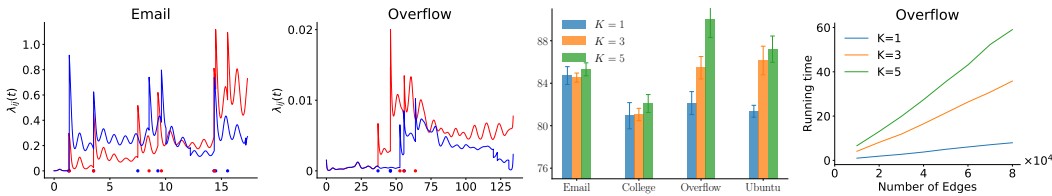

Figure 3: Left two panels: example nodal pair-intensity function $\lambda_{ij}(t)$ for NP-HP-C on *Email* and *Overflow*. middle right panel: AUC prediction performance for NP-HP-C with $K = 1, 3, 5$. Right most: Running time (seconds per iteration on a laptop) comparison for NP-HP-C with $K = 1, 3, 5$.

inference algorithm allows the models to work at scale. The experimental results on four real world data sets demonstrate superior performance compared to the state-of-the-art.

## Acknowledgements

Xuhui Fan and Scott A. Sisson are supported by the Australian Research Council through the Australian Centre of Excellence in Mathematical and Statistical Frontiers (ACEMS, CE140100049), and Scott A. Sisson through the Future Fellow Scheme (FT170100079). Bin Li is supported by the Program for Professor of Special Appointment (Eastern Scholar) at Shanghai Institutions of Higher Learning. Feng Zhou is partially funded by China Postdoctoral Science Foundation.

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
