# Supplementary material: Continuous-time edge modelling using non-parametric point processes

**Xuhui Fan**[1], **Bin Li**[2], **Feng Zhou**[3], and **Scott A. Sisson**[1]

[1]UNSW Data Science Hub, and School of Mathematics & Statistics, University of New South Wales
[2]Shanghai Key Lab of IIP, School of Computer Science, Fudan University
[3]Department of Computer Science & Technology, Tsinghua University,
*{xuhui.fan, scott.sisson}@unsw.edu.au; libin@fudan.edu.cn; zhoufeng6288@tsinghua.edu.cn*

## 1 Review on Multivariate Hawkes process

There are other approaches in using Hawkes process to model continuous-time edge data. For example, Multivariate Hawkes processes [5, 9, 6, 10, 4, 3, 8]. regard each node's activities (e.g. sending or receiving messages) as the events in one particular dimension. For a total $N$ nodes, we have an $N$-dimensional Hawkes process. The edges between nodes are inferred through the dimension-wise endogenous matrix. These models miss some important aspects of information. For example, their observations simply records the one-way direction of nodes sending out edges and does not record the information of receiving edges, whereas the observations for the ConTIM model and the models described above also record the receiver node's information. Even more restrictively, these multivariate models also usually infer a static network structure.

## 2 Experiments

### 2.1 Initial results on synthetic data

We first provide initial model fitting results on synthetic data to validate the effectiveness of our model. We use NP-PP as the fitted model, focusing on exogenous functions and the corresponding latent exogenous features. We set $N = 50, T = 5, K = 2$ and use Gaussian process to generate random functions to form the exogenous functions.

Figure 1 shows the average groundtruth exogenous functions and our fitted exogenous functions. It is easy to see that our model can correctively identify the peak regions of the exogenous functions.

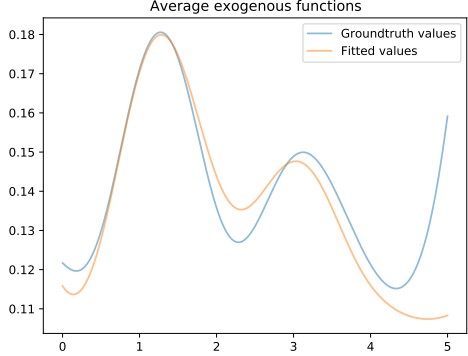

Figure 1: Average exogenous function values on synthetic data.

35th Conference on Neural Information Processing Systems (NeurIPS 2021).

Table 1: Testing log-likelihood and Area Under ROC Curve (AUC).

| Models | Testing Log-Likelihood | | | |
|---|---|---|---|---|
| | *Email* | *College* | *Overflow* | *Ubuntu* |
| Poisson-IRM | $-62\,320 \pm 769$ | $-58\,552 \pm 575$ | $-90\,786 \pm 797$ | $-96\,630 \pm 1\,107$ |
| Hawkes-IRM | $-58\,341 \pm 334$ | $-59\,461 \pm 1\,448$ | $-88\,746 \pm 1\,043$ | $-90\,435 \pm 376$ |
| Hawkes-DLS | $-59\,810 \pm 859$ | $-56\,237 \pm 1\,207$ | $-97\,505 \pm 332$ | $-93\,485 \pm 299$ |
| HawkesNetOC | $-57\,483 \pm 550$ | $-57\,706 \pm 424$ | $\mathbf{-88\,193} \pm 582$ | $-89\,850 \pm 1\,022$ |
| NP-PP | $-68\,475 \pm 1\,274$ | $-71\,154 \pm 549$ | $-133\,839 \pm 829$ | $-136\,830 \pm 810$ |
| NP-HP | $-53\,121 \pm 964$ | $-55\,729 \pm 813$ | $-114\,277 \pm 985$ | $-120\,419 \pm 796$ |
| NP-HP-C | $\mathbf{-50\,126} \pm 568$ | $\mathbf{-53\,958} \pm 475$ | $-125\,387 \pm 376$ | $-134\,027 \pm 394$ |

| Models | AUC | | | |
|---|---|---|---|---|
| | *Email* | *College* | *Overflow* | *Ubuntu* |
| Poisson-IRM | $77.46 \pm 0.87$ | $79.18 \pm 1.07$ | $83.14 \pm 0.93$ | $83.20 \pm 0.74$ |
| Hawkes-IRM | $79.42 \pm 0.41$ | $80.14 \pm 1.28$ | $85.16 \pm 0.85$ | $84.14 \pm 0.97$ |
| Hawkes-DLS | $80.15 \pm 1.02$ | $80.16 \pm 1.07$ | $81.63 \pm 0.46$ | $80.19 \pm 1.03$ |
| HawkesNetOC | $83.17 \pm 1.30$ | $81.38 \pm 0.89$ | $83.48 \pm 0.79$ | $82.24 \pm 0.94$ |
| NP-PP | $78.16 \pm 0.94$ | $79.71 \pm 0.92$ | $89.15 \pm 0.74$ | $86.77 \pm 0.90$ |
| NP-HP | $81.78 \pm 0.97$ | $81.57 \pm 0.64$ | $89.58 \pm 0.98$ | $86.78 \pm 0.85$ |
| NP-HP-C | $\mathbf{85.29} \pm 0.88$ | $\mathbf{82.17} \pm 0.65$ | $\mathbf{89.91} \pm 0.83$ | $\mathbf{88.02} \pm 0.38$ |

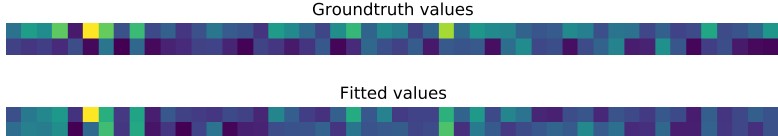

Figure 2: Visualisations for latent features $\boldsymbol{\pi}$ on synthetic data.

Figure 2 also shows the corresponding latent exogenous features. Nodes with large values of latent exogenous features are also identified through our NP-PP.

## 2.2 Link Prediction performance (with standard error reported)

Table 1 provides the link prediction performance, with the standard error reported.

### 2.2.1 Variable visualisations on all the edges

We provide visualisations for random variables, which are not shown in the main content, of our NP-PP, NP-HP, NP-HP-C.

Figure 3 visualises the triggering factors of follow-up edges, which contains historical oppositely-directed edge. We first remove the initial edges within nodal pairs, which are the ones without historical oppositely-directed ones. Then, we displays the probabilities of remaining edges instantiated by the exogenous rates for all these approaches. We can see that, for NP-PP and NP-HP-C, the instant-response datasets (*Email*, *College*) contain more follow-up typed edges than the slow-response datasets (*Overflow*, *Ubuntu*). For all these datasets, the probabilities of the follow-up typed edges belonging to the exogenous rates are usually very small (i.e. $P(b_m = 0) < 0.05$). That is, almost all these follow-up typed edges are triggered by its historical oppositely-directed edges for NP-PP and NP-HP-C.

Figure 4 visualises average values of the latent label vector $P(z_m)$, which is the mean of 50 consecutive edges. The instant-response datasets have larger variance in latent labels, whereas the slow-response data usually have smaller variance in latent labels. Especially, we find that our NP-HP-C seems favours one dominate latent label in the slow-response datasets, which is likely because its clusters edges mechanism.

Figure 5 shows 5 exogenous functions $(\sigma(f_k(t)))$ for all the three approaches. Firstly, we can see that there isn't much variations for the latent features' proportions, which is calculated as $\sum_{i \neq j} \pi_{ik}\pi_{jk} / \sum_k \sum_{i \neq j} \pi_{ik}\pi_{jk}$, as all these values are around the average proportion of 20%. The

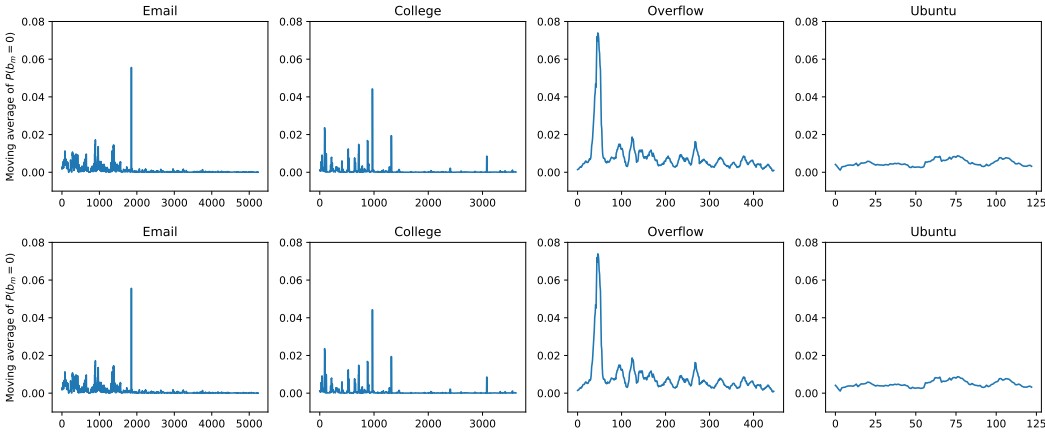

Figure 3: Row 1: NP-PP; row 2: NP-HP-C. Moving average values (with lags $= 100$) of $P(b_m = 0)$ for the edges with historical oppositely-directed edges in the datasets *Email*, *College*, *Overflow* and *Ubuntu* (from left to right). $x$-axis represents the edge number $m = 1, \ldots, 10\,000$.

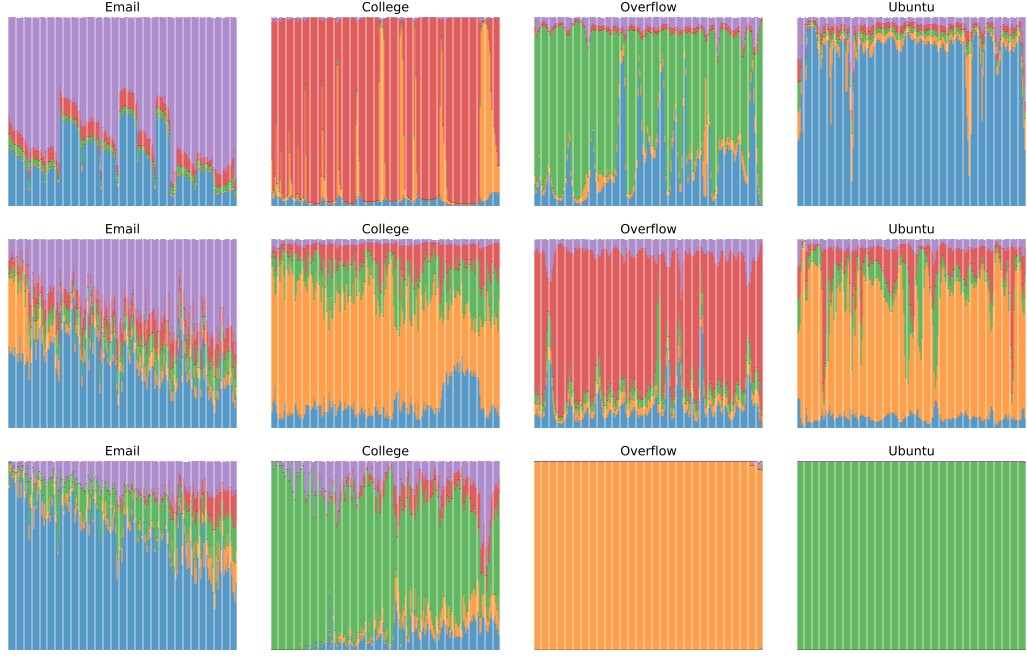

Figure 4: From top to bottom: NP-PP, NP-HP, NP-HP-C. Visualisations of $\{\bar{\boldsymbol{\phi}}_{m'}\}_{m'}$ for *Email*, *College*, *Overflow* and *Ubuntu*. $\bar{\boldsymbol{\phi}}_{m'} = \frac{1}{50} \sum_{m''=50m'-49}^{50m'} P(z_{m''})$ represents averages values of $P(z_m)$ in each 50 observations.

scaling differences of $\sigma(f_k(t))$ may be presented by themselves. Secondly, we find almost all these functions presents some periodicities. This periodicities in instant-response datasets are much clearer than those in slow-response datasets. Finally, our NP-HP-C seems favours one dominate function and suppress others in representing the model, whereas all the exogenous functions seem to present the patterns for the other two approaches. This may possibly due to that we are clustering edges and using the edge's latent label to determine its endogenous functions.

Figure 6 provides the endogenous function visualisations for NP-HP, as the related visualization for NP-HP-C is provided in the main content. We find that the endogenous functions in NP-HP seems

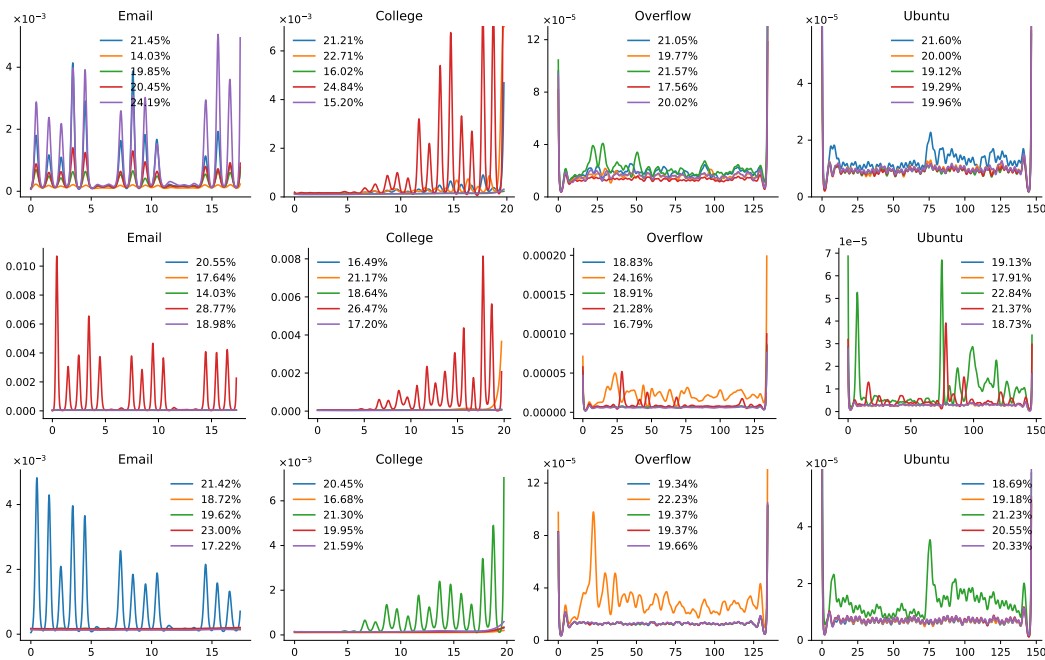

Figure 5: From top to bottom: NP-PP, NP-HP, NP-HP-C. Visualisations on the 5 exogenous functions for our three approaches. The legend for each exogenous function presents its proportion of global latent feature value $\sum_{i \neq j} \pi_{i_m k} \pi_{j_m k}$. Row 1: Poisson Processes with nodal pair-specific intensity functions; row 2, Hawkes Processes with nodal pair-specific exogenous functions and nodal pair-specific endogenous functions; row 3, Hawkes Processes with nodal pair-specific exogenous functions and clustered endogenous functions.

perform a bit different from our expectation. The values do not always have a sharp decrease in the beginning period. The patterns also seem less regular as those in NP-HP-C. This may affect the prediction capability for NP-HP, as shown in the performance table in the main content.

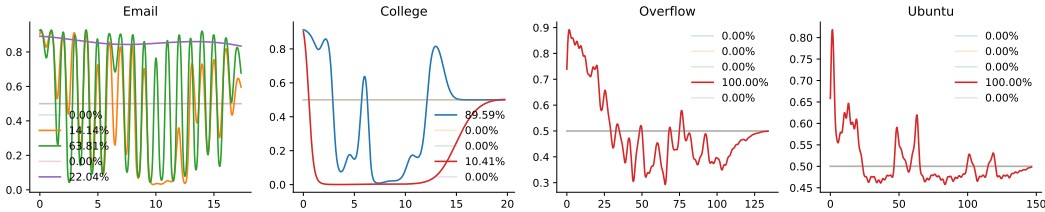

Figure 6: Endogenous functions $\{\sigma(g_k(t))\}_k$ for NP-HP. The legends in each panel represents their proportions in the endogenous function triggered edges, i.e., $\sum_m \sum_{m' \in F_{i_m j_m}(t_m)} \psi_{m,m'} \phi_{m,k}$. In order to better visualise the "main" endogenous functions, we transparent the exogenous function if its weight are less than 10%.

# 3   Variational Inference for Poisson Processes with $K$ random functions

**Likelihood of observation** The likelihood of the time observations $\{t_m\}_m$ for ConTIM can be expressed as:

$$P(\{t_m\}_m|-) = \prod_k \left[ \prod_{i\neq j} e^{\int_0^T \pi_{ik}\pi_{jk}\sigma(f_k(t))dt} \right] \cdot \prod_k \prod_m \left[ \pi_{i_m k}\pi_{j_m k}\sigma(f_k(t_m)) \right]^{\mathbf{1}_{z=k}(z_m)} \quad (1)$$

In Eq. (1), the sigmoid function $\sigma(\cdot)$ and the Gaussian process function $\{f_k(\cdot)\}_k$ make it difficult to form convenient format to enable mean-field variational inference. Thus, we use the technique of [2] to overcome these issues. In general, [2] would use latent marked Poisson processes and Polya-Gamma random variables to augment the model.

$$P\left(\{t_m\}_m, \left\{\omega_{m,k}^{(f)}\right\}_{k,m}, \left\{\Pi_{ij,k}^{(f)}\right\}_{k,i\neq j}|-\right)$$
$$= \prod_k \prod_m \left[ (\pi_{i_m k}\pi_{j_m k} e^{h(\omega_{m,k}^{(f)}, f_k(t_m))})^{\mathbf{1}_{z=k}(z_m)} P_{\text{PG}}(\omega_{m,k}^{(f)}|1,0) \right]$$
$$\cdot \prod_k \prod_{i\neq j} \left[ \prod_{(x,\omega)_n \in \Pi_{k,ij}^{(f)}} \left( e^{h(\omega_{k,ij,n}^{(f)}, -f_k(x_{k,ij,n}^{(f)}))} \pi_{ik}\pi_{jk} P_{\text{PG}}(\omega_{k,ij,n}^{(f)}|1,0) \right) e^{-\pi_{ik}\pi_{jk}T} \right] \quad (2)$$

where $h(\omega, x) = x/2 - \omega x^2/2 - \log 2$, $\omega_{m,k}^{(f)}, \sim P_{\text{PG}}(\omega|1,0)$, $\Pi_{k,ij}^{(f)}$ is the marked Poisson process with intensity $\pi_{i_m k}\pi_{j_m k}P_{\text{PG}}(\omega|1,0)dtd\omega$ in the support of $(0,T] \times \mathbb{R}^+$.

$$\log P\left(\{t_m\}_m, \left\{\omega_{m,k}^{(f)}\right\}_{k,m}, \left\{\Pi_{ij,k}^{(f)}\right\}_{k,i\neq j}|-\right)$$
$$= \sum_k \sum_m \left[ \mathbf{1}_{z=k}(z_m)\cdot(\log(\pi_{i_m k}\pi_{j_m k}) + h(\omega_{m,k}^{(f)}, f_k(t_m))) + \log P_{\text{PG}}(\omega_{m,k}^{(f)}|1,0) \right]$$
$$+ \sum_k \sum_{i\neq j} \left[ \left( \sum_{(x,\omega)_n \in \Pi_{k,ij}^{(f)}} h(\omega_{k,ij,n}^{(f)}, -f_k(x_{k,ij,n}^{(f)})) + \log(\pi_{ik}\pi_{jk}\cdot P_{\text{PG}}(\omega_{k,ij,n}^{(f)}|1,0)) \right) - \pi_{ik}\pi_{jk}T \right]$$
$$= \sum_k \sum_m \left[ \mathbf{1}_{z=k}(z_m)\cdot\left(\log(\pi_{i_m k}\pi_{j_m k}) + \frac{f_k(t_m)}{2} - \frac{(f_k(t_m))^2}{2}\omega_{m,k}^{(f)} - \log(2)\right) + \log P_{\text{PG}}(\omega_{m,k}^{(f)}|1,0) \right]$$
$$+ \sum_k \sum_{i\neq j} \left[ \sum_{(x,\omega)_n \in \Pi_{k,ij}^{(f)}} \left( -\frac{f_k(t_{k,ij,n}^{(f)})}{2} - \frac{(f_k(t_{k,ij,n}^{(f)}))^2}{2}\omega_{k,ij,n}^{(f)} - \log(2) + \log(\pi_{ik}\pi_{jk}\cdot P_{\text{PG}}(\omega_{k,ij,n}^{(f)}|1,0)) \right) - \pi_{ik}\pi_{jk}T \right]$$
$$(3)$$

$$\mathbb{E}_q\left[ \log P\left(\{t_m\}_m, \left\{\omega_{m,k}^{(f)}\right\}_{k,m}, \left\{\Pi_{ij,k}^{(f)}\right\}_{k,i\neq j}, |-\right) \right]$$
$$= \sum_k \sum_m \left[ \phi_{m,k}\cdot\mathbb{E}_q\left(\log(\pi_{i_m k}\pi_{j_m k}) + \frac{f_k(t_m)}{2} - \frac{(f_k(t_m))^2}{2}\omega_{m,k}^{(f)} - \log(2)\right) + \mathbb{E}_q[\log P_{\text{PG}}(\omega_{m,k}^{(f)}|1,0)] \right]$$
$$+ \sum_{i\neq j}\sum_k \left[ \int_0^\infty \int_0^T \left( -\frac{f_k(t)}{2} - \frac{(f_k(t))^2}{2}\omega - \log(2) + \log(\pi_{i_m k}\pi_{j_m k}P_{\text{PG}}(\omega|1,0)) \right) \Lambda_{k,ij}^{(f)}(t,\omega)dtd\omega - \mathbb{E}_q[\pi_{i_m k}\pi_{j_m k}]T \right]$$
$$(4)$$

We make the following adjustments to make the method of [2] fit in our model. For $f_k(t)$, we incorporate auxiliary Polya-Gamma variables ($\omega_{m,k}^{(f)} \sim P_{\text{PG}}(\omega|1,0)$) [7] for each edge $(i_m, j_m, t_m)$ and $N^2 - N$ marked Poisson processes for all the potential nodal pairs (each with the intensity function $\pi_{i_m k}\pi_{j_m k}P_{\text{PG}}(\omega|1,0)dtd\omega$).

By using standard method of the mean-field variational inference method [1], we can obtain each random variable's optimal variational distribution through the following equations:

$$\ln q_{x_i}(x_i) = \mathbb{E}_{q_{\backslash x_i}}\left[\ln L(\mathcal{D}, \{x_i\}_i)\right] + \text{const} \tag{5}$$

Eq. (129) will be used in deriving the optimal variational distribution for all the random variables.

**Optimal Polya-Gamma density** $q(\omega_{m,k}^{(f)})$

$$\log q(\omega_{m,k}^{(f)}) = \phi_{m,k}\mathbb{E}_q\left[-f_k^2(t_m)\omega_{m,k}^{(f)}/2\right] + \log P_{\text{PG}}(\omega_{m,k}^{(f)}|1,0) + \text{const} \tag{6}$$

Thus, we get

$$q(\omega_{m,k}^{(f)}) \propto \exp\left[-\phi_{m,k}\mathbb{E}_q(f_k^2(t_m))\omega_{m,k}/2\right] \cdot P_{\text{PG}}(\omega_{m,k}^{(f)}|1,0) \tag{7}$$

which leads to

$$q(\omega_{m,k}^{(f)}) = P_{\text{PG}}\left(\omega_{m,k}^{(f)}|1, \sqrt{\phi_{m,k}\mathbb{E}_q(f_k^2(t_m))}\right) \tag{8}$$

We can let $c_{m,k}^{(f)} = \sqrt{\phi_{m,k}\mathbb{E}_q(f_k^2(t_m))}$.

**Optimal Poisson process** $q(\Pi_{k,ij}^{(f)})$ Using the mean-field update equation, we get the rate functions for the latent marked Poisson processes as:

$$\Lambda_{k,ij}^{(f)}(t,\omega) = \frac{\exp(\mathbb{E}_q\left[\log \pi_{ik}\pi_{jk}\right] - \frac{\mathbb{E}_q[f_k(t)]}{2})}{2\cosh(\frac{c_k^{(f)}(t)}{2})} P_{\text{PG}}(\omega|1, c_k^{(f)}(t)) \tag{9}$$

where $c_k^{(f)}(t) = \sqrt{\mathbb{E}_q[f_k(t)^2]}$. Again, we emphasize that the support of $\Lambda_{k,ij}^{(f)}(t,\omega)$ is $(0,T] \times \mathbb{R}^+$.

**Optimal latent label** $q(z_m)$

$$\begin{aligned}
&\log \phi_{m,k} \\
=&\mathbb{E}_q\left[\left(\log(\pi_{i_mk}\pi_{j_mk}) + h(\omega_{m,k}^{(f)}, f_k(t_m))\right)\right] \\
=&\mathbb{E}_q[\log \pi_{i_mk}] + \mathbb{E}_q[\log \pi_{j_mk}] + \frac{\mathbb{E}[f_k(t_m)]}{2} - \frac{\mathbb{E}[f_k(t_m)^2]}{2}\mathbb{E}_q[\omega_{m,k}^{(f)}] - \log 2 \tag{10}
\end{aligned}$$

**Optimal feature vector** $\boldsymbol{\pi}_i$ The optimal density of $\pi_{ik}$ is a Gamma distribution as:

$$q(\pi_{ik}) \propto [\pi_{ik}]^{a_{\pi,ik}^{(q)}-1} e^{-b_{\pi,ik}^{(q)}\pi_{ik}} \tag{11}$$

where $a_{\pi,ik}^{(q)} = \sum_{j:j\neq i}\int_0^T (\Lambda_{k,ij}(t) + \Lambda_{k,ji}(t))dt + a_k + \sum_m \phi_{m,k}\mathbf{1}_{i_m=i}\mathbf{1}_{j_m=i}, b_{\pi,ik}^{(q)} = b_k + 2(\sum_{j:j\neq i}\pi_{jk})T$.

**Optimal Gaussian processes** $f_k(t)$

$$\log q(f_k) \propto \exp(U(f_k)) \tag{12}$$

$U(f_k)$ is defined as:

$$\begin{aligned}
&U(f_k) \\
=&\mathbb{E}_q\left[\sum_{i\neq j}\sum_{(x,\omega)_n \in \Pi_{k,ij}^{(f)}} h(\omega_{k,ij,n}^{(f)}, -f_k(x_{k,ij,n}^{(f)}))\right] + \sum_m \mathbb{E}_q\left[h(\omega_{m,k}^{(f)}, f_k(t_m))\right] \\
=&-\frac{1}{2}\int_{\mathcal{T}} A(t)f_k^2(t)dt + \int_{\mathcal{T}} B(x)f_k^2(t)dt \tag{13}
\end{aligned}$$

where

$$A(t) = \sum_m \phi_{m,k} \mathbb{E}_q[\omega_{m,k}^{(f)}] \delta(t - t_m) + \sum_{i \neq j} \int_0^\infty \omega \Lambda_{k,ij}^{(f)}(t, \omega) d\omega \tag{14}$$

$$B(t) = \frac{1}{2} \sum_m \phi_{m,k} \delta(t - t_m) - \frac{1}{2} \sum_{i \neq j} \int_0^\infty \Lambda_{k,ij}^{(f)}(t, \omega) d\omega \tag{15}$$

in which we have the following equations for the above integration:

$$\mathbb{E}_q[\omega_{m,k}^{(f)}] = \frac{1}{2c_{k,m}^{(f)}} \tanh\left(\frac{c_{k,m}^{(f)}}{2}\right) \tag{16}$$

$$\int_{\mathcal{T}} \Lambda_{k,ij}^{(f)}(t, \omega) d\omega = \Lambda_{k,ij}^{(f)}(t) \tag{17}$$

$$\int_{\mathcal{T}} \omega \Lambda_{k,ij}^{(f)}(t, \omega) d\omega = \frac{1}{2c_{k,ij}^{(f)}(t)} \tanh\left(\frac{c_{k,ij}^{(f)}(t)}{2}\right) \Lambda_{k,ij}^{(f)}(t) \tag{18}$$

By using sparse Gaussian process methods, we get the sparse posterior distribution for the function values at the inducing points as:

$$q^{(s)} = \mathcal{N}(\boldsymbol{\mu}^{(s)}, \Sigma^{(s)}) \tag{19}$$

where the covariance matrix and mean is expressed as:

$$\Sigma^{(s)} = \left[ K_s^{-1} \int_{\mathcal{T}} A(t) k_s(t) k_s^\top dt K_s^{-1} + K_s^{-1} \right]^{-1} \tag{20}$$

$$\boldsymbol{\mu}^{(s)} = \Sigma^{(s)} \left( K_s^{-1} \int_{\mathcal{T}} B(t) k_s(t) dt \right) \tag{21}$$

Given the sparse solution, we can obtain the point estimation as

$$\boldsymbol{\mu}(t) = k_s^\top(t) K_s^{-1} \boldsymbol{\mu}^{(s)} \tag{22}$$

$$(s(t))^2 = k(t, t) - k_s^\top(t) K_s^{-1} (\boldsymbol{I} - \sum K_s^{-1}) k_s(t) \tag{23}$$

**Update the hyper-parameters of the Kernel functions in the Gaussian process** Given $\Lambda_0(x, \omega) = \lambda P_{PG}(\omega|1, 0) dx d\omega$, the lower bound for the original paper is:

$$
\begin{aligned}
\mathcal{L}(\boldsymbol{q}) = \sum_k \Bigg\{ &\sum_{i \neq j} \Bigg[ \int_0^\infty \int_0^T \Bigg( \mathbb{E}_q[h(\omega, -f_k(x))] + \mathbb{E}_q[\log \pi_{i_m k} \pi_{j_m k}] - \mathbb{E}_q[\log \Lambda_{k,ij}^{(f)}(x)] - \mathbb{E}_q \log \cosh(\frac{c_1(x)}{2}) \\
&+ \mathbb{E}_q[\frac{c_1(x)^2 \omega}{2}] + 1 \Bigg) \Lambda_{k,ij}^{(f)}(x, \omega)) dx d\omega - \mathbb{E}_q[\pi_{i_m k} \pi_{j_m k}] \cdot T \Bigg] \\
&+ \sum_m \Bigg[ \phi_{m,k} \Big( \mathbb{E}_q[\log \pi_{i_m k} \pi_{j_m k}] + \mathbb{E}_q[h(\omega_{m,k}^{(f)}, f_k(t_m))] \Big) - \mathbb{E}_q \log \cosh(\frac{c_{ij,n,k}^{(f)}}{2}) + \mathbb{E}_q[\frac{(c_{ij,n,k}^{(f)})^2 \omega_{m,k}^{(f)}}{2}] \Bigg] \\
&+ \sum_k \Bigg\{ -\frac{1}{2} \mathrm{Tr} \left( [K_k^{(f,s)}]^{-1} (\Sigma_k^{(f,s)} + \boldsymbol{\mu}_k^{(f,s)} [\boldsymbol{\mu}_k^{(f,s)}]^\top) \right) - \frac{1}{2} \log \det K_k^{(f,s)} + \frac{1}{2} \log \det \Sigma_k^{(f,s)} \Bigg\}
\end{aligned}
\tag{24}
$$

$$\frac{\partial \mathcal{L}(\boldsymbol{q})}{\partial \theta_k^{(f)}} = \sum_{i \neq j} \int_0^\infty \int_0^T \frac{\partial \mathbb{E}_q[h(\omega, -f_k(x))]}{\partial \theta_k^{(f)}} \Lambda_{k,ij}^{(f)}(x, \omega)) dx d\omega + \sum_m \frac{\partial \mathbb{E}_q[h(\omega_{m,k}^{(f)}, f_k(t_m))]}{\partial \theta_k^{(f)}}$$

$$- \frac{1}{2} \mathrm{Tr}\left( [K_k^{(f,s)}]^{-1} (\Sigma_k^{(f,s)} + \boldsymbol{\mu}_k^{(f,s)} [\boldsymbol{\mu}_k^{(f,s)}]^\top) \right) - \frac{1}{2} \log \det K_k^{(f,s)}$$

$$= \sum_{i \neq j} \int_0^\infty \int_0^T \frac{\partial \mathbb{E}_q[h(\omega, -f_k(x))]}{\partial \theta_k^{(f)}} \Lambda_{k,ij}^{(f)}(x, \omega)) dx d\omega + \sum_m \phi_{m,k} \frac{\partial \mathbb{E}_q[h(\omega_{m,k}^{(f)}, f_k(t_m))]}{\partial \theta_k^{(f)}}$$

$$+ \frac{1}{2} \mathrm{Tr}\left( [K_k^{(f,s)}]^{-1} \frac{\partial K_k^{(f,s)}}{\partial \theta_k^{(f)}} [K_k^{(f,s)}]^{-1} (\Sigma_k^{(f,s)} + \boldsymbol{\mu}_k^{(f,s)} [\boldsymbol{\mu}_k^{(f,s)}]^\top) \right) - \frac{1}{2} \mathrm{Tr}\left( [K_k^{(f,s)}]^{-1} \frac{\partial K_k^{(f,s)}}{\partial \theta_k^{(f)}} \right)$$

$$(25)$$

The gradient of the Evidence lower bound to the hyper-parameters (we let $\Phi_k^{(f)} := (\theta_k^{(f)}, \delta_k^{(f)})$ in the kernel functions of the Gaussian process) is:

$$\frac{\partial \mathbb{E}_q[h(\omega, f_k(x))]}{\partial \theta_k^{(f)}} = \frac{1}{2} \left( \frac{\partial \mathbb{E}_Q[f_k(x)]}{\partial \theta_k^{(f)}} - \frac{\partial \mathbb{E}_Q\left[f_k(x)^2\right]}{\partial \theta_k^{(f)}} \mathbb{E}_Q[\omega] \right) \tag{26}$$

$$\frac{\partial \mathbb{E}_q[f_k(x)]}{\partial \theta_k^{(f)}} = \frac{\partial \boldsymbol{\kappa}(x)}{\partial \theta_k^{(f)}} \boldsymbol{\mu}_2^{(f,s)} \tag{27}$$

$$\frac{\partial \mathbb{E}_q\left[f_k(x)^2\right]}{\partial \theta_k^{(f)}} = \frac{\partial \tilde{k}(x,x)}{\partial \theta_k^{(f)}} + \frac{\partial \boldsymbol{\kappa}(x)^\top}{\partial \theta_k^{(f)}} \left( \Sigma_2^{(f,s)} + \boldsymbol{\mu}_2^{(f,s)} \left( \boldsymbol{\mu}_2^{(f,s)} \right)^\top \right) \boldsymbol{\kappa}(x)$$

$$+ \boldsymbol{\kappa}(x)^\top \left( \Sigma_2^{(f,s)} + \boldsymbol{\mu}_2^{(f,s)} \left( \boldsymbol{\mu}_2^{(f,s)} \right)^\top \right) \frac{\partial \boldsymbol{\kappa}(x)}{\partial \theta_k^{(f)}} \tag{28}$$

where $\boldsymbol{\kappa}(x) = \boldsymbol{k}_s(x)^\top K_s^{-1}$ and $\tilde{k}(t,t) = k(x,x) - \boldsymbol{k}_s(x) K_s^{-1} \boldsymbol{k}_s(x)^\top$. The remaining two terms are:

$$\frac{\partial \tilde{k}(x,x)}{\partial \theta_k^{(f)}} = \frac{\partial k(x,x)}{\partial \theta_k^{(f)}} - \frac{\partial \boldsymbol{\kappa}(x)}{\partial \theta_k^{(f)}} \boldsymbol{k}_s(x) - \boldsymbol{\kappa}(x) \frac{\partial \boldsymbol{k}_s(t)}{\partial \theta_k^{(f)}} \tag{29}$$

$$\frac{\partial \boldsymbol{\kappa}(x)}{\partial \theta_k^{(f)}} = \frac{\partial \boldsymbol{k}_s(x)^\top}{\partial \theta_k^{(f)}} K_s^{-1} - \boldsymbol{k}_s(x) K_s^{-1} \frac{\partial K_s}{\partial \theta_k^{(f)}} K_s^{-1} \tag{30}$$

After each variational step, the hyper-parameters are updated by

$$\boldsymbol{\theta}_k^{(f)} = \boldsymbol{\theta}_k^{(f)} + \varepsilon \frac{\partial \mathcal{L}(q)}{\partial \boldsymbol{\theta}_k^{(f)}} \tag{31}$$

where $\epsilon$ is the stepsize.

## 4 Variational Inference for nodal pair-specific Exogenous and Endogenous functions

**Likelihood of observation** The likelihood of the time observations $\{t_m\}_m$, latent label $\{z_m\}_m$ for ConTIM can be expressed as:

$$P(\{t_m, z_m\}_m | -)$$

$$= \prod_k \left[ \prod_{i \neq j} e^{-\int_0^T \pi_{ik} \pi_{jk} \sigma(f_k(t)) dt} \cdot \prod_m [\pi_{i_m k} \pi_{j_m k} \sigma(f_k(t_m))]^{\mathbf{1}_{z=(0,k)}(z_m)} \right]$$

$$\cdot \prod_k \prod_m \left[ e^{-v_{i_m k} v_{j_m k} \int_0^{T-t_m} \sigma(g_k(t)) dt} \cdot \prod_{m':m' \in F_m} [v_{i_m k} v_{j_m k} \sigma(g_k(t_m - t_{m'}))]^{\mathbf{1}_{z=(m',k)}(z_m)} \right]$$

where $F_m = \{m' : (i_m, j_m) = (j, i) \cap t_{m'} < t_m\}$. In Eq. (78), the sigmoid function $\sigma(\cdot)$ and the Gaussian process function $\{f_k(\cdot), g_k(\cdot)\}_k$ make it difficult to form convenient format to enable mean-field variational inference. Thus, we use the technique of [2] to overcome these issues. In general, [2] would use latent marked Poisson processes and Polya-Gamma random variables to augment the model.

$$
P\left(\{t_m, z_m,\}_m, \left\{\omega_{m,k}^{(f)}\right\}_{k,m}, \left\{\omega_{mm',k}^{(g)}\right\}_{k,m,m'}, \left\{\Pi_{ij,k}^{(f)}\right\}_{k,i\neq j}, \left\{\Pi_{m,k}^{(g)}\right\}_{k,m} \Big| -\right)
$$

$$
= \prod_k \prod_m \left[ (\pi_{i_m k}\pi_{j_m k} e^{h(\omega_{m,k}^{(f)}, f_k(t_m))})^{\mathbf{1}_{z=(0,k)}(z_m)} P_{\text{PG}}(\omega_{m,k}^{(f)}|1,0) \right]
$$

$$
\cdot \prod_k \prod_m \prod_{m':m'\in F_m} \left[ (v_{i_m k}v_{j_m k} e^{h(\omega_{mm',k}^{(g)}, g_k(t_m - t_{m'}))})^{\mathbf{1}_{z=(m',k)}(z_m)} P_{\text{PG}}(\omega_{mm',k}^{(g)}|1,0) \right]
$$

$$
\cdot \prod_k \prod_{i\neq j} \left[ \prod_{(x,\omega)_n \in \Pi_{k,ij}^{(f)}} \left( e^{h(\omega_{k,ij,n}^{(f)}, -f_k(x_{k,ij,n}^{(f)}))} \pi_{ik}\pi_{jk} P_{\text{PG}}(\omega_{k,ij,n}^{(f)}|1,0) \right) e^{-\pi_{ik}\pi_{jk}T} \right]
$$

$$
\cdot \prod_k \prod_m \left[ \prod_{(x,\omega)_m \in \Pi_{m,k}^{(g)}} \left( e^{h(\omega_{m,k}^{(g)}, -g_k(x_{m,k}^{(g)}))} v_{i_m k}v_{j_m k} P_{\text{PG}}(\omega_{m,k}^{(g)}|1,0) \right) e^{-v_{i_m k}v_{j_m k}(T-t_m)} \right] \tag{32}
$$

where $h(\omega, x) = x/2 - \omega x^2/2 - \log 2$, $\omega_{m,k}^{(f)}, \omega_{mm',k}^{(g)} \sim P_{\text{PG}}(\omega|1,0)$, $\Pi_{k,ij}^{(f)}$ is the marked Poisson process with intensity $\pi_{i_m k}\pi_{j_m k}P_{\text{PG}}(\omega|1,0)dtd\omega$ in the support of $(0,T] \times \mathbb{R}^+$ and $\Pi_{m,k}^{(g)}$ is the marked Poisson process with intensity $v_{i_m k}v_{j_m k}P_{\text{PG}}(\omega|1,0)dtd\omega$ in the support of $(0, T - t_m] \times \mathbb{R}^+$.

$$\log P\left(\{t_m, z_m, \}_m, \left\{\omega_{m,k}^{(f)}\right\}_{k,m}, \left\{\omega_{mm',k}^{(g)}\right\}_{k,m,m'}, \left\{\Pi_{ij,k}^{(f)}\right\}_{k,i\neq j}, \left\{\Pi_{m,k}^{(g)}\right\}_{k,m} |-\right)$$

$$= \sum_k \sum_m \left[ \mathbf{1}_{z=(0,k)}(z_m) \left( \log(\pi_{i_m k}\pi_{j_m k}) + h(\omega_{m,k}^{(f)}, f_k(t_m)) \right) + \log P_{\mathrm{PG}}(\omega_{m,k}^{(f)}|1,0) \right]$$

$$+ \sum_k \sum_m \sum_{m':m'\in F_m} \left[ \mathbf{1}_{z=(m',k)}(z_m) \left( \log(v_{i_m k}v_{j_m k}) + h(\omega_{mm',k}^{(g)}, g_k(t_m - t_{m'})) \right) \right.$$

$$+ \log P_{\mathrm{PG}}(\omega_{mm',k}^{(g)}|1,0) \Big]$$

$$+ \sum_k \sum_{i\neq j} \left[ \left( \sum_{(x,\omega)_n \in \Pi_{k,ij}^{(f)}} h(\omega_{k,ij,n}^{(f)}, -f_k(x_{k,ij,n}^{(f)})) + \log(\pi_{ik}\pi_{jk}\cdot P_{\mathrm{PG}}(\omega_{k,ij,n}^{(f)}|1,0)) \right) - \pi_{ik}\pi_{jk}T \right]$$

$$+ \sum_k \sum_m \left[ \left( \sum_{(x,\omega)_n \in \Pi_{m,k}^{(g)}} h(\omega_{m,k}^{(g)}, -g_k(x_{m,k}^{(g)})) + \log(v_{i_m k}v_{j_m k}\cdot P_{\mathrm{PG}}(\omega_{m,k}^{(g)}|1,0)) \right) - v_{i_m k}v_{j_m k}(T-t_m) \right]$$

$$= \sum_k \sum_m \left[ \mathbf{1}_{z=(0,k)}(z_m) \left( \log(\pi_{i_m k}\pi_{j_m k}) + \frac{f_k(t_m)}{2} - \frac{(f_k(t_m))^2}{2}\omega_{m,k}^{(f)} - \log(2) \right) \right.$$

$$+ \log P_{\mathrm{PG}}(\omega_{m,k}^{(f)}|1,0) \Big]$$

$$+ \sum_k \sum_m \sum_{m':m'\in F_m} \left[ \mathbf{1}_{z=(m',k)}(z_m) \left( \log(v_{i_m k}v_{j_m k}) + \frac{g_k(t_m - t_{m'})}{2} \right. \right.$$

$$\left. - \frac{(g_k(t_m - t_{m'}))^2}{2}\omega_{mm',k}^{(g)} - \log(2) \right) + \log P_{\mathrm{PG}}(\omega_{mm',k}^{(g)}|1,0) \Big]$$

$$+ \sum_k \sum_{i\neq j} \left[ \sum_{(x,\omega)_n \in \Pi_{k,ij}^{(f)}} \left( -\frac{f_k(t_{k,ij,n}^{(f)})}{2} - \frac{(f_k(t_{k,ij,n}^{(f)}))^2}{2}\omega_{k,ij,n}^{(f)} - \log(2) + \log(\pi_{ik}\pi_{jk}\cdot P_{\mathrm{PG}}(\omega_{k,ij,n}^{(f)}|1,0)) \right) - \pi_{ik}\pi_{jk}T \right]$$

$$+ \sum_k \sum_m \left[ \sum_{(x,\omega)_n \in \Pi_{m,k}^{(g)}} \left( (-\frac{g_k(x_{k,m,n}^{(g)})}{2} - \frac{(g_k(x_{k,m,n}^{(g)}))^2}{2}\omega_{k,m,n}^{(g)} - \log(2)) \right. \right.$$

$$\left. + \log(v_{i_m k}v_{j_m k}\cdot P_{\mathrm{PG}}(\omega_{k,m,n}^{(g)}|1,0)) \right) - v_{i_m k}v_{j_m k}(T-t_m) \Big] \tag{33}$$

$$\mathbb{E}_q\left[\log P\left(\{t_m, z_m, \}_m, \left\{\omega_{m,k}^{(f)}\right\}_{k,m}, \left\{\omega_{mm',k}^{(g)}\right\}_{k,m,m'}, \left\{\Pi_{ij,k}^{(f)}\right\}_{k,i\neq j}, \left\{\Pi_{m,k}^{(g)}\right\}_{k,m} |-\right)\right]$$

$$= \sum_k \sum_m \left[ \phi_{m,0,k}\mathbb{E}_q\left( \log(\pi_{i_m k}\pi_{j_m k}) + \frac{f_k(t_m)}{2} - \frac{(f_k(t_m))^2}{2}\omega_{m,k}^{(f)} - \log(2) \right) + \mathbb{E}_q[\log P_{\mathrm{PG}}(\omega_{m,k}^{(f)}|1,0)] \right]$$

$$+ \sum_k \sum_m \sum_{m':m'\in F_m} \left[ \phi_{m,m',k}\mathbb{E}_q\left( \log(v_{i_m k}v_{j_m k}) + \frac{g_k(t_m - t_{m'})}{2} \right. \right.$$

$$\left. - \frac{(g_k(t_m - t_{m'}))^2}{2}\omega_{mm',k}^{(g)} - \log(2) \right) + \mathbb{E}_q[\log P_{\mathrm{PG}}(\omega_{mm',k}^{(g)}|1,0)] \Big]$$

$$+ \sum_{i\neq j} \sum_k \left[ \int_0^\infty \int_0^T \left( -\frac{f_k(t)}{2} - \frac{(f_k(t))^2}{2}\omega - \log(2) + \log(\pi_{ik}\pi_{jk}P_{\mathrm{PG}}(\omega|1,0)) \right) \Lambda_{k,ij}^{(f)}(t,\omega)dtd\omega - \mathbb{E}_q[\pi_{ik}\pi_{jk}]T \right]$$

$$+ \sum_k \sum_m \left[ \int_0^\infty \int_0^{T-t_m} \left( -\frac{g_k(t)}{2} - \frac{(g_k(t))^2}{2}\omega - \log(2) + \log(v_{i_m k}v_{j_m k}P_{\mathrm{PG}}(\omega|1,0)) \right) \Lambda_{m,k}^{(g)}(t,\omega)dtd\omega \right.$$

$$- \mathbb{E}_q[v_{i_m k}v_{j_m k}](T-t_m) \Big] \tag{34}$$

We make the following adjustments to make the method of [2] fit in our model. For $f_k(t)$, we incorporate auxiliary Polya-Gamma variables ($\omega_{m,k}^{(f)} \sim P_{\mathrm{PG}}(\omega|1,0)$) [7] for each edge $m$ (under the

case of $b_m = 0, z_m = k$) and $N^2 - N$ marked Poisson processes for all the potential nodal pairs (each with the intensity function $\pi_{i_m k}\pi_{j_m k}P_{\text{PG}}(\omega|1,0)dtd\omega$). For $g_k(t)$, we incorporates auxiliary Polya-Gamma variables ($\omega_{mm',k}^{(g)} \sim P_{\text{PG}}(\omega|1,0)$) for $m$ (under the case of $b_m = m', z_m = k$) and $K$ marked Poisson processes for the potential edges initiated or triggered by the $k$th label (with the intensity function $v_{i_m k}v_{j_m k}P_{\text{PG}}(\omega|1,0)dtd\omega$).

By using standard method of the mean-field variational inference method [1], we can obtain each random variable's optimal variational distribution through the following equations:

$$\ln q_{x_i}(x_i) = \mathbb{E}_{q_{\setminus x_i}}\left[\ln L(\mathcal{D}, \{x_i\}_i)\right] + \text{const} \tag{35}$$

Eq. (129) will be used in deriving the optimal variational distribution for all the random variables.

**Optimal Polya-Gamma density** $q(\omega_{m,k}^{(f)}), q(\omega_{mm',k}^{(g)})$

$$\log q(\omega_{m,k}^{(f)}) = \mathbb{E}_q\left[-\mathbf{1}_{z=(0,k)}(z_m) \cdot f_k^2(t_m)\omega_{m,k}^{(f)}/2\right] + \log P_{PG}(\omega_{m,k}^{(f)}|1,0) + \text{const} \tag{36}$$

Thus, we get

$$q(\omega_{m,k}^{(f)}) \propto \exp\left[-\phi_{m,0,k}\mathbb{E}_q(f_k^2(t_m))\omega_{m,k}^{(f)}/2\right] \cdot P_{PG}(\omega_{m,k}^{(f)}|1,0) \tag{37}$$

which leads to

$$q(\omega_{m,k}^{(f)}) = P_{\text{PG}}\left(\omega_{m,k}^{(f)}|1, \sqrt{\phi_{m,0,k}\mathbb{E}_q(f_k^2(t_m))}\right) \tag{38}$$

Similarly, we get

$$q(\omega_{mm',k}^{(g)}) = P_{\text{PG}}\left(\omega_{mm',k}^{(g)}|1, \sqrt{\phi_{m,m',k}\mathbb{E}_q(g_k^2(t_m - t_{m'}))}\right) \tag{39}$$

We can let $c_{m,k}^{(f)} = \sqrt{\phi_{m,0,k}\mathbb{E}_q(f_k^2(t_m))}$ and $c_{mm',k}^{(g)} = \sqrt{\phi_{m,m',k}\mathbb{E}_q(g_k^2(t_m - t_{m'}))}$.

**Optimal Poisson process** $q(\Pi_{k,ij}^{(f)}), q(\Pi_{m,k}^{(g)})$ Using the mean-field update equation, we get the rate functions for the latent marked Poisson processes as:

$$\Lambda_{k,ij}^{(f)}(t,\omega) = \frac{\exp(\mathbb{E}_q\left[\log \pi_{ik}\pi_{jk}\right] - \frac{\mathbb{E}_q[f_k(t)]}{2})}{2\cosh(\frac{c_k^{(f)}(t)}{2})}P_{\text{PG}}(\omega|1, c_k^{(f)}(t)) \tag{40}$$

$$\Lambda_{m,k}^{(g)}(t,\omega) = \frac{\exp(\mathbb{E}_q\left[\log v_{i_m k}v_{j_m k}\right] - \frac{\mathbb{E}_q(g_k(t))}{2})}{2\cosh(\frac{c_k^{(g)}(t)}{2})}P_{\text{PG}}(\omega|1, c_k^{(g)}(t)) \tag{41}$$

where $c_k^{(f)}(t) = \sqrt{\mathbb{E}_q[f_k(t)^2]}$, $c_k^{(g)}(t) = \sqrt{\mathbb{E}_q[g_k(t)^2]}$. Again, we emphasize that the support of $\Lambda_{k,ij}^{(f)}(t,\omega)$ is $(0,T] \times \mathbb{R}^+$ and the support of $\Lambda_{m,k}^{(g)}(t,\omega)$ is $(0, T - t_m] \times \mathbb{R}^+$.

**Optimal latent label** $q(z_m)$

$$\log \phi_{m,0,k} = \mathbb{E}_q\left[\log(\pi_{i_m k}\pi_{j_m k}) + h(\omega_{m,k}^{(f)}, f_k(t_m)\right]$$

$$= \left[\mathbb{E}_q[\log \pi_{i_m k}] + \mathbb{E}_q[\log \pi_{j_m k}] + \frac{\mathbb{E}_q[f_k(t_m)]}{2} - \frac{\mathbb{E}_q[f_k(t_m)^2]}{2}\mathbb{E}_q[\omega_{m,k}^{(f)}] - \log 2\right]$$

$$\forall m' : m' \in F_m,$$

$$\log \phi_{m,m',k} = \mathbb{E}_q\left[\log(v_{i_m k}v_{j_m k}) + h(\omega_{mm',k}^{(g)}, g_k(t_m - t_{m'}))\right]$$

$$= \left[\mathbb{E}_q[\log v_{i_m k}v_{j_m k}] + \frac{\mathbb{E}_q[g_k(t_m - t_{m'})]}{2} - \frac{\mathbb{E}_q[g_k(t_m - t_{m'})^2]}{2}\mathbb{E}_q[\omega_{mm',k}^{(g)}] - \log 2\right]$$

**Optimal feature vector $\boldsymbol{\pi}_i, \boldsymbol{v}_k$** The optimal density of $\pi_{ik}$ is a Gamma distribution as:

$$q(\pi_{ik}) \propto [\pi_{ik}]^{a_{\pi,ik}^{(q)}-1} e^{-b_{\pi,ik}^{(q)} \pi_{ik}} \tag{42}$$

where $a_{\pi,ik}^{(q)} = \sum_{j:j\neq i} \int_0^T (\Lambda_{k,ij}(t) + \Lambda_{k,ji}(t))dt + \sum_{m:i_m=i\cup j_m=i} \phi_{m,0,k} + a_k, b_{\pi,ik}^{(q)} = b_k + 2(\sum_{j:j\neq i} \pi_{jk})T$.

The optimal density of $v_{ik}$ is a Gamma distribution as:

$$q(v_{ik}) \propto [v_{ik}]^{a_{v,ik}^{(q)}-1} e^{-b_{v,ik}^{(q)} v_{ik}} \tag{43}$$

where $a_{v,ik}^{(q)} = a_k + \sum_{m:i_m=i\cup j_m=i} \int_0^T \Lambda_{k,m}(t)dt + \sum_{m:i_m=i\cup j_m=i} \sum_{m'\in F_m} \phi_{m,m',k}, b_{v,ik}^{(q)} = b_k + \sum_{m:j_m=i} v_{i_m k}(T - t_m) + \sum_{m:i_m=i} v_{j_m k}(T - t_m)$.

**Optimal Gaussian processes $f_k(t)$**

$$\log q(f_k) \propto \exp(U(f_k)) \tag{44}$$

$U(f_k)$ is defined as:

$$
\begin{aligned}
&U(f_k)\\
&=\mathbb{E}_q\left[\sum_{i\neq j}\sum_{(x,\omega)_n\in\Pi_k} h(\omega_{k,ij,n}^{(f)}, -f_k(x_{k,ij,n}^{(f)}))\right] + \sum_m \mathbb{E}_q\left[\mathbf{1}_{z=(0,k)}(z_m)h(\omega_{m,k}^{(f)}, f_k(t_m))\right]\\
&=-\frac{1}{2}\int_{\mathcal{T}} A(t)f_k^2(t)dt + \int_{\mathcal{T}} B(x)f_k^2(t)dt
\end{aligned} \tag{45}
$$

where

$$A(t) = \sum_m \phi_{m,0,k}\mathbb{E}_q[\omega_{m,k}^{(f)}]\delta(t - t_m) + \sum_{i\neq j}\int_0^\infty \omega\Lambda_{k,ij}^{(f)}(t,\omega)d\omega \tag{46}$$

$$B(t) = \frac{1}{2}\sum_m \phi_{m,0,k}\delta(t - t_m) - \frac{1}{2}\sum_{i\neq j}\int_0^\infty \Lambda_{k,ij}^{(f)}(t,\omega)d\omega \tag{47}$$

in which we have the following equations for the above integrations:

$$\mathbb{E}_q[\omega_{m,k}^{(f)}] = \frac{1}{2c_1^{(n)}} \tanh\left(\frac{c_1^{(n)}}{2}\right) \tag{48}$$

$$\int_{\mathcal{T}} \Lambda_1(t,\omega)d\omega = \Lambda_1(t) \tag{49}$$

$$\int_{\mathcal{T}} \omega\Lambda_1(t,\omega)d\omega = \frac{1}{2c_1(t)} \tanh\left(\frac{c_1(t)}{2}\right)\Lambda_1(t) \tag{50}$$

By using sparse Gaussian process methods, we get the sparse posterior distribution for the function values at the inducing points as:

$$q^{(s)} = \mathcal{N}(\boldsymbol{\mu}^{(s)}, \Sigma^{(s)}) \tag{51}$$

where the covariance matrix and mean is expressed as:

$$\Sigma^{(s)} = \left[K_s^{-1}\int_{\mathcal{T}} A(t)k_s(t)k_s^\top dt K_s^{-1} + K_s^{-1}\right]^{-1} \tag{52}$$

$$\boldsymbol{\mu}^{(s)} = \Sigma^{(s)}\left(K_s^{-1}\int_{\mathcal{T}} B(t)k_s(t)dt\right) \tag{53}$$

Given the sparse solution, we can obtain the point estimation as

$$\boldsymbol{\mu}(t) = k_s^\top(t) K_s^{-1} \boldsymbol{\mu}^{(s)} \tag{54}$$

$$(s(t))^2 = k(t,t) - k_s^\top(t) K_s^{-1} (\boldsymbol{I} - \sum K_s^{-1}) k_s(t) \tag{55}$$

**Optimal Gaussian processes** $g_k(t)$ For the triggering Gaussian processs $g_k(t)$, we can re-formulate the likelihood related to $g_k(t)$ as:

$$P(\mathcal{D}|g, v_k, z_m, b_m)$$
$$= \prod_m \prod_{(x,\omega)_m \in \Pi_{m,k}^{(g)}} e^{h(\omega_m, -g_k(x_m))} \cdot \prod_m \prod_{m':m' \in F_m} \left[ e^{h(\omega_{mm',k}^{(g)}, g_k(t_m - t_{m'}))} \right]^{\mathbf{1}_{z=(m',k)}(z_m)} \tag{56}$$

where $\Pi_{m,k}^{(g)}$ denotes a marked points process defined in $(0, T - t_m] \times \mathbb{R}^+$, with intensity function $v_k P_{\mathrm{PG}}(\omega|1,0)dtd\omega$.

$$\log q(g_k) \propto \exp(U(g_k)) \tag{57}$$

$U(g_k)$ is defined as:

$$U(g_k)$$
$$= \sum_m \left\{ \mathbb{E}_q \left[ \sum_{(x,\omega)_n \in \Pi_{m,k}^{(g)}} h(\omega_{m,k}^{(g)}, -g_k(x_{m,k}^{(g)})) \right] \right.$$
$$\left. + \phi_{m,m',k} \mathbb{E}_q \left[ \sum_{m':m' \in F_m} h(\omega_{mm',k}^{(g)}, g_k(t_m - t_{m'})) \right] \right\}$$
$$= \left\{ -\frac{1}{2} \int_{\mathcal{T}} A(t) g_k^2(t) dt + \int_{\mathcal{T}} B(t) g_k(t) dt \right\} \tag{58}$$

where

$$A(t) = \sum_m \int_0^\infty \omega \Lambda_{m,k}^{(g)}(t,\omega) d\omega + \sum_m \sum_{m':m' \in F_m} \left[ \phi_{m,m',k} \mathbb{E}_q \left[ \omega_{mm',k}^{(g)} \right] \delta(t - (t_m - t_{m'})) \right] \tag{59}$$

$$B(t) = -\frac{1}{2} \sum_m \int_0^\infty \Lambda_{m,k}^{(g)}(t,\omega) d\omega + \sum_m \sum_{m':m' \in F_m} \left[ \frac{1}{2} \phi_{m,m',k} \delta(t - (t_m - t_{m'})) \right] \tag{60}$$

in which we have the following equations for the above integration:

$$\mathbb{E}_q[\omega_{mm',k}^{(g)}] = \frac{1}{2c_{mm',k}^{(g)}} \tanh\left( \frac{c_{mm',k}^{(g)}}{2} \right) \tag{61}$$

$$\int_{\mathcal{T}} \Lambda_{m,k}^{(g)}(t,\omega) d\omega = \Lambda_{m,k}^{(g)}(t) \tag{62}$$

$$\int_{\mathcal{T}} \omega \Lambda_{m,k}^{(g)}(t,\omega) d\omega = \frac{1}{2c_{k,m}^{(g)}(t)} \tanh\left( \frac{c_{k,m}^{(g)}(t)}{2} \right) \Lambda_{m,k}^{(g)}(t) \tag{63}$$

By using sparse Gaussian process methods, we get the sparse posterior distribution for the function values at the inducing points as:

$$q^{(s)} = \mathcal{N}(\boldsymbol{\mu}^{(s)}, \Sigma^{(s)}) \tag{64}$$

where the covariance matrix and mean is expressed as:

$$\Sigma^{(s)} = \left[ K_s^{-1} \int_{\mathcal{T}} A(t) k_s(t) k_s^\top dt K_s^{-1} + K_s^{-1} \right]^{-1} \tag{65}$$

$$\boldsymbol{\mu}^{(s)} = \Sigma^{(s)} \left( K_s^{-1} \int_{\mathcal{T}} B(t) k_s(t) dt \right) \tag{66}$$

Given the sparse solution, we can obtain the point estimation as

$$\boldsymbol{\mu}(t) = k_s^\top(t) K_s^{-1} \boldsymbol{\mu}^{(s)} \tag{67}$$

$$(s(t))^2 = k(t,t) - k_s^\top(t) K_s^{-1} (\boldsymbol{I} - \sum K_s^{-1}) k_s(t) \tag{68}$$

**Update the hyper-parameters of the Kernel functions in the Gaussian process** Given $\Lambda_0(x, \omega) = \lambda P_{PG}(\omega|1, 0) dx d\omega$, the lower bound for the original paper is:

$$
\begin{aligned}
\mathcal{L}(\boldsymbol{q}) = \sum_k \Bigg\{ & \sum_{i \neq j} \Bigg[ \int_0^\infty \int_0^T \Bigg( \mathbb{E}_q[h(\omega, -f_k(x))] + \mathbb{E}_q[\log \pi_{i_m k} \pi_{j_m k}] - \mathbb{E}_q[\log \Lambda_{k,ij}^{(f)}(x)] - \mathbb{E}_q \log \cosh(\frac{c_1(x)}{2}) \\
& + \mathbb{E}_q[\frac{c_1(x)^2 \omega}{2}] + 1 \Bigg) \Lambda_{k,ij}^{(f)}(x, \omega)) dx d\omega - \mathbb{E}_q[\pi_{i_m k} \pi_{j_m k}] \cdot T \Bigg] \\
& + \sum_m \Bigg[ \int_0^\infty \int_0^{T - t_m} \Bigg( \mathbb{E}_q[h(\omega, -g_k(x))] + \mathbb{E}_q[\log v_{i_m k} v_{j_m k}] - \mathbb{E}_q[\log \Lambda_{m,k}^{(g)}(x)] - \mathbb{E}_q \log \cosh(\frac{c_1(x)}{2}) \\
& + \mathbb{E}_q[\frac{c_1(x)^2 \omega}{2}] + 1 \Bigg) \Lambda_{m,k}^{(g)}(x, \omega)) dx d\omega - \mathbb{E}_q[v_{i_m k} v_{j_m k}] \cdot (T - t_m) \Bigg] \\
& + \sum_m \Bigg[ \phi_{m,0,k} \Big( \mathbb{E}_q[\log \pi_{i_m k} \pi_{j_m k}] + \mathbb{E}_q[h(\omega_{m,k}^{(f)}, f_k(t_m))] \Big) - \mathbb{E}_q \log \cosh(\frac{c_{ij,n,k}^{(f)}}{2}) + \mathbb{E}_q[\frac{(c_{ij,n,k}^{(f)})^2 \omega_{m,k}^{(f)}}{2}] \Bigg] \\
& + \sum_m \sum_{m':m' \in F_m} \Bigg[ \phi_{m,m',k} \Big( \mathbb{E}_q[\log v_k] + \mathbb{E}_q[h(\omega_{mm',k}^{(g)}, g_k(t_m - t_{m'}))] \Big) - \mathbb{E}_q \log \cosh(\frac{c_{mm',k}^{(g)}}{2}) \\
& + \mathbb{E}_q[\frac{(c_{mm',k}^{(g)})^2 \omega_{mm',k}^{(g)}}{2}] \Bigg] \Bigg\} - \sum_k \sum_m \sum_{m':m' \in F_m} \phi_{m,m',k} \log \phi_{m,m',k} \\
& + \sum_k \Bigg\{ -\frac{1}{2} \mathrm{Tr} \Big( [K_k^{(f,s)}]^{-1} (\Sigma_k^{(f,s)} + \boldsymbol{\mu}_k^{(f,s)} [\boldsymbol{\mu}_k^{(f,s)}]^\top) \Big) - \frac{1}{2} \log \det K_k^{(f,s)} + \frac{1}{2} \log \det \Sigma_k^{(f,s)} \\
& - \frac{1}{2} \mathrm{Tr} \Big( [K_k^{(g,s)}]^{-1} (\Sigma_k^{(g,s)} + \boldsymbol{\mu}_k^{(g,s)} [\boldsymbol{\mu}_k^{(g,s)}]^\top) \Big) - \frac{1}{2} \log \det K_k^{(g,s)} + \frac{1}{2} \log \det \Sigma_k^{(g,s)} \Bigg\} + K N_s \\
& + \sum_{i,k} \Big\{ a^{(\pi)} \log b^{(\pi)} - \log \Gamma(a^{(\pi)}) + (a^{(\pi)} - 1) \mathbb{E}_q[\log \pi_{ik}] - b^{(\pi)} \mathbb{E}_q[\pi_{ik}] \\
& + a_{ik}^{(\pi,2)} - \log b_{ik}^{(\pi,2)} + \log \Gamma(a_{ik}^{(\pi,2)}) + (1 - a_{ik}^{(\pi,2)}) F(a_{ik}^{(\pi,2)}) \Big\} \\
& + \sum_{i,k} \Big\{ a^{(v)} \log b^{(v)} - \log \Gamma(a^{(v)}) + (a^{(v)} - 1) \mathbb{E}_q[\log v_{ik}] - b^{(v)} \mathbb{E}_q[v_{ik}] \\
& + a_{ik}^{(v,2)} - \log b_{ik}^{(v,2)} + \log \Gamma(a_{ik}^{(v,2)}) + (1 - a_{ik}^{(v,2)}) F(a_{ik}^{(v,2)}) \Big\}
\end{aligned}
\tag{69}
$$

$$\frac{\partial \mathcal{L}(\boldsymbol{q})}{\partial \theta_k^{(f)}} = \sum_{i \neq j} \int_0^\infty \int_0^T \frac{\partial \mathbb{E}_q[h(\omega, -f_k(x))]}{\partial \theta_k^{(f)}} \Lambda_{k,ij}^{(f)}(x, \omega)) dx d\omega + \sum_m \phi_{m,0,k} \frac{\partial \mathbb{E}_q[h(\omega_{m,k}^{(f)}, f_k(t_m))]}{\partial \theta_k^{(f)}}$$

$$- \frac{1}{2} \mathrm{Tr}\left([K_k^{(f,s)}]^{-1}(\Sigma_k^{(f,s)} + \boldsymbol{\mu}_k^{(f,s)}[\boldsymbol{\mu}_k^{(f,s)}]^\top)\right) - \frac{1}{2}\log \det K_k^{(f,s)}$$

$$= \sum_{i \neq j} \int_0^\infty \int_0^T \frac{\partial \mathbb{E}_q[h(\omega, -f_k(x))]}{\partial \theta_k^{(f)}} \Lambda_{k,ij}^{(f)}(x, \omega)) dx d\omega + \sum_m \phi_{m,0,k} \frac{\partial \mathbb{E}_q[h(\omega_{m,k}^{(f)}, f_k(t_m))]}{\partial \theta_k^{(f)}}$$

$$+ \frac{1}{2} \mathrm{Tr}\left([K_k^{(f,s)}]^{-1} \frac{\partial K_k^{(f,s)}}{\partial \theta_k^{(f)}}[K_k^{(f,s)}]^{-1}(\Sigma_k^{(f,s)} + \boldsymbol{\mu}_k^{(f,s)}[\boldsymbol{\mu}_k^{(f,s)}]^\top)\right) - \frac{1}{2}\mathrm{Tr}\left([K_k^{(f,s)}]^{-1} \frac{\partial K_k^{(f,s)}}{\partial \theta_k^{(f)}}\right)$$

$$(70)$$

$$\frac{\partial \mathcal{L}(\boldsymbol{q})}{\partial \theta_k^{(g)}} = \sum_m \int_0^\infty \int_0^{T-t_m} \frac{\partial \mathbb{E}_q[h(\omega, -g_k(x))]}{\partial \theta_k^{(g)}} \Lambda_{m,k}^{(g)}(x, \omega)) dx d\omega$$

$$+ \sum_m \sum_{m':m' \in F_m} \phi_{m,m',k} \frac{\partial \mathbb{E}_q[h(\omega_{mm',k}^{(g)}, g_k(t_m - t_{m'}))]}{\partial \theta_k^{(g)}}$$

$$+ \frac{1}{2} \mathrm{Tr}\left([K_k^{(g,s)}]^{-1} \frac{\partial K_k^{(g,s)}}{\partial \theta_k^{(g)}}[K_k^{(g,s)}]^{-1}(\Sigma_k^{(g,s)} + \boldsymbol{\mu}_k^{(g,s)}[\boldsymbol{\mu}_k^{(g,s)}]^\top)\right) - \frac{1}{2}\mathrm{Tr}\left([K_k^{(g,s)}]^{-1} \frac{\partial K_k^{(g,s)}}{\partial \theta_k^{(g)}}\right)$$

$$(71)$$

The gradient of the Evidence lower bound to the hyper-parameters (we let $\Phi_k^{(f)} := (\theta_k^{(f)}, \delta_k^{(f)})$ in the kernel functions of the Gaussian process) is:

$$\frac{\partial \mathbb{E}_q[h(\omega, f_k(x))]}{\partial \theta_k^{(f)}} = \frac{1}{2}\left(\frac{\partial \mathbb{E}_Q[f_k(x)]}{\partial \theta_k^{(f)}} - \frac{\partial \mathbb{E}_Q[f_k(x)^2]}{\partial \theta_k^{(f)}} \mathbb{E}_Q[\omega]\right) \tag{72}$$

$$\frac{\partial \mathbb{E}_q[f_k(x)]}{\partial \theta_k^{(f)}} = \frac{\partial \boldsymbol{\kappa}(x)}{\partial \theta_k^{(f)}} \boldsymbol{\mu}_2^{(f,s)} \tag{73}$$

$$\frac{\partial \mathbb{E}_q[f_k(x)^2]}{\partial \theta_k^{(f)}} = \frac{\partial \tilde{k}(x,x)}{\partial \theta_k^{(f)}} + \frac{\partial \boldsymbol{\kappa}(x)^\top}{\partial \theta_k^{(f)}}\left(\Sigma_2^{(f,s)} + \boldsymbol{\mu}_2^{(f,s)}\left(\boldsymbol{\mu}_2^{(f,s)}\right)^\top\right)\boldsymbol{\kappa}(x)$$

$$+ \boldsymbol{\kappa}(x)^\top \left(\Sigma_2^{(f,s)} + \boldsymbol{\mu}_2^{(f,s)}\left(\boldsymbol{\mu}_2^{(f,s)}\right)^\top\right) \frac{\partial \boldsymbol{\kappa}(x)}{\partial \theta_k^{(f)}} \tag{74}$$

where $\boldsymbol{\kappa}(x) = \boldsymbol{k}_s(x)^\top K_s^{-1}$ and $\tilde{k}(t,t) = k(x,x) - \boldsymbol{k}_s(x) K_s^{-1} \boldsymbol{k}_s(x)^\top$. The remaining two terms are:

$$\frac{\partial \tilde{k}(x,x)}{\partial \theta_k^{(f)}} = \frac{\partial k(x,x)}{\partial \theta_k^{(f)}} - \frac{\partial \boldsymbol{\kappa}(x)}{\partial \theta_k^{(f)}} \boldsymbol{k}_s(x) - \boldsymbol{\kappa}(x) \frac{\partial \boldsymbol{k}_s(t)}{\partial \theta_k^{(f)}} \tag{75}$$

$$\frac{\partial \boldsymbol{\kappa}(x)}{\partial \theta_k^{(f)}} = \frac{\partial \boldsymbol{k}_s(x)^\top}{\partial \theta_k^{(f)}} K_s^{-1} - \boldsymbol{k}_s(x) K_s^{-1} \frac{\partial K_s}{\partial \theta_k^{(f)}} K_s^{-1} \tag{76}$$

After each variational step, the hyper-parameters are updated by

$$\boldsymbol{\theta}_k^{(f)} = \boldsymbol{\theta}_k^{(f)} + \varepsilon \frac{\partial \mathcal{L}(q)}{\partial \boldsymbol{\theta}_k^{(f)}} \tag{77}$$

where $\epsilon$ is the stepsize.

# 5 Variational Inference for $K$ exogenous functions

**Likelihood of observation** The likelihood of the time observations $\{t_m\}_m$, latent label $\{z_m\}_m$ and the branching variable $\{b_m\}_m$ for ConTIM can be expressed as:

$$
P(\{t_m, b_m, z_m\}_m|-)
$$

$$
= \prod_k \left[ \prod_{i \neq j} e^{\int_0^T \pi_{ik} \pi_{jk} \sigma(f_k(t)) dt} \right] \cdot \prod_k \prod_m [\pi_{i_m k} \pi_{j_m k} \sigma(f_k(t_m))]^{\mathbf{1}_{z=k}(z_m)\mathbf{1}_{b=0}(b_m)}
$$

$$
\cdot \prod_k \prod_m e^{\int_0^{T-t_m} v_k \sigma(g_k(t)) dt \cdot \mathbf{1}_{z=k}(z_m))} \cdot \prod_k \prod_m \prod_{m':m' \in F_m} [v_k \sigma(g_k(t_m - t_{m'}))]^{\mathbf{1}_{b=m'}(b_m)\mathbf{1}_{z=k}(z_{m'})}
$$

where $F_m = \{m' : (i_m, j_m) = (j, i) \cap t_{m'} < t_m\}$. In Eq. (78), the sigmoid function $\sigma(\cdot)$ and the Gaussian process function $\{f_k(\cdot), g_k(\cdot)\}_k$ make it difficult to form convenient format to enable mean-field variational inference. Thus, we use the technique of [2] to overcome these issues. In general, [2] would use latent marked Poisson processes and Polya-Gamma random variables to augment the model.

$$
P\left(\{t_m, z_m, b_m\}_m, \left\{\omega_{m,k}^{(f)}\right\}_{k,m}, \left\{\omega_{mm',k}^{(g)}\right\}_{k,m,m'}, \left\{\Pi_{ij,k}^{(f)}\right\}_{k,i \neq j}, \left\{\Pi_{m,k}^{(g)}\right\}_{k,m}|-\right)
$$

$$
= \prod_k \prod_m \left[ (\pi_{i_m k} \pi_{j_m k} e^{h(\omega_{m,k}^{(f)}, f_k(t_m))})^{\mathbf{1}_{z=k}(z_m)\mathbf{1}_{b=0}(b_m)} P_{\text{PG}}(\omega_{m,k}^{(f)}|1,0) \right]
$$

$$
\cdot \prod_k \prod_m \prod_{m':m' \in F_m} \left[ (m_k e^{h(\omega_{mm',k}^{(g)}, g_k(t_m - t_{m'}))})^{\mathbf{1}_{b=m'}(b_m)\mathbf{1}_{z=k}(z_{m'})} P_{\text{PG}}(\omega_{mm',k}^{(g)}|1,0) \right]
$$

$$
\cdot \prod_k \prod_{i \neq j} \left[ \prod_{(x,\omega)_n \in \Pi_{k,ij}^{(f)}} \left( e^{h(\omega_{k,ij,n}^{(f)}, -f_k(x_{k,ij,n}^{(f)}))} \pi_{ik} \pi_{jk} P_{\text{PG}}(\omega_{k,ij,n}^{(f)}|1,0) \right) e^{-\pi_{ik}\pi_{jk}T} \right]
$$

$$
\cdot \prod_k \prod_m \left[ \prod_{(x,\omega)_m \in \Pi_{m,k}^{(g)}} \left( e^{h(\omega_{m,k}^{(g)}, -g_k(x_{m,k}^{(g)}))\mathbf{1}_{z=k}(z_m)} v_k P_{\text{PG}}(\omega_{m,k}^{(g)}|1,0) \right) e^{-v_k(T-t_m)} \right] \quad (78)
$$

where $h(\omega, x) = x/2 - \omega x^2/2 - \log 2$, $\omega_{m,k}^{(f)}, \omega_{mm',k}^{(g)} \sim P_{\text{PG}}(\omega|1,0)$, $\Pi_{k,ij}^{(f)}$ is the marked Poisson process with intensity $\pi_{i_m k} \pi_{j_m k} P_{\text{PG}}(\omega|1,0) dt d\omega$ in the support of $(0,T] \times \mathbb{R}^+$ and $\Pi_{m,k}^{(g)}$ is the marked Poisson process with intensity $v_k P_{\text{PG}}(\omega|1,0) dt d\omega$ in the support of $(0, T - t_m] \times \mathbb{R}^+$.

$$\log P\left(\{t_m, z_m, b_m\}_m, \left\{\omega_{m,k}^{(f)}\right\}_{k,m}, \left\{\omega_{mm',k}^{(g)}\right\}_{k,m,m'}, \left\{\Pi_{ij,k}^{(f)}\right\}_{k,i\neq j}, \left\{\Pi_{m,k}^{(g)}\right\}_{k,m} |-\right)$$

$$= \sum_k \sum_m \left[ \mathbf{1}_{z=k}(z_m)\mathbf{1}_{b=0}(b_m)\left(\log(\pi_{i_m k}\pi_{j_m k}) + h(\omega_{m,k}^{(f)}, f_k(t_m))\right) + \log P_{\text{PG}}(\omega_{m,k}^{(f)}|1,0)\right]$$

$$+ \sum_k \sum_m \sum_{m':m'\in F_m} \left[ \mathbf{1}_{b=m'}(b_m)\mathbf{1}_{z=k}(z_{m'})\left(\log(m_k) + h(\omega_{mm',k}^{(g)}, g_k(t_m - t_{m'}))\right)\right.$$

$$\left. + \log P_{\text{PG}}(\omega_{mm',k}^{(g)}|1,0)\right]$$

$$+ \sum_k \sum_{i\neq j}\left[\left(\sum_{(x,\omega)_n \in \Pi_{k,ij}^{(f)}} h(\omega_{k,ij,n}^{(f)}, -f_k(x_{k,ij,n}^{(f)})) + \log(\pi_{ik}\pi_{jk}\cdot P_{\text{PG}}(\omega_{k,ij,n}^{(f)}|1,0))\right) - \pi_{ik}\pi_{jk}T\right]$$

$$+ \sum_k \sum_m \left[\left(\sum_{(x,\omega)_n \in \Pi_{m,k}^{(g)}} \mathbf{1}_{z=k}(z_m)h(\omega_{m,k}^{(g)}, -g_k(x_{m,k}^{(g)})) + \log(v_k \cdot P_{\text{PG}}(\omega_{m,k}^{(g)}|1,0))\right) - v_k(T - t_m)\right]$$

$$= \sum_k \sum_m \left[ \mathbf{1}_{z=k}(z_m)\mathbf{1}_{b=0}(b_m)\left(\log(\pi_{i_m k}\pi_{j_m k}) + \frac{f_k(t_m)}{2} - \frac{(f_k(t_m))^2}{2}\omega_{m,k}^{(f)} - \log(2)\right)\right.$$

$$\left. + \log P_{\text{PG}}(\omega_{m,k}^{(f)}|1,0)\right]$$

$$+ \sum_k \sum_m \sum_{m':m'\in F_m} \left[ \mathbf{1}_{b=m'}(b_m)\mathbf{1}_{z=k}(z_{m'})\left(\log(m_k) + \frac{g_k(t_m - t_{m'})}{2}\right.\right.$$

$$\left.\left. - \frac{(g_k(t_m - t_{m'}))^2}{2}\omega_{mm',k}^{(g)} - \log(2)\right) + \log P_{\text{PG}}(\omega_{mm',k}^{(g)}|1,0)\right]$$

$$+ \sum_k \sum_{i\neq j}\left[\sum_{(x,\omega)_n \in \Pi_{k,ij}^{(f)}}\left(-\frac{f_k(t_{k,ij,n}^{(f)})}{2} - \frac{(f_k(t_{k,ij,n}^{(f)}))^2}{2}\omega_{k,ij,n}^{(f)} - \log(2) + \log(\pi_{ik}\pi_{jk}\cdot P_{\text{PG}}(\omega_{k,ij,n}^{(f)}|1,0))\right) - \pi_{ik}\pi_{jk}T\right]$$

$$+ \sum_k \sum_m \left[\sum_{(x,\omega)_n \in \Pi_{m,k}^{(g)}}\left(\mathbf{1}_{z=k}(z_m)(-\frac{g_k(x_{k,m,n}^{(g)})}{2} - \frac{(g_k(x_{k,m,n}^{(g)}))^2}{2}\omega_{k,m,n}^{(g)} - \log(2))\right.\right.$$

$$\left.\left. + \log(v_k \cdot P_{\text{PG}}(\omega_{k,m,n}^{(g)}|1,0))\right) - v_k(T - t_m)\right] \tag{79}$$

$$\mathbb{E}_q\left[\log P\left(\{t_m, z_m, b_m\}_m, \left\{\omega_{m,k}^{(f)}\right\}_{k,m}, \left\{\omega_{mm',k}^{(g)}\right\}_{k,m,m'}, \left\{\Pi_{ij,k}^{(f)}\right\}_{k,i\neq j}, \left\{\Pi_{m,k}^{(g)}\right\}_{k,m} |-\right)\right]$$

$$= \sum_k \sum_m \left[ \phi_{m,k}\psi_{m,0}\mathbb{E}_q\left(\log(\pi_{i_m k}\pi_{j_m k}) + \frac{f_k(t_m)}{2} - \frac{(f_k(t_m))^2}{2}\omega_{m,k}^{(f)} - \log(2)\right) + \mathbb{E}_q[\log P_{\text{PG}}(\omega_{m,k}^{(f)}|1,0)]\right]$$

$$+ \sum_k \sum_m \sum_{m':m'\in F_m} \left[ \phi_{m',k}\psi_{m,m'}\mathbb{E}_q\left(\log(m_k) + \frac{g_k(t_m - t_{m'})}{2}\right.\right.$$

$$\left.\left. - \frac{(g_k(t_m - t_{m'}))^2}{2}\omega_{mm',k}^{(g)} - \log(2)\right) + \mathbb{E}_q[\log P_{\text{PG}}(\omega_{mm',k}^{(g)}|1,0)]\right]$$

$$+ \sum_{i\neq j}\sum_k \left[\int_0^\infty \int_0^T \left(-\frac{f_k(t)}{2} - \frac{(f_k(t))^2}{2}\omega - \log(2) + \log(\pi_{ik}\pi_{jk}P_{\text{PG}}(\omega|1,0))\right)\Lambda_{k,ij}^{(f)}(t,\omega)dtd\omega - \mathbb{E}_q[\pi_{ik}\pi_{jk}]T\right]$$

$$+ \sum_k \sum_m \left[\int_0^\infty \int_0^{T-t_m}\left(\phi_{m,k}\left(-\frac{g_k(t)}{2} - \frac{(g_k(t))^2}{2}\omega - \log(2)\right) + \log(m_k P_{\text{PG}}(\omega|1,0))\right)\Lambda_{m,k}^{(g)}(t,\omega)dtd\omega\right.$$

$$- \mathbb{E}_q[v_k](T - t_m)] \tag{80}$$

We make the following adjustments to make the method of [2] fit in our model. For $f_k(t)$, we incorporate auxiliary Polya-Gamma variables ($\omega_{m,k}^{(f)} \sim P_{\text{PG}}(\omega|1,0)$) [7] for each edge $m$ (under the

case of $b_m = 0, z_m = k$) and $N^2 - N$ marked Poisson processes for all the potential nodal pairs (each with the intensity function $\pi_{i_m k}\pi_{j_m k}P_{PG}(\omega|1, 0)dtd\omega$). For $g_k(t)$, we incorporates auxiliary Polya-Gamma variables ($\omega_{mm',k}^{(g)} \sim P_{PG}(\omega|1, 0)$) for $m$ (under the case of $b_m = m', z_m = k$) and $K$ marked Poisson processes for the potential edges initiated or triggered by the $k$th label (with the intensity function $v_k P_{PG}(\omega|1, 0)dtd\omega$).

By using standard method of the mean-field variational inference method [1], we can obtain each random variable's optimal variational distribution through the following equations:

$$\ln q_{x_i}(x_i) = \mathbb{E}_{q_{\setminus x_i}}\left[\ln L(\mathcal{D}, \{x_i\}_i)\right] + \text{const} \tag{81}$$

Eq. (129) will be used in deriving the optimal variational distribution for all the random variables.

**Optimal Polya-Gamma density** $q(\omega_{m,k}^{(f)}), q(\omega_{mm',k}^{(g)})$

$$\log q(\omega_{m,k}^{(f)}) = \mathbb{E}_q\left[-\mathbf{1}_{z=k}(z_m)\mathbf{1}_{b=0}(b_m) \cdot f_k^2(t_m)\omega_{m,k}^{(f)}/2\right] + \log P_{PG}(\omega_{m,k}^{(f)}|1, 0) + \text{const} \tag{82}$$

Thus, we get

$$q(\omega_{m,k}^{(f)}) \propto \exp\left[-\phi_{m,k}\psi_{m,0}\mathbb{E}_q(f_k^2(t_m))\omega_{m,k}^{(f)}/2\right] \cdot P_{PG}(\omega_{m,k}^{(f)}|1, 0) \tag{83}$$

which leads to

$$q(\omega_{m,k}^{(f)}) = P_{PG}\left(\omega_{m,k}^{(f)}|1, \sqrt{\phi_{m,k}\psi_{m,0}\mathbb{E}_q(f_k^2(t_m))}\right) \tag{84}$$

Similarly, we get

$$q(\omega_{mm',k}^{(g)}) = P_{PG}\left(\omega_{mm',k}^{(g)}|1, \sqrt{\phi_{m',k}\psi_{m,m'}\mathbb{E}_q(g_k^2(t_m - t_{m'}))}\right) \tag{85}$$

We can let $c_{m,k}^{(f)} = \sqrt{\phi_{m,k}\psi_{m,0}\mathbb{E}_q(f_k^2(t_m))}$ and $c_{mm',k}^{(g)} = \sqrt{\phi_{m',k}\psi_{m,m'}\mathbb{E}_q(g_k^2(t_m - t_{m'}))}$.

**Optimal Poisson process** $q(\Pi_{k,ij}^{(f)}), q(\Pi_{m,k}^{(g)})$ Using the mean-field update equation, we get the rate functions for the latent marked Poisson processes as:

$$\Lambda_{k,ij}^{(f)}(t, \omega) = \frac{\exp(\mathbb{E}_q\left[\log \pi_{ik}\pi_{jk}\right] - \frac{\mathbb{E}_q[f_k(t)]}{2})}{2\cosh(\frac{c_k^{(f)}(t)}{2})}P_{PG}(\omega|1, c_k^{(f)}(t)) \tag{86}$$

$$\Lambda_{m,k}^{(g)}(t, \omega) = \frac{\exp(\mathbb{E}_q\left[\log v_k\right] - \frac{\phi_{m,k}\mathbb{E}_q(g_k(t))}{2})}{2^{\phi_{m,k}}\cosh(\frac{c_{k,m}^{(g)}(t)}{2})}P_{PG}(\omega|1, c_{k,m}^{(g)}(t)) \tag{87}$$

where $c_k^{(f)}(t) = \sqrt{\mathbb{E}_q[f_k(t)^2]}, c_{k,m}^{(g)}(t) = \sqrt{\phi_{m,k}\mathbb{E}_q[g_k(t)^2]}$. Again, we emphasize that the support of $\Lambda_{k,ij}^{(f)}(t, \omega)$ is $(0, T] \times \mathbb{R}^+$ and the support of $\Lambda_{m,k}^{(g)}(t, \omega)$ is $(0, T - t_m] \times \mathbb{R}^+$.

**Optimal branching variable** $q(b_m)$

$$\log \psi_{m,0} = \sum_k \phi_{m,k}\mathbb{E}_q\left[\log(\pi_{i_m k}\pi_{j_m k}) + h(\omega_{m,k}^{(f)}, f_k(t_m))\right]$$

$$= \sum_k \phi_{m,k}\left[\mathbb{E}_q[\log \pi_{i_m k}] + \mathbb{E}_q[\log \pi_{j_m k}] + \frac{\mathbb{E}_q[f_k(t_m)]}{2} - \frac{\mathbb{E}_q[f_k(t_m)^2]}{2}\mathbb{E}_q[\omega_{m,k}^{(f)}] - \log 2\right]$$

$$\forall m' : m' \in F_m,$$

$$\log \psi_{m,m'} = \sum_k \phi_{m',k}\mathbb{E}_q\left[\log(v_k) + h(\omega_{ji,n'n,k}^{(g)}, g_k(t_m - t_{m'}))\right]$$

$$= \sum_k \phi_{m',k}\left[\mathbb{E}_q[\log v_k] + \frac{\mathbb{E}_q[g_k(t_m - t_{m'})]}{2} - \frac{\mathbb{E}_q[g_k(t_m - t_{m'})^2]}{2}\mathbb{E}_q[\omega_{mm',k}^{(g)}] - \log 2\right]$$

**Optimal latent label** $q(z_m)$

$$\log \phi_{m,k}$$

$$=\mathbb{E}_q\left[\mathbf{1}_{b=0}(b_m)\left(\log(\pi_{i_m k}\pi_{j_m k}) + h(\omega_{m,k}^{(f)}, f_k(t_m))\right)\right]$$

$$+\sum_{m''\in F_m'}\mathbb{E}_q\left\{\mathbf{1}_{b=m}(b_{m''})\left[\log(v_k) + h(\omega_{m''m,k}^{(g)}, g_k(t_{m''}-t_m))\right]\right\}$$

$$+\mathbb{E}_q\left[\sum_{(x,\omega)_n\in\Pi_{m,k}^{(g)}} h(\omega_m, -g_k(x_m))\right]$$

$$=\psi_{m,0}\left[\mathbb{E}_q[\log\pi_{i_m k}] + \mathbb{E}_q[\log\pi_{j_m k}] + \frac{\mathbb{E}[f_k(t_m)]}{2} - \frac{\mathbb{E}[f_k(t_m)^2]}{2}\mathbb{E}_q[\omega_{m,k}^{(f)}] - \log 2\right]$$

$$+\sum_{m''\in F_m'}\psi_{m'',m}\left(\mathbb{E}_q[\log v_k] + \frac{\mathbb{E}_q[g_k(t_{m''}-t_m)]}{2} - \frac{\mathbb{E}_q[g_k(t_{m''}-t_m)^2]}{2}\mathbb{E}_q[\omega_{m''m,k}^{(g)}] - \log 2\right)$$

$$-\int_0^{T-t_m}\left[\frac{\mathbb{E}_q[g_k(t)^2]}{2}\mathbb{E}[\omega_{k,m}^{(g)}(t)]\cdot\Lambda_{m,k}^{(g)}(t) + \left(\frac{\mathbb{E}_q[g_k(t)]}{2} + \log(2)\right)\cdot\Lambda_{m,k}^{(g)}(t)\right]dt \qquad (88)$$

where $F_m' = \{m'' : (i_{m''}, j_{m''}) = (j,i) \cap t_{m''} > t_m\}$.

**Optimal bound variable** $v_k$ The optimal density of $v_k$ is a Gamma distribution as:

$$q(v_k) \propto [v_k]^{a_m^{(k)}-1}e^{-b_m^{(k)}v_k} \qquad (89)$$

where $a_m^{(k)} = a_m + \sum_m\sum_{m':m'\in F_m}\phi_{m',k}\psi_{m,m'} + \sum_m\int_0^{T-t_m}\Lambda_{m,k}^{(g)}(t)dt$, $b_m^{(k)} = b_m + \sum_m(T-t_m)$.

**Optimal feature vector** $\boldsymbol{\pi}_i$ The optimal density of $\pi_{ik}$ is a Gamma distribution as:

$$q(\pi_{ik}) \propto [\pi_{ik}]^{a_{\pi,ik}^{(q)}-1}e^{-b_{\pi,ik}^{(q)}\pi_{ik}} \qquad (90)$$

where $a_{\pi,ik}^{(q)} = \sum_{j:j\neq i}\int_0^T(\Lambda_{k,ij}(t) + \Lambda_{k,ji}(t))dt + \sum_{j':E_{ij'}^{(n)}}\phi_{ij',k}^{(n)}\psi_{ij',0}^{(n)} + a_k + \sum_{j':E_{j'i}^{(n)}}\Phi_{j'i,k}^{(n)}\psi_{j'i,0}^{(n)}$, $b_{\pi,ik}^{(q)} = b_k + 2(\sum_{j:j\neq i}\pi_{jk})T$.

**Optimal Gaussian processes** $f_k(t)$

$$\log q(f_k) \propto \exp(U(f_k)) \qquad (91)$$

$U(f_k)$ is defined as:

$$U(f_k)$$

$$=\mathbb{E}_q\left[\sum_{i\neq j}\sum_{(x,\omega)_n\in\Pi_k} h(\omega_{k,ij,n}^{(f)}, -f_k(x_{k,ij,n}^{(f)}))\right] + \sum_m\mathbb{E}_q\left[\mathbf{1}_{z=k}(z_m)\mathbf{1}_{b=0}(b_m)h(\omega_{m,k}^{(f)}, f_k(t_m))\right]$$

$$=-\frac{1}{2}\int_\mathcal{T} A(t)f_k^2(t)dt + \int_\mathcal{T} B(x)f_k^2(t)dt \qquad (92)$$

where

$$A(t) = \sum_m\phi_{m,k}\psi_{m,0}\mathbb{E}_q[\omega_{m,k}^{(f)}]\delta(t-t_m) + \sum_{i\neq j}\int_0^\infty\omega\Lambda_{k,ij}^{(f)}(t,\omega)d\omega \qquad (93)$$

$$B(t) = \frac{1}{2}\sum_m\phi_{m,k}\psi_{m,0}\delta(t-t_m) - \frac{1}{2}\sum_{i\neq j}\int_0^\infty\Lambda_{k,ij}^{(f)}(t,\omega)d\omega \qquad (94)$$

in which we have the following equations for the above integrations:

$$\mathbb{E}_q[\omega_{m,k}^{(f)}] = \frac{1}{2c_1^{(n)}} \tanh\left(\frac{c_1^{(n)}}{2}\right) \tag{95}$$

$$\int_{\mathcal{T}} \Lambda_1(t,\omega)d\omega = \Lambda_1(t) \tag{96}$$

$$\int_{\mathcal{T}} \omega \Lambda_1(t,\omega)d\omega = \frac{1}{2c_1(t)} \tanh\left(\frac{c_1(t)}{2}\right) \Lambda_1(t) \tag{97}$$

By using sparse Gaussian process methods, we get the sparse posterior distribution for the function values at the inducing points as:

$$q^{(s)} = \mathcal{N}(\boldsymbol{\mu}^{(s)}, \Sigma^{(s)}) \tag{98}$$

where the covariance matrix and mean is expressed as:

$$\Sigma^{(s)} = \left[ K_s^{-1} \int_{\mathcal{T}} A(t)k_s(t)k_s^\top dt K_s^{-1} + K_s^{-1} \right]^{-1} \tag{99}$$

$$\boldsymbol{\mu}^{(s)} = \Sigma^{(s)} \left( K_s^{-1} \int_{\mathcal{T}} B(t)k_s(t)dt \right) \tag{100}$$

Given the sparse solution, we can obtain the point estimation as

$$\boldsymbol{\mu}(t) = k_s^\top(t) K_s^{-1} \boldsymbol{\mu}^{(s)} \tag{101}$$

$$(s(t))^2 = k(t,t) - k_s^\top(t) K_s^{-1}(\boldsymbol{I} - \sum K_s^{-1}) k_s(t) \tag{102}$$

**Optimal Gaussian processes** $g_k(t)$ For the triggering Gaussian processs $g_k(t)$, we can re-formulate the likelihood related to $g_k(t)$ as:

$$
\begin{aligned}
&P(\mathcal{D}|g, v_k, z_m, b_m) \\
&= \prod_m \prod_{(x,\omega)_m \in \Pi_{m,k}^{(g)}} e^{h(\omega_m, -g_k(x_m))\mathbf{1}_{z=k}(z_m)} \\
&\cdot \prod_m \prod_{m':m' \in F_m} \left[ v_k e^{h(\omega_{mm',k}^{(g)}, g_k(t_m - t_{m'}))} \right]^{\mathbf{1}_{b=m'}(b_m)\mathbf{1}_{z=k}(z_{m'})}
\end{aligned} \tag{103}
$$

where $\Pi_{m,k}^{(g)}$ denotes a marked points process defined in $(0, T - t_m] \times \mathbb{R}^+$, with intensity function $v_k P_{\text{PG}}(\omega|1, 0)dtd\omega$.

$$\log q(g_k) \propto \exp(U(g_k)) \tag{104}$$

$U(g_k)$ is defined as:

$$
\begin{aligned}
&U(g_k) \\
&= \sum_m \left\{ \phi_{m,k} \mathbb{E}_q \left[ \sum_{(x,\omega)_n \in \Pi_{m,k}^{(g)}} h(\omega_{m,k}^{(g)}, -g_k(x_{m,k}^{(g)})) \right] \right. \\
&\quad \left. + \psi_{m,m'} \phi_{m',k} \mathbb{E}_q \left[ \sum_{m':m' \in F_m} h(\omega_{mm',k}^{(g)}, g_k(t_m - t_{m'})) \right] \right\} \\
&= \left\{ -\frac{1}{2} \int_{\mathcal{T}} A(t)g_k^2(t)dt + \int_{\mathcal{T}} B(t)g_k(t)dt \right\}
\end{aligned} \tag{105}
$$

where

$$A(t)$$
$$= \sum_m \phi_{m,k} \cdot \int_0^\infty \omega \Lambda_{m,k}^{(g)}(t,\omega) d\omega$$
$$+ \sum_m \sum_{m':m' \in F_m} \left[ \psi_{m,m'} \phi_{m',k} \mathbb{E}_q \left[ \omega_{mm',k}^{(g)} \right] \delta(t - (t_m - t_{m'})) \right] \tag{106}$$

$$B(t)$$
$$= -\frac{1}{2} \sum_m \phi_{m,k} \cdot \int_0^\infty \Lambda_{m,k}^{(g)}(t,\omega) d\omega$$
$$+ \sum_m \sum_{m':m' \in F_m} \left[ \frac{1}{2} \cdot \psi_{m,m'} \phi_{m',k} \delta(t - (t_m - t_{m'})) \right] \tag{107}$$

in which we have the following equations for the above integration:

$$\mathbb{E}_q[\omega_{mm',k}^{(g)}] = \frac{1}{2c_{mm',k}^{(g)}} \tanh\left( \frac{c_{mm',k}^{(g)}}{2} \right) \tag{108}$$

$$\int_{\mathcal{T}} \Lambda_{m,k}^{(g)}(t,\omega) d\omega = \Lambda_{m,k}^{(g)}(t) \tag{109}$$

$$\int_{\mathcal{T}} \omega \Lambda_{m,k}^{(g)}(t,\omega) d\omega = \frac{1}{2c_{k,m}^{(g)}(t)} \tanh\left( \frac{c_{k,m}^{(g)}(t)}{2} \right) \Lambda_{m,k}^{(g)}(t) \tag{110}$$

By using sparse Gaussian process methods, we get the sparse posterior distribution for the function values at the inducing points as:

$$q^{(s)} = \mathcal{N}(\boldsymbol{\mu}^{(s)}, \Sigma^{(s)}) \tag{111}$$

where the covariance matrix and mean is expressed as:

$$\Sigma^{(s)} = \left[ K_s^{-1} \int_{\mathcal{T}} A(t) k_s(t) k_s^\top dt K_s^{-1} + K_s^{-1} \right]^{-1} \tag{112}$$

$$\boldsymbol{\mu}^{(s)} = \Sigma^{(s)} \left( K_s^{-1} \int_{\mathcal{T}} B(t) k_s(t) dt \right) \tag{113}$$

Given the sparse solution, we can obtain the point estimation as

$$\boldsymbol{\mu}(t) = k_s^\top(t) K_s^{-1} \boldsymbol{\mu}^{(s)} \tag{114}$$

$$(s(t))^2 = k(t,t) - k_s^\top(t) K_s^{-1} (\boldsymbol{I} - \sum K_s^{-1}) k_s(t) \tag{115}$$

**Update the hyper-parameters of the Kernel functions in the Gaussian process** Given $\Lambda_0(x,\omega) = \lambda P_{PG}(\omega|1,0) dx d\omega$, the lower bound for the original paper is:

$$\mathcal{L}(\boldsymbol{q}) = \sum_k \left\{ \sum_{i \neq j} \left[ \int_0^\infty \int_0^T \left( \mathbb{E}_q[h(\omega, -f_k(x))] + \mathbb{E}_q[\log \pi_{i_m k} \pi_{j_m k}] - \mathbb{E}_q[\log \Lambda_{k,ij}^{(f)}(x)] - \mathbb{E}_q \log \cosh(\frac{c_1(x)}{2}) \right. \right. \right.$$

$$\left. + \mathbb{E}_q[\frac{c_1(x)^2 \omega}{2}] + 1 \right) \Lambda_{k,ij}^{(f)}(x,\omega)) dx d\omega - \mathbb{E}_q[\pi_{i_m k} \pi_{j_m k}] \cdot T \right]$$

$$+ \sum_m \left[ \int_0^\infty \int_0^{T-t_m} \left( \phi_{m,k} \mathbb{E}_q[h(\omega, -g_k(x))] + \mathbb{E}_q[\log v_k] - \mathbb{E}_q[\log \Lambda_{m,k}^{(g)}(x)] - \mathbb{E}_q \log \cosh(\frac{c_1(x)}{2}) \right. \right.$$

$$\left. + \mathbb{E}_q[\frac{c_1(x)^2 \omega}{2}] + 1 \right) \Lambda_{m,k}^{(g)}(x,\omega)) dx d\omega - \mathbb{E}_q[v_k] \cdot (T - t_m) \right]$$

$$+ \sum_m \left[ \phi_{m,k} \psi_{m,0} \left( \mathbb{E}_q[\log \pi_{i_m k} \pi_{j_m k}] + \mathbb{E}_q[h(\omega_{m,k}^{(f)}, f_k(t_m))] \right) - \mathbb{E}_q \log \cosh(\frac{c_{ij,n,k}^{(f)}}{2}) + \mathbb{E}_q[\frac{(c_{ij,n,k}^{(f)})^2 \omega_{m,k}^{(f)}}{2}] \right]$$

$$+ \sum_m \sum_{m': m' \in F_m} \left[ \phi_{m',k} \psi_{m,m'} \left( \mathbb{E}_q[\log v_k] + \mathbb{E}_q[h(\omega_{mm',k}^{(g)}, g_k(t_m - t_{m'}))] \right) - \mathbb{E}_q \log \cosh(\frac{c_{mm',k}^{(g)}}{2}) \right.$$

$$\left. \left. + \mathbb{E}_q[\frac{(c_{mm',k}^{(g)})^2 \omega_{mm',k}^{(g)}}{2}] \right] \right\} - \sum_k \sum_m \phi_{m,k} \log \phi_{m,k} - \sum_m \sum_{m': m' \in F_m} \psi_{m,m'} \log \psi_{m,m'}$$

$$+ \sum_k \left\{ -\frac{1}{2} \text{Tr} \left( [K_k^{(f,s)}]^{-1} (\Sigma_k^{(f,s)} + \boldsymbol{\mu}_k^{(f,s)} [\boldsymbol{\mu}_k^{(f,s)}]^\top) \right) - \frac{1}{2} \log \det K_k^{(f,s)} + \frac{1}{2} \log \det \Sigma_k^{(f,s)} \right.$$

$$\left. -\frac{1}{2} \text{Tr} \left( [K_k^{(g,s)}]^{-1} (\Sigma_k^{(g,s)} + \boldsymbol{\mu}_k^{(g,s)} [\boldsymbol{\mu}_k^{(g,s)}]^\top) \right) - \frac{1}{2} \log \det K_k^{(g,s)} + \frac{1}{2} \log \det \Sigma_k^{(g,s)} \right\} + K N_s$$

$$+ \sum_{i,k} \left\{ a^{(\pi)} \log b^{(\pi)} - \log \Gamma(a^{(\pi)}) + (a^{(\pi)} - 1) \mathbb{E}_q[\log \pi_{ik}] - b^{(\pi)} \mathbb{E}_q[\pi_{ik}] \right.$$

$$\left. + a_{ik}^{(\pi,2)} - \log b_{ik}^{(\pi,2)} + \log \Gamma(a_{ik}^{(\pi,2)}) + (1 - a_{ik}^{(\pi,2)}) F(a_{ik}^{(\pi,2)}) \right\}$$

$$+ \sum_k \left\{ a^{(m)} \log b^{(m)} - \log \Gamma(a^{(m)}) + (a^{(m)} - 1) \mathbb{E}_q[\log v_k] - b^{(m)} \mathbb{E}_q[m_k] \right.$$

$$\left. + a_k^{(m,2)} - \log b_k^{(m,2)} + \log \Gamma(a_k^{(m,2)}) + (1 - a_k^{(m,2)}) F(a_k^{(m,2)}) \right\} \tag{116}$$

$$\frac{\partial \mathcal{L}(\boldsymbol{q})}{\partial \theta_k^{(f)}} = \sum_{i \neq j} \int_0^\infty \int_0^T \frac{\partial \mathbb{E}_q[h(\omega, -f_k(x))]}{\partial \theta_k^{(f)}} \Lambda_{k,ij}^{(f)}(x,\omega)) dx d\omega + \sum_m \phi_{m,k} \psi_{m,0} \frac{\partial \mathbb{E}_q[h(\omega_{m,k}^{(f)}, f_k(t_m))]}{\partial \theta_k^{(f)}}$$

$$- \frac{1}{2} \text{Tr} \left( [K_k^{(f,s)}]^{-1} (\Sigma_k^{(f,s)} + \boldsymbol{\mu}_k^{(f,s)} [\boldsymbol{\mu}_k^{(f,s)}]^\top) \right) - \frac{1}{2} \log \det K_k^{(f,s)}$$

$$= \sum_{i \neq j} \int_0^\infty \int_0^T \frac{\partial \mathbb{E}_q[h(\omega, -f_k(x))]}{\partial \theta_k^{(f)}} \Lambda_{k,ij}^{(f)}(x,\omega)) dx d\omega + \sum_m \phi_{m,k} \psi_{m,0} \frac{\partial \mathbb{E}_q[h(\omega_{m,k}^{(f)}, f_k(t_m))]}{\partial \theta_k^{(f)}}$$

$$+ \frac{1}{2} \text{Tr} \left( [K_k^{(f,s)}]^{-1} \frac{\partial K_k^{(f,s)}}{\partial \theta_k^{(f)}} [K_k^{(f,s)}]^{-1} (\Sigma_k^{(f,s)} + \boldsymbol{\mu}_k^{(f,s)} [\boldsymbol{\mu}_k^{(f,s)}]^\top) \right) - \frac{1}{2} \text{Tr} \left( [K_k^{(f,s)}]^{-1} \frac{\partial K_k^{(f,s)}}{\partial \theta_k^{(f)}} \right) \tag{117}$$

$$\frac{\partial \mathcal{L}(\boldsymbol{q})}{\partial \theta_k^{(g)}} = \sum_m \int_0^\infty \int_0^{T-t_m} \phi_{m,k} \frac{\partial \mathbb{E}_q[h(\omega, -g_k(x))]}{\partial \theta_k^{(g)}} \Lambda_{m,k}^{(g)}(x,\omega)) dx d\omega$$

$$+ \sum_m \sum_{m': m' \in F_m} \phi_{m',k} \psi_{m,m'} \frac{\partial \mathbb{E}_q[h(\omega_{mm',k}^{(g)}, g_k(t_m - t_{m'}))]}{\partial \theta_k^{(g)}}$$

$$+ \frac{1}{2} \text{Tr} \left( [K_k^{(g,s)}]^{-1} \frac{\partial K_k^{(g,s)}}{\partial \theta_k^{(g)}} [K_k^{(g,s)}]^{-1} (\Sigma_k^{(g,s)} + \boldsymbol{\mu}_k^{(g,s)} [\boldsymbol{\mu}_k^{(g,s)}]^\top) \right) - \frac{1}{2} \text{Tr} \left( [K_k^{(g,s)}]^{-1} \frac{\partial K_k^{(g,s)}}{\partial \theta_k^{(g)}} \right) \tag{118}$$

The gradient of the Evidence lower bound to the hyper-parameters (we let $\Phi_k^{(f)} := (\theta_k^{(f)}, \delta_k^{(f)})$ in the kernel functions of the Gaussian process) is:

$$\frac{\partial \mathbb{E}_q[h(\omega, f_k(x))]}{\partial \theta_k^{(f)}} = \frac{1}{2}\left(\frac{\partial \mathbb{E}_Q[f_k(x)]}{\partial \theta_k^{(f)}} - \frac{\partial \mathbb{E}_Q\left[f_k(x)^2\right]}{\partial \theta_k^{(f)}}\mathbb{E}_Q[\omega]\right) \tag{119}$$

$$\frac{\partial \mathbb{E}_q[f_k(x)]}{\partial \theta_k^{(f)}} = \frac{\partial \boldsymbol{\kappa}(x)}{\partial \theta_k^{(f)}}\boldsymbol{\mu}_2^{(f,s)} \tag{120}$$

$$\frac{\partial \mathbb{E}_q\left[f_k(x)^2\right]}{\partial \theta_k^{(f)}} = \frac{\partial \tilde{k}(x,x)}{\partial \theta_k^{(f)}} + \frac{\partial \boldsymbol{\kappa}(x)^\top}{\partial \theta_k^{(f)}}\left(\Sigma_2^{(f,s)} + \boldsymbol{\mu}_2^{(f,s)}\left(\boldsymbol{\mu}_2^{(f,s)}\right)^\top\right)\boldsymbol{\kappa}(x)$$

$$+ \boldsymbol{\kappa}(x)^\top\left(\Sigma_2^{(f,s)} + \boldsymbol{\mu}_2^{(f,s)}\left(\boldsymbol{\mu}_2^{(f,s)}\right)^\top\right)\frac{\partial \boldsymbol{\kappa}(x)}{\partial \theta_k^{(f)}} \tag{121}$$

where $\boldsymbol{\kappa}(x) = \boldsymbol{k}_s(x)^\top K_s^{-1}$ and $\tilde{k}(t,t) = k(x,x) - \boldsymbol{k}_s(x)K_s^{-1}\boldsymbol{k}_s(x)^\top$. The remaining two terms are:

$$\frac{\partial \tilde{k}(x,x)}{\partial \theta_k^{(f)}} = \frac{\partial k(x,x)}{\partial \theta_k^{(f)}} - \frac{\partial \boldsymbol{\kappa}(x)}{\partial \theta_k^{(f)}}\boldsymbol{k}_s(x) - \boldsymbol{\kappa}(x)\frac{\partial \boldsymbol{k}_s(t)}{\partial \theta_k^{(f)}} \tag{122}$$

$$\frac{\partial \boldsymbol{\kappa}(x)}{\partial \theta_k^{(f)}} = \frac{\partial \boldsymbol{k}_s(x)^\top}{\partial \theta_k^{(f)}}K_s^{-1} - \boldsymbol{k}_s(x)K_s^{-1}\frac{\partial K_s}{\partial \theta_k^{(f)}}K_s^{-1} \tag{123}$$

After each variational step, the hyper-parameters are updated by

$$\boldsymbol{\theta}_k^{(f)} = \boldsymbol{\theta}_k^{(f)} + \varepsilon\frac{\partial \mathcal{L}(q)}{\partial \boldsymbol{\theta}_k^{(f)}} \tag{124}$$

where $\epsilon$ is the stepsize.

## 6 Variational Inference for Constant exogenous function

**Likelihood of observation** The likelihood of the time observations $\{(i_m, j_m, t_m)\}_{m=1}^M$, latent label $\{z_m\}_{m=1}^M$ and the branching variable $\{b_m\}_{m=1}^M$ for ConTIM can be expressed as:

$$P(\{(i_m, j_m, t_m), b_m, z_m\}_m | -) = \left[\prod_k \prod_{i \neq j} e^{-\pi_{i_m k}\pi_{j_m k}T}\right] \cdot \prod_m \left[\pi_{i_m k}\pi_{j_m k}\right]^{\mathbf{1}_{z=k}(z_m)\mathbf{1}_{b=0}(b_m)}$$

$$\cdot \prod_k \prod_m \left[e^{-\int_0^{T-t_m} v_k\sigma(g_k(t))dt \cdot \mathbf{1}_{z=k}(z_m))} \prod_{m':m' \in F_m} [v_k\sigma(g_k(t_m - t_{m'}))]^{\mathbf{1}_{b=m'}(b_m)\mathbf{1}_{z=k}(z_{m'})}\right] \tag{125}$$

where $F_m = \{m' : (i_{m'}, j_{m'}) = (j, i) \cap t_{m'} < t_m\}$.

In Eq. (125), the sigmoid function $\sigma(\cdot)$ and the Gaussian process function $\{g_k(\cdot)\}_k$ make it difficult to form convenient format to enable mean-field variational inference. Thus, we use the technique of [2] to overcome these issues. In general, [2] would use latent marked Poisson processes and

Polya-Gamma random variables to augment the model.

$$
P\left(\{t_m, z_m, b_m\}_m, \left\{\omega^{(g)}_{mm',k}\right\}_{k,m,m'}, \left\{\Pi^{(g)}_{m,k}\right\}_{k,m} \Big| -\right)
$$

$$
= \prod_k \prod_{i \neq j} \left[e^{-\pi_{ik}\pi_{jk}T}\right] \prod_{m,k} \left[(\pi_{i_m k}\pi_{j_m k})^{\mathbf{1}_{z=k}(z_m)\mathbf{1}_{b=0}(b_m)}\right]
$$

$$
\cdot \prod_k \prod_m \prod_{m':m' \in F_m} \left[(m_k e^{h(\omega^{(g)}_{mm',k}, g_k(t_m - t_{m'}))})^{\mathbf{1}_{b=m'}(b_m)\mathbf{1}_{z=k}(z_{m'})} P_{\mathrm{PG}}(\omega^{(g)}_{mm',k}|1,0)\right]
$$

$$
\cdot \prod_k \prod_m \left[\prod_{(x,\omega)_n \in \Pi^{(g)}_{m,k}} \left(e^{h(\omega^{(g)}_{k,m,n}, -g_k(x^{(g)}_{k,m,n}))\mathbf{1}_{z=k}(z_m)} v_k P_{\mathrm{PG}}(\omega^{(g)}_{k,m,n}|1,0)\right) e^{-v_k(T - t_m)}\right] \tag{126}
$$

where $h(\omega, x) = x/2 - \omega x^2/2 - \log 2$, $\omega^{(g)}_{mm',k} \sim P_{\mathrm{PG}}(\omega|1,0)$, $\Pi^{(g)}_{m,k}$ is the marked Poisson process with intensity $v_k P_{\mathrm{PG}}(\omega|1,0)dtd\omega$ in the support of $(0, T - t_m) \times \mathbb{R}^+$.

$$
\log P\left(\{t_m, z_m, b_m\}_m, \left\{\omega^{(g)}_{mm',k}\right\}_{k,m,m'}, \left\{\Pi^{(g)}_{m,k}\right\}_{k,m} \Big| -\right)
$$

$$
= -\sum_k \sum_{i \neq j} [\pi_{i_m k}\pi_{j_m k}T] + \sum_{m,k} [\mathbf{1}_{z=k}(z_m)\mathbf{1}_{b=0}(b_m) \log(\pi_{i_m k}\pi_{j_m k})]
$$

$$
+ \sum_k \sum_m \sum_{m':m' \in F_m} \left[\mathbf{1}_{b=m'}(b_m)\mathbf{1}_{z=k}(z_{m'}) \left(\log(m_k) + h(\omega^{(g)}_{mm',k}, g_k(t_m - t_{m'}))\right)\right.
$$

$$
+ \log P_{\mathrm{PG}}(\omega^{(g)}_{mm',k}|1,0)\Big]
$$

$$
+ \sum_k \sum_m \left[\left(\sum_{(x,\omega)_n \in \Pi^{(g)}_{m,k}} \mathbf{1}_{z=k}(z_m)h(\omega^{(g)}_{n,k}, -g_k(x^{(g)}_{k,m,n})) + \log(v_k \cdot P_{\mathrm{PG}}(\omega^{(g)}_{n,k}|1,0))\right) - v_k(T - t_m)\right]
$$

$$
= -\sum_k \sum_{i \neq j} [\pi_{i_m k}\pi_{j_m k}T] + \sum_{m,k} [\mathbf{1}_{z=k}(z_m)\mathbf{1}_{b=0}(b_m) \log(\pi_{i_m k}\pi_{j_m k})]
$$

$$
+ \sum_k \sum_m \sum_{m':m' \in F_m} \left[\mathbf{1}_{b=m'}(b_m)\mathbf{1}_{z=k}(z_{m'}) \left(\log(m_k) + \frac{g_k(t_m - t_{m'})}{2}\right.\right.
$$

$$
\left.\left. - \frac{(g_k(t_m - t_{m'}))^2}{2}\omega^{(g)}_{mm',k} - \log(2)\right) + \log P_{\mathrm{PG}}(\omega^{(g)}_{mm',k}|1,0)\right]
$$

$$
+ \sum_k \sum_m \left[\sum_{(x,\omega)_n \in \Pi^{(g)}_{m,k}} \left(\mathbf{1}_{z=k}(z_m)(-\frac{g_k(x^{(g)}_{k,m,n})}{2} - \frac{(g_k(x^{(g)}_{k,m,n}))^2}{2}\omega^{(g)}_{k,m,n} - \log 2)\right.\right.
$$

$$
\left.\left. + \log(v_k \cdot P_{\mathrm{PG}}(\omega^{(g)}_{k,m,n}|1,0))\right) - v_k(T - t_m)\right] \tag{127}
$$

$$
\mathbb{E}_q\left[\log P\left(\{t_m, z_m, b_m\}_m, \left\{\omega^{(g)}_{mm',k}\right\}_{k,m,m'}, \left\{\Pi^{(g)}_{m,k}\right\}_{k,m} \Big| -\right)\right]
$$

$$
= -\sum_{k,i \neq j} \mathbb{E}_q[\pi_{i_m k}\pi_{j_m k}]T + \sum_{m,k} \{\phi_{m,k}\psi_{m,0}\mathbb{E}_q[\log(\pi_{i_m k}\pi_{j_m k})]\}
$$

$$
+ \sum_{k,m} \sum_{m':m' \in F_m} \left\{\phi_{m',k}\psi_{m,m'}\mathbb{E}_q\left[\log v_k + \frac{g_k(t_m - t_{m'})}{2} - \frac{(g_k(t_m - t_{m'}))^2}{2}\omega^{(g)}_{mm',k} - \log 2\right]\right.
$$

$$
\left. + \mathbb{E}_q[\log P_{\mathrm{PG}}(\omega^{(g)}_{mm',k}|1,0)]\right\}
$$

$$
+ \sum_{m,k} \left[\int_0^\infty \int_0^{T-t_m} \left(\phi_{m,k}\left(-\frac{g_k(x)}{2} - \frac{(g_k(x))^2}{2}\omega - \log 2\right) + \log(m_k P_{\mathrm{PG}}(\omega|1,0))\right) \Lambda^{(g)}_{m,k}(x,\omega)dxd\omega\right.
$$

$$
- \mathbb{E}_q[v_k](T - t_m) \tag{128}
$$

We make the following adjustments to make the method of [2] fit in our model. For $g_k(t)$, we incorporates auxiliary Polya-Gamma variables ($\omega_{mm',k}^{(g)} \sim P_{\text{PG}}(\omega|1,0)$) for $(i_m, j_m, t_m)$ (under the case of $b_m = m', z_{m'} = k$) and $K$ marked Poisson processes for the potential edges initiated or triggered by the $k$th label (with the intensity function $v_k P_{\text{PG}}(\omega|1,0)dtd\omega$).

By using standard method of the mean-field variational inference method [1], we can obtain each random variable's optimal variational distribution through the following equations:

$$\ln q_{x_i}(x_i) = \mathbb{E}_{q_{\backslash x_i}}\left[\ln L(\mathcal{D}, \{x_i\}_i)\right] + \text{const} \tag{129}$$

Eq. (129) will be used in deriving the optimal variational distribution for all the random variables.

**Optimal Polya-Gamma density** $q(\omega_{mm',k}^{(g)})$ We get

$$q(\omega_{mm',k}^{(g)}) = P_{\text{PG}}\left(\omega_{mm',k}^{(g)}|1, \sqrt{\phi_{m',k}\psi_{m,m'}\mathbb{E}_q(g_k^2(t_m - t_{m'}))}\right) \tag{130}$$

We can let $c_{mm',k}^{(g)} = \sqrt{\phi_{m',k}\psi_{m,m'}\mathbb{E}_q(g_k^2(t_m - t_{m'}))}$.

**Optimal Poisson process** $q(\Pi_{m,k}^{(g)})$ Using the mean-field update equation, we get the rate functions for the latent marked Poisson processes as:

$$\Lambda_{m,k}^{(g)}(t,\omega) = \frac{\exp(\mathbb{E}_q\left[\log v_k\right] - \frac{\phi_{m,k}\mathbb{E}_q(g_k(t))}{2})}{2^{\phi_{m,k}}\cosh(\frac{c_{k,m}^{(g)}(t)}{2})}P_{\text{PG}}(\omega|1, c_{k,m}^{(g)}(t)) \tag{131}$$

where $c_{k,m}^{(g)}(t) = \sqrt{\phi_{m,k}\mathbb{E}_q[g_k^2(t)]}$. Again, we emphasize that the support of $\Lambda_{m,k}^{(g)}(t,\omega)$ is $(0, T - t_m] \times \mathbb{R}^+$.

**Optimal branching variable** $q(b_m)$

$$log\psi_{m,0} = \sum_k \phi_{m,k}\mathbb{E}_q\left[\log(\pi_{i_mk}\pi_{j_mk})\right] = \sum_k \phi_{m,k}\left\{\mathbb{E}_q[\log\pi_{ik}] + \mathbb{E}_q[\log\pi_{jk}]\right\} \tag{132}$$

$$\forall m' : m' \in F_m,$$
$$\log\psi_{m,m'} = \sum_k \phi_{m',k}\mathbb{E}_q\left[\log(v_k) + h(\omega_{mm',k}^{(g)}, g_k(t_m - t_{m'}))\right]$$
$$= \sum_k \phi_{m',k}\left[\mathbb{E}_q[\log v_k] + \frac{\mathbb{E}_q[g_k(t_m - t_{m'})]}{2} - \frac{\mathbb{E}_q[g_k(t_m - t_{m'})^2]}{2}\mathbb{E}_q[\omega_{mm',k}^{(g)}] - \log 2\right] \tag{133}$$

**Optimal latent label** $q(z_m) = \phi_{m,k}$

$$\log\phi_{m,k}$$
$$= \mathbb{E}_q\left[\mathbf{1}_{b=0}(b_m)\log(\pi_{i_mk}\pi_{j_mk})\right] + \sum_{m'':m''\in F'_m}\mathbb{E}_q\left\{\mathbf{1}_{b=m}(b_{m''})\left[\log(v_k) + h(\omega_{m''m,k}^{(g)}, g_k(t_{m''} - t_m))\right]\right\}$$
$$+ \mathbb{E}_q\left[\sum_{(x,\omega)_n\in\Pi_{m,k}^{(g)}} h(\omega_n, -g_k(x_n))\right]$$
$$= \psi_{m,0}\left[\mathbb{E}_q[\log\pi_{i_mk}] + \mathbb{E}_q[\log\pi_{j_mk}]\right]$$
$$+ \sum_{m'':m''\in F'_m}\psi_{m'',m}\left(\mathbb{E}_q[\log v_k] + \frac{\mathbb{E}_q[g_k(t_{m''} - t_m)]}{2} - \frac{\mathbb{E}_q[g_k(t_{m''} - t_m)^2]}{2}\mathbb{E}_q[\omega_{m''m,k}^{(g)}] - \log 2\right)$$
$$- \int_0^{T-t_m}\left[\frac{\mathbb{E}_q[g_k(t)^2]}{2}\mathbb{E}[\omega_{k,m}^{(g)}(t)]\cdot\Lambda_{m,k}^{(g)}(t) + \left(\frac{\mathbb{E}_q[g_k(t)]}{2} + \log(2)\right)\cdot\Lambda_{m,k}^{(g)}(t)\right]dt \tag{134}$$

where $F'_m = \{m'' : (i_{m''}, j_{m''}) = (j, i) \cap t_{m''} > t_m\}$

**Optimal bound variable** $v_k$ The optimal density of $v_k$ is a Gamma distribution as:

$$q(v_k) \propto [v_k]^{a_m^{(k)}-1} e^{-b_m^{(k)} v_k} \tag{135}$$

where $a_m^{(k)} = a_m + \sum_m \sum_{m':m' \in F_m} \phi_{m',k} \psi_{m,m'} + \sum_m \int_0^{T-t_m} \Lambda_{m,k}^{(g)}(t)dt$, $b_m^{(k)} = b_m + \sum_m (T - t_m)$.

**Optimal feature vector** $\pi_i$ The optimal density of $\pi_{ik}$ is a Gamma distribution as:

$$q(\pi_{ik}) \propto [\pi_{ik}]^{a_{\pi,ik}^{(q)}-1} e^{-b_{\pi,ik}^{(q)} \pi_{ik}} \tag{136}$$

where $a_{\pi,ik}^{(q)} = \sum_{m':i_{m'}=i \cup j_{m'}=i} \Phi_{m',k} \psi_{m',0} + a_k$, $b_{\pi,ik}^{(q)} = b_k + 2(\sum_{j:j\neq i} \pi_{jk})T$.

**Optimal Gaussian processes** $g_k(t)$ For the triggering Gaussian processs $g_k(t)$, we can re-formulate the likelihood related to $g_k(t)$ as:

$$P(\mathcal{D}|g, v_k, z_m, b_m)$$
$$= \prod_m \prod_{(x,\omega)_m \in \Pi_{m,k}^{(g)}} e^{h(\omega_m, -g_k(x_m))\mathbf{1}_{z=k}(z_m)} \cdot \prod_m \prod_{m':m' \in F_m} \left[ v_k e^{h(\omega_{mm',k}^{(g)}, g_k(t_m - t_{m'}))} \right]^{\mathbf{1}_{b=m'}(b_m)\mathbf{1}_{z=k}(z_{m'})}$$
$$\tag{137}$$

where $\Pi_{m,k}^{(g)}$ denotes a marked points process defined in $(0, T - t_m] \times \mathbb{R}^+$, with intensity function $v_k P_{\text{PG}}(\omega|1, 0)dtd\omega$.

$$\log q(g_k) \propto \exp(U(g_k)) \tag{138}$$

$U(g_k)$ is defined as:

$$U(g_k)$$
$$= \sum_m \left\{ \phi_{m,k} \mathbb{E}_q \left[ \sum_{(x,\omega)_n \in \Pi_{m,k}^{(g)}} h(\omega_{k,m,n}^{(g)}, -g_k(x_{k,m,n}^{(g)})) \right] + \psi_{m,m'} \phi_{m',k} \mathbb{E}_q \left[ \sum_{m':m' \in F_m} h(\omega_{mm',k}, g_k(t_m - t_{m'})) \right] \right\}$$
$$= \left\{ -\frac{1}{2} \int_{\mathcal{T}} A(t) g_k^2(t)dt + \int_{\mathcal{T}} B(t) g_k(t)dt \right\}$$
$$\tag{139}$$

where

$$A(t)$$
$$= \sum_m \phi_{m,k} \cdot \int_0^\infty \omega \Lambda_{m,k}^{(g)}(t,\omega)d\omega + \sum_m \sum_{m':m' \in F_m} \left[ \psi_{m,m'} \phi_{m',k} \mathbb{E}_q \left[ \omega_{mm',k}^{(g)} \right] \delta(t - (t_m - t_{m'})) \right]$$
$$\tag{140}$$

$$B(t)$$
$$= -\frac{1}{2} \sum_m \phi_{m,k} \cdot \int_0^\infty \Lambda_{m,k}^{(g)}(t,\omega)d\omega + \sum_m \sum_{m':m' \in F_m} \left[ \frac{1}{2} \cdot \psi_{m,m'} \phi_{m',k} \delta(t - (t_m - t_{m'})) \right] \tag{141}$$

in which we have the following equations for the above integration:

$$\mathbb{E}_q[\omega_{mm',k}^{(g)}] = \frac{1}{2c_{mm',k}^{(g)}} \tanh \left( \frac{c_{mm',k}^{(g)}}{2} \right) \tag{142}$$

$$\int_{\mathcal{T}} \Lambda_{m,k}^{(g)}(t,\omega)d\omega = \Lambda_{m,k}^{(g)}(t) \tag{143}$$

$$\int_{\mathcal{T}} \omega \Lambda_{m,k}^{(g)}(t,\omega) d\omega = \frac{1}{2c_{k,m}^{(g)}(t)} \tanh\left(\frac{c_{k,m}^{(g)}(t)}{2}\right) \Lambda_{m,k}^{(g)}(t) \tag{144}$$

By using sparse Gaussian process methods, we get the sparse posterior distribution for the function values at the inducing points as:

$$q^{(s)} = \mathcal{N}(\boldsymbol{\mu}^{(s)}, \Sigma^{(s)}) \tag{145}$$

where the covariance matrix and mean is expressed as:

$$\Sigma^{(s)} = \left[ K_s^{-1} \int_{\mathcal{T}} A(t) k_s(t) k_s^\top dt K_s^{-1} + K_s^{-1} \right]^{-1} \tag{146}$$

$$\boldsymbol{\mu}^{(s)} = \Sigma^{(s)} \left( K_s^{-1} \int_{\mathcal{T}} B(t) k_s(t) dt \right) \tag{147}$$

Given the sparse solution, we can obtain the point estimation as

$$\boldsymbol{\mu}(t) = k_s^\top(t) K_s^{-1} \boldsymbol{\mu}^{(s)} \tag{148}$$

$$(s(t))^2 = k(t,t) - k_s^\top(t) K_s^{-1} (\boldsymbol{I} - \sum K_s^{-1}) k_s(t) \tag{149}$$

**Update the hyper-parameters of the Kernel functions in the Gaussian process** Given $\Lambda_0(x,\omega) = \lambda P_{PG}(\omega|1,0)dxd\omega$, the lower bound for the original paper is:

$$
\begin{aligned}
\mathcal{L}(\boldsymbol{q}) = &- \sum_{k,i\neq j} [\mathbb{E}_q[\pi_{i_m k}\pi_{j_m k}] \cdot T] + \sum_m \left[ \sum_k \phi_{m,k}\psi_{m,0}\mathbb{E}_q[\log \pi_{i_m k}\pi_{j_m k}] \right] \\
&+ \sum_{m,k} \left[ \int_0^\infty \int_0^{T-t_m} \left( \phi_{m,k}\mathbb{E}_q[h(\omega, -g_k(x))] + \mathbb{E}_q[\log v_k] - \mathbb{E}_q[\log \Lambda_{m,k}^{(g)}(x)] - \mathbb{E}_q \log \cosh(\frac{c_{k,m}^{(g)}(x)}{2}) \right. \right. \\
&\left. + \mathbb{E}_q[\frac{c_{k,m}^{(g)}(x)^2\omega}{2}] + 1 \right) \Lambda_{m,k}^{(g)}(x,\omega))dxd\omega - \mathbb{E}_q[v_k]\cdot(T-t_m) \Big] \\
&+ \sum_{m,k} \sum_{m':m'\in F_m} \Big[ \phi_{m',k}\psi_{m,m'} \left( \mathbb{E}_q[\log v_k] + \mathbb{E}_q[h(\omega_{mm',k}^{(g)}, g_k(t_m - t_{m'}))] \right) - \mathbb{E}_q \log\cosh(\frac{c_{mm',k}^{(g)}}{2}) \\
&+ \mathbb{E}_q[\frac{(c_{mm',k}^{(g)})^2\omega_{mm',k}^{(g)}}{2}] \Big] \Big\} - \sum_{m,k} \phi_{m,k}\log\phi_{m,k} - \sum_m \sum_{m':m'\in F_m} \psi_{m,m'}\log\psi_{m,m'} \\
&+ \sum_k \left\{ -\frac{1}{2}\mathrm{Tr}\left( [K_k^{(g,s)}]^{-1}(\Sigma_k^{(g,s)} + \boldsymbol{\mu}_k^{(g,s)}[\boldsymbol{\mu}_k^{(g,s)}]^\top) \right) - \frac{1}{2}\log\det K_k^{(g,s)} + \frac{1}{2}\log\det\Sigma_k^{(g,s)} \right\} + \frac{K}{2}N_s \\
&+ \sum_{i,k} \left\{ a^{(\pi)}\log b^{(\pi)} - \log\Gamma(a^{(\pi)}) + (a^{(\pi)} - 1)\mathbb{E}_q[\log\pi_{ik}] - b^{(\pi)}\mathbb{E}_q[\pi_{ik}] \right. \\
&+ a_{ik}^{(\pi,2)} - \log b_{ik}^{(\pi,2)} + \log\Gamma(a_{ik}^{(\pi,2)}) + (1 - a_{ik}^{(\pi,2)})F(a_{ik}^{(\pi,2)}) \Big\} \\
&+ \sum_k \left\{ a^{(m)}\log b^{(m)} - \log\Gamma(a^{(m)}) + (a^{(m)} - 1)\mathbb{E}_q[\log v_k] - b^{(m)}\mathbb{E}_q[m_k] \right. \\
&+ a_k^{(m,2)} - \log b_k^{(m,2)} + \log\Gamma(a_k^{(m,2)}) + (1 - a_k^{(m,2)})F(a_k^{(m,2)}) \Big\}
\end{aligned} \tag{150}
$$

$$
\begin{aligned}
\frac{\partial\mathcal{L}(\boldsymbol{q})}{\partial\Phi_k^{(g)}} = &\sum_m \int_0^\infty \int_0^{T-t_m} \phi_{m,k} \frac{\partial\mathbb{E}_q[h(\omega, -g_k(x))]}{\partial\Phi_k^{(g)}} \Lambda_{m,k}^{(g)}(x,\omega))dxd\omega \\
&+ \sum_m \sum_{m':m'\in F_m} \phi_{m',k}\psi_{m,m'} \frac{\partial\mathbb{E}_q[h(\omega_{mm',k}^{(g)}, g_k(t_m - t_{m'}))]}{\partial\Phi_k^{(g)}} \\
&+ \frac{1}{2}\mathrm{Tr}\left( [K_k^{(g,s)}]^{-1}\frac{\partial K_k^{(g,s)}}{\partial\Phi_k^{(g)}}[K_k^{(g,s)}]^{-1}(\Sigma_k^{(g,s)} + \boldsymbol{\mu}_k^{(g,s)}[\boldsymbol{\mu}_k^{(g,s)}]^\top) \right) - \frac{1}{2}\mathrm{Tr}\left( [K_k^{(g,s)}]^{-1}\frac{\partial K_k^{(g,s)}}{\partial\Phi_k^{(g)}} \right)
\end{aligned} \tag{151}
$$

The gradient of the Evidence lower bound to the hyper-parameters (we let $\Phi^{(g)} := (\theta^{(g)}, \delta^{(g)})$ in the kernel functions of the Gaussian process) is:

$$\frac{\partial \mathbb{E}_q[h(\omega, f(x))]}{\partial \Phi^{(g)}} = \frac{1}{2}\left(\frac{\partial \mathbb{E}_Q[f(x)]}{\partial \Phi^{(g)}} - \frac{\partial \mathbb{E}_Q\left[f(x)^2\right]}{\partial \Phi^{(g)}}\mathbb{E}_Q[\omega]\right) \tag{152}$$

$$\frac{\partial \mathbb{E}_q[f(x)]}{\partial \Phi^{(g)}} = \frac{\partial \boldsymbol{\kappa}(x)}{\partial \Phi^{(g)}}\boldsymbol{\mu}_2^{(g,s)} \tag{153}$$

$$\frac{\partial \mathbb{E}_q\left[f(x)^2\right]}{\partial \Phi^{(g)}} = \frac{\partial \tilde{k}(x,x)}{\partial \Phi^{(g)}} + \frac{\partial \boldsymbol{\kappa}(x)^\top}{\partial \Phi^{(g)}}\left(\Sigma_2^{(g,s)} + \boldsymbol{\mu}_2^{(g,s)}\left(\boldsymbol{\mu}_2^{(g,s)}\right)^\top\right)\boldsymbol{\kappa}(x)$$
$$+ \boldsymbol{\kappa}(x)^\top\left(\Sigma_2^{(g,s)} + \boldsymbol{\mu}_2^{(g,s)}\left(\boldsymbol{\mu}_2^{(g,s)}\right)^\top\right)\frac{\partial \boldsymbol{\kappa}(x)}{\partial \Phi^{(g)}} \tag{154}$$

where $\boldsymbol{\kappa}(x) = \boldsymbol{k}_s(x)^\top K_s^{-1}$ and $\tilde{k}(t,t) = k(x,x) - \boldsymbol{k}_s(x)K_s^{-1}\boldsymbol{k}_s(x)^\top$. The remaining two terms are:

$$\frac{\partial \tilde{k}(x,x)}{\partial \Phi^{(g)}} = \frac{\partial k(x,x)}{\partial \Phi^{(g)}} - \frac{\partial \boldsymbol{\kappa}(x)}{\partial \Phi^{(g)}}\boldsymbol{k}_s(x) - \boldsymbol{\kappa}(x)\frac{\partial \boldsymbol{k}_s(t)}{\partial \Phi^{(g)}} \tag{155}$$

$$\frac{\partial \boldsymbol{\kappa}(x)}{\partial \Phi^{(g)}} = \frac{\partial \boldsymbol{k}_s(x)^\top}{\partial \Phi^{(g)}}K_s^{-1} - \boldsymbol{k}_s(x)K_s^{-1}\frac{\partial K_s}{\partial \Phi^{(g)}}K_s^{-1} \tag{156}$$

After each variational step, the hyper-parameters are updated by

$$\boldsymbol{\Phi}^{(g)} = \boldsymbol{\Phi}^{(g)} + \varepsilon\frac{\partial \mathcal{L}(q)}{\partial \boldsymbol{\Phi}^{(g)}} \tag{157}$$

$$\boldsymbol{\Phi}_k^{(g)} = \boldsymbol{\Phi}_k^{(g)} + \varepsilon\frac{\partial \mathcal{L}(q)}{\partial \boldsymbol{\Phi}_k^{(g)}} \tag{158}$$

where $\epsilon$ is the stepsize.