# OpenReview forum: "Continuous-time edge modelling using non-parametric point processes"
_NeurIPS.cc/2021/Conference — NeurIPS 2021 Poster_

### Official Review · Reviewer_Kebi · 2021-07-05

**Rating:** 7
**Confidence:** 4

**Summary:**

The paper proposes three  non-parametric point process variants for continuous-time edge modelling (CTEM) and the performance of the proposed framework evaluated on four real world dynamic networks. The three cases are
1) a Poisson process in which K time series are generated according to a (sigmoidal) GP and weighted by node specific feature weights utilizing these temporal evolutions.
2) An extension to the Hawkes process such that reciprocity is invoked by an additional sigmoidal GP weighted by other node specific feature weights defined through time to past reciprocative interactive events.
3) A further extension incorporating edge clustering rather than node specific weights for the reciprocating effects (i.e., endogeneous functions) reducing the parameter space and enabling specific endogenous effects according to the cluster used for the specific reciprocative event of each dyad.

VB is used for large scale inference and inducing points invoked to reduce complexity of the GP. The model is contrasted the Poisson and Hawkes IRM as well as Hawkes-DLS and HawkesNetOC and found on the four considered datasets to have superior performance as quantified using time-binned AUC but inferior results on two datasets for the (test I assume) log-likelihood. In particular, the edge clustering approach 3) forming the NP-HP-C is found to be the superior of the proposed approaches.


**Ethical Concerns:**

The analysis is based on open and publicly available datasets.

**Limitations And Societal Impact:**

There are no broader impact discussed of the paper nor potential negative societal impact. Enhanced abilities modeling dynamic networks and understanding their structure inevitably can also be used for surveillance, the enhancement of filter bubbles and be used to negatively influence people based on the improved understanding of their behaviors. Some of these potential negative aspects by the enhanced ability to predict behavior in social dynamic networks could be discussed.

**Main Review:**

This is a well written and solid contribution in which multiple novel dynamic network models for directed dynamic networks are proposed.

$\textbf{Originality}$

The paper is based on a series of well-established methodologies, i.e. Hawkes process for dynamic network modeling , sparse GP for modeling smooth temporal trajectories efficiently, VB for efficient scalable inference. However, their combination is novel and the proposed framework both sound and efficient enabling accurate dynamic network analysis accounting for reciprocative relationships. The edge clustering idea is neat and the use of dynamic feature representations that are weighted in terms of node specific weights sound and useful. The inference is non-trivial and appears well implemented.


$\textbf{Quality}$

The approach developed is technically sound and well decribed. The experimentation is solid with reasonable comparisons included to existing methods considering a sufficient  number of (four) datasets. The results are convincing, the approach well motivated and with potential wide applications within dynamic network analysis.

It would be good to position the paper also in relation to work using GPs for dynamic network analysis, see also:

Durante, Daniele, and David Dunson. "Bayesian logistic gaussian process models for dynamic networks." Artificial Intelligence and Statistics. PMLR, 2014.

and nested GPs

Durante, Daniele, and David B. Dunson. "Locally adaptive dynamic networks." The Annals of Applied Statistics 10.4 (2016): 2203-2232.

There are further more advanced Poisson Process stochastic block modeling approaches than the PP IRM worth discussing. This includes the work of:
Matias, C., Rebafka, T., & Villers, F. (2018). A semiparametric extension of the stochastic block model for longitudinal networks. Biometrika, 105(3), 665-680.

There are no error bars reported in the results. It would strengthen the paper to have multiple restarts and provide the robustness of the results in terms of initialization of the inference procedure. In particular Table 2 does not provide error bars. The authors state in checlist point 3.c "Due to the space limit, we put the average results in the main content and complete results with errors are provided in the Supplementary Material." However, I was unable to find such error bars in the supplementary for the most important Table 2 results.

$\textbf{Clarity}$

The paper is well written and in general easy to follow. However, method 3) is quite convoluted in its description and although the authors provide a reasonable attempt in describing the steps and intuition behind the generative model this section could be improved in clarity and the reader guided more carefully through the steps.

The authors set the number of features to K=5 but does not discuss this choice further nor its impact on the results. It would be good to elaborate on this.

Furthermore, there are very minor comments to be considered:

these models oo -> these models to

In table 2 it is a bit confusing to call it training log-likelihood when it is in fact the test log likelihood I believe on last 30% of the available time segment.v Please clarify what the results refer to as it should be test-log likelihood.

$\textbf{Significance}$

Although the use of Hawkes processes is well established the proposed approach is very useful in having time-evolving rates that are flexibly and well implemented through feature representations and clustering. The results are convincing and the proposed approach can have a solid impact within dynamic network analyses in general.

--------------------Response to the authors rebuttal-------------------------------------

I thank the authors for their responses to my concerns. It is also good that error bars are now provided and can be assessed.
In regards to 1) I agree that the work of the authors differ. However, the point was to position the manuscript in context of existing work utilizing GPs for dynamic network analysis. I agree that the work I referred to using GPs differ considering time-discretized network and a Bernoulli likelihood. However, the approach also utilizes GPs defined in continuous time.

The authors could have elaborated in their rebuttal on the work of:
Matias, C., Rebafka, T., & Villers, F. (2018). A semiparametric extension of the stochastic block model for longitudinal networks. Biometrika, 105(3), 665-680.
Which also considers continuous time modeling using a stochastic block model including kernel and histogram approximations of the rate function.



**Time Spent Reviewing:**

2.5

---

> ### Author Response · Authors · 2021-08-10
> **Response to Reviewer Kebi**
>
> 1. ''**It would be good to position the paper also in relation to work using GPs for dynamic network analysis,**''
>
> **Answer** Thank you for the references. We note that our approach here has a different problem setting from the referenced work. Our approach models the continuous-time observed edges, whereas the referenced work studies snapshots of networks. Both the observed and unobserved edges are used to infer the variables/parameters of the model.
>
> 2. ''**There are no error bars reported in the results**''
>
> **Answer** Thank you! Despite stating otherwise in the manuscript, we mistakenly (and embarrassingly) forgot to include the standard errors in the Supplementary Material. We will include these in the revised version.
>
> 3. ''**The authors set the number of features to K=5 but does not discuss this choice further nor its impact on the results. It would be good to elaborate on this.**''
>
> **Answer** Thank-you. Similarly to our response to the other Reviewers, we are not implying that $K$ should be fixed at the same value for all data sets. We fix $K=5$ (unless otherwise varied) simply as an expedient value we think is large enough to capture all of the variability in each of the datasets, to permit us to understand model performance. Where we are interested in more parsimonious modelling, we can explore smaller (or larger) values of $K$ for a given data set.
>
> More generally, our understanding is that the number of clusters scales sub-linearly with the number of edges. For instance, the number of clusters in the Chinese Restaurant Process scales logarithmically with the number of data points. According to Figures 4-6 of the Supplementary Material, around 2-3 latent Sigmoid Gaussian processes are enough to model the continuous-time edge data. That is, around $2$-$3$ out of the $K=5$ latent labels, exogenous rate functions and endogenous functions dominate the others.
>
> We will improve the discussion on this point in the revision.
>
> 4. ''**In table 2 it is a bit confusing to call it training log-likelihood when it is in fact the test log likelihood I believe on last 30\% of the available time segment.v Please clarify what the results refer to as it should be test-log likelihood.**''
>
> **Answer** Thank you for pointing out this typo. These results refer to the test-log likelihood and we will correct it in the paper.

---

> > ### Comment · Reviewer_Kebi · 2021-08-27
> > **Error bars could be included in response**
> >
> > I have read the authors rebuttal.
> > In regards to 2) I am somewhat surprised the authors in their rebuttal do not provide the error bars that are missing but promise to include them in a final version of the manuscript. What is preventing the authors from including the error bars in the rebuttal?

---

> > > ### Author Response · Authors · 2021-08-27
> > > **Thank you for the follow up and error bars are provided here**
> > >
> > > Thank you for the follow up. In our previous response, as we were intending to put the missing error bars in a Figure, we did not think that we were able to include this as part of our answer. However, we are able to do this in the form of a table, which we include below as follows:
> > >
> > > **Test-log likelihood** (mean$\pm$ standard error)
> > >
> > > |    | Email | College | Overflow | Ubuntu |
> > > | ------- | ------- | ------- | ------- | ------- |
> > >   Poisson-IRM | $-62\,320\pm769$  | $-58\,552\pm575$  | $-90\,786\pm797$  | $-96\,630\pm1\,107$ |
> > >   Hawkes-IRM | $-58\,341\pm334$  | $- 59\,461\pm1\,448$  | $-88\,746\pm1043$  | $-90\,435\pm376$  |
> > >   Hawkes-DLS | $-59\,810\pm859$  | $-56\,237\pm1\,207$  | $-97\,505\pm332$  | $-93\,485\pm299$  |
> > >   HawkesNetOC | $-57\,483\pm550$  | $-57\,706\pm424$  | $\pmb{-88\,193}\pm582$  | $\pmb{-89\,850}\pm1\,022$  |
> > >   NP-PP | $-68\,475\pm1\,274$  | $-71\,154\pm549$  | $-133\,839\pm829$  | $-136\,830\pm810$  |
> > >   NP-HP | $-53\,121\pm964$  | $-55\,729\pm813$  | $-114\,277\pm985$  | $-120\,419\pm796$  |
> > >   NP-HP-C | $\pmb{-50\,126}\pm568$  | $\pmb{-53\,958}\pm475$  | ${-125\,387}\pm376$  | ${-134\,027}\pm394$  |
> > >
> > > **AUC** (mean$\pm$ standard error)
> > >
> > > |    | Email | College | Overflow | Ubuntu |
> > > | ------- |  ------- | ------- | ------- | ------- |
> > >   Poisson-IRM   | $77.46\pm0.87$  | $79.18\pm1.07$  | $83.14\pm0.93$  | $83.20\pm0.74$ |
> > >   Hawkes-IRM   | $79.42\pm0.41$  | $80.14\pm1.28$  | $85.16\pm0.85$  | $84.14\pm0.97$ |
> > >   Hawkes-DLS   | $80.15\pm1.02$  | $80.16\pm1.07$  | $81.63\pm0.46$  | $80.19\pm1.03$   |
> > >   HawkesNetOC | $83.17\pm1.30$  | $81.38\pm0.89$  | $83.48\pm0.79$  | $82.24 \pm0.94$   |
> > >   NP-PP   | $78.16\pm0.94$  | $79.71\pm0.92$  | $89.15\pm0.74$  | $86.77\pm0.90$   |
> > >   NP-HP   | $81.78\pm0.97$  | $81.57\pm0.64$  | $89.58\pm9.81$  | $86.78\pm0.85$   |
> > >   NP-HP-C   | $\pmb{85.23}\pm0.88$  | $\pmb{82.17}\pm0.65$  | $\pmb{89.91}\pm0.83$  | $\pmb{88.02}\pm0.38$   |

---

> ### Author Response · Authors · 2021-09-02
> **Discussions on the suggested related works will be included in the revised version**
>
> **Answer:** Thank you so much for suggesting these related works. We will of course cite these papers and include the following discussions in the revised version.
>
> Instead of using point processes to model the continuous-time edges, the approaches of [1,2] studied time-discretised networks and used Bernoulli emission distributions to model all the binary-valued edges at each observed timestamp. They adopted Gaussian processes/Nested Gaussian processes to generate continuous-time features for each node at the observed timestamps. The approach of [3] also considered continuous-time edges. It extended the Poisson IRM by using histogram or kernel approximations to model the time-varying exogenous rate function, whereas our NP-PP uses Sigmoidal Gaussian Processes approximation.
>
> [1] Durante, Daniele, and David Dunson. "Bayesian logistic gaussian process models for dynamic networks." Artificial Intelligence and Statistics. PMLR, 2014.
>
> [2] Durante, Daniele, and David B. Dunson. "Locally adaptive dynamic networks." The Annals of Applied Statistics 10.4 (2016): 2203-2232.
>
> [3] Matias, C., Rebafka, T., & Villers, F. (2018). A semiparametric extension of the stochastic block model for longitudinal networks. Biometrika, 105(3), 665-680.

---

### Official Review · Reviewer_sJ8A · 2021-07-11

**Rating:** 6
**Confidence:** 4

**Summary:**

-This work develops a family of novel non-parametric point processes for continuous-time edge data. In details, the model captures the periodicity exhibited in its base rate using K sigmoidal Gaussian processes(SGPs) \sigmoid(f_k(t)).
-For each pair of two nodes, the nodal pair specific coefficients \p_{ik}, \pi_{jk} is used to rescale the pair-wise base intensity of (i,j) and (j,i).
-To  capture the reciprocity, the model employs a second set of functions \sigmoid(g_k(t)) drawn from SGPs, which has more flexibility in capturing exciting behaviors, compared to the exponential decaying kernel used in Hawkes processes.
- A mean-field variational scheme is derived to perform model inference.
-The model is demonstrated on four real data to illustrate its estimated periodically varying base intensity and the endogenous functions. They also show the superior performance of the model in terms of training log-likelihood and link prediction.



**Ethics Review Area:**

["I don’t know"]

**Limitations And Societal Impact:**

The authors come up with some interesting phenomenons, which cannot be fully captured using Hawkes process with constant base rate and parametric kernels. In addition, I think the proposed symmetric intensity function may also some limitations. For instance, an employee may immediately reply to an email from her/his team leader, while the leader may not reply to the employees quickly. Hence, I am wondering how to model asymmetric nodal pair-specific intensity by extending the framework.


**Main Review:**

-Originality
The idea of modelling time-varying base rate, and flexible endogenous functions is interesting and novel, to me.
Nonetheless, both the sigmoidal Gaussian processes and the main techniques for the proposed variational inference scheme have been exploited well before.

-Quality
The novel point processes models and the derived variational algorithm are sound.

-Clarity
The notations are not clear to me, e.g., why you have nodal pair specific coefficients \rho_{ik}\rho_{jk} in Eq.(4,5) but you change to v in Eq.(6)?
For experimental setup, it is not clear how to choose the number of features K, and why K is fixed for all data sets. I would consider the features may represent the latent group or clusters in the Email, Overflow and College data. Thus, some explainations are missing for the setting of K.
It is not clear why to chose the first 70% of edges as training data while chose remaining 30% as test data. As you are motivated, the exogenous intensity of events may have some periodical pattern. Thus, it is not clear how to judge the remaining 30% of edges still have the same periodicity, has entered a novel period that are not observed before.

-Significance
To me, the novel point processes may have some impacts for a small group of audience who are interested in modeling excitatory behaviors of continuous-time dyadic data.


**Time Spent Reviewing:**

3 hours

---

> ### Author Response · Authors · 2021-08-10
> **Response to Reviewer sJ8A**
>
>
> 1. ''**why you have nodal pair specific coefficients $\rho_{ik}\rho_{jk}$ in Eq.(4,5) but you change to v in Eq.(6)?**''
>
> **Answer** Thank you. These two different notations represent different quantities.  $\rho_{ik}$ represents node $i$'s $k$-th latent endogenous feature value, whereas $v_k$ denotes the scaling value for the $k$-th endogenous function. We will aim to improve the clarity on this point in the revision.
>
> 2. ''**some explanations are missing for the setting of K'' "Why is K fixed for all data sets?**"
>
> **Answer** Thank you. Similarly to our response to Reviewer 2, question 3, our understanding is that the number of clusters scales sub-linearly with the number of edges. For instance, the number of clusters in the Chinese Restaurant Process scales logarithmically with the number of data points. According to Figures 4-6 of the Supplementary Material, around 2-3 latent Sigmoid Gaussian processes are enough to model the continuous-time edge data. That is, around $2$-$3$ out of the $K=5$ latent labels, exogenous rate functions and endogenous functions dominate the others. We will improve the discussion on this point in the revision.
>
> We are not implying that $K$ should be fixed at the same value for all data sets. We fix $K=5$ (unless otherwise varied) simply as an expedient value we think is large enough to capture all of the variability in each of the datasets, to permit us to understand model performance. Where we are interested in more parsimonious modelling, we can explore smaller (or larger) values of $K$ for a given data set.
>
> 3. ''**it is not clear how to judge the remaining 30\% of edges still have the same periodicity, has entered a novel period that are not observed before.**''
>
> **Answer** Thank you. This is an important point, relevant for all model testing scenarios. We have implicitly assumed that the edges in the testing period have the same periodicity as those in the training period. If this was not thought to hold, we would need to vary our training and test datasets. We will improve our discussion on this point in the revision.
>
> 4. ''**the proposed symmetric intensity function may also some limitations. For instance, an employee may immediately reply to an email from her/his team leader, while the leader may not reply to the employees quickly. Hence, I am wondering how to model asymmetric nodal pair-specific intensity by extending the framework.**''
>
> **Answer** Thank you. The proposed symmetric intensity is indeed a simplified setting, with the benefit that the computational complexity scales with $\mathcal{O}(K)$. It would be straightforward to implement the asymmetric case, by constructing $\mu_{ij}(t):=\sum_{k_1k_2}\pi_{ik_1}\pi_{jk_2}$ $\sigma(f_{k_1k_2}(t))$.
>
> However, in this case the computational complexity of the asymmetric case scales with $\mathcal{O}(K^2)$. In this presented example, the different response time of team leaders and employees can be modelled through different types of endogenous function within the NP-HP-C model. We will discuss this point in the revision.

---

### Official Review · Reviewer_2ejX · 2021-07-21

**Rating:** 6
**Confidence:** 3

**Summary:**

The authors propose a family of models based on mutually-exciting Hawkes process that aim at modeling continuous time edges. The framework is based on a variational inference procedure for a fast and efficient computation. The models have been quantitatively and qualitatively evaluated with respect to baselines, showing that, in general, they outperform the baselines in terms of accuracy.

**Limitations And Societal Impact:**

I think limitations of the models have not been fully discussed. In particular, based on the complexity of the models it does not seem that they are able to scale to large set of edges, in particular when the number of clusters is high. This would be a relevant limitation as I argue that, when increasing the number of edges, an higher number of clusters would be needed to better describe all of the network.

**Main Review:**

The paper include many details (in particular in the supplementary material) that help comprehension. However, the math section of the main paper is not always clear. In particular, it took me a while to check what some notation meant (such as M_ij), before realising it was explained later on in the section. I would suggest to explain the notation before or at the point when it is first introduced to simplify reading.

I like the experiments section. The results are clear and well explained. They also aligned with the properties of the models presented in the paper. However, the choice of K=1,3,5 seem a bit restrictive, in particular with the order of thousands of edges. I understand this is because the model scales linearly with the number of K as well as the number of edges. How much would be the computational limit of the model in terms of the edges and clusters?

In this terms I'm not fully convinced the model can really scale to large datasets, as I believe a linear dependency from the number of edges and the number of clusters would not let the model be applicable to (say) social networks.
I think this is important as, when increasing the number of edges in the network, an higher number of clusters would in turn be required to better model the data at hand.


Minor:
line 33: "oo" instead of "to"
line 146 it's

**Time Spent Reviewing:**

2.5

---

> ### Author Response · Authors · 2021-08-10
> **Response to Reviewer 2ejX**
>
> 1. ''**I would suggest to explain the notation before or at the point when it is first introduced to simplify reading.**''
>
> **Answer** Thank you -- we agree that the presentation could be improved in this way. We will ensure that all notation is explained when it is first introduced, in the revision.
>
> 2. ''**How much would be the computational limit of the model in terms of the edges and clusters?**''
>
> **Answer** Thank you. The right most panel of Figure 3 illustrates the computational cost of the NP-HP-C, by demonstrating that around $6$ hours are needed for posterior inference in the case of $80\,000$ edges and $K=5$. Extrapolation of these results will provide an idea of restrictions on edges and clusters for given computational limits. For example, if $24$ hours is the computational limit, we can perform posterior inference for approximately $300\,000$ edges with $K=5$\, $500\,000$ edges with $K=3$ and $1\,500\,000$ edges for $K=1$.
>
> 3. ''**In this terms I’m not fully convinced the model can really scale to large datasets, as I believe a linear dependency from the number of edges and the number of clusters would not let the model be applicable to (say) social networks...This would be a relevant limitation as I argue that, when increasing the number of edges, an higher number of clusters would be needed to better describe all of the network.**''
>
> **Answer** Thank you for this observation. Our understanding is that the number of clusters scales sub-linearly with the number of edges. For instance, the number of clusters in the Chinese Restaurant Process scales logarithmically with the number of data points. According to Figures 4-6 of the Supplementary Material, around 2-3 latent Sigmoid Gaussian processes are enough to model the continuous-time edge data. That is, around $2$-$3$ out of the $K=5$ latent labels, exogenous rate functions and endogenous functions dominate the others. We will improve the discussion on this point in the revision.

---

### Official Review · Reviewer_Cj5q · 2021-07-25

**Rating:** 6
**Confidence:** 4

**Summary:**

The paper presents a Gaussian process based nonparametric method that enables flexible modeling of temporal interacting events. With a fast variational inference method, the proposed approach shows improved empirical predictive performance on four real datasets compared to several existing work.

**Limitations And Societal Impact:**

No potential negative societal impacts are involved in this paper.

**Main Review:**

Strength:

1. A Gaussian process based nonparametric method for modeling the conditional intensity function to capture temporal events between pairs of interacting nodes (NP-HP).
2. An extension of NP-HP to capture the clustering structures among interacting nodes, referred to as NP-HP-C by introducing latent group label similar to the stochastic blocking model
3. Strong empirical results show competitive modeling capabilities compared to other SOTA approaches.

Limits:

1. Do the conditional intensity functions in (4) and (5) need some minimum positive background intensity to make sure that they are not too small to be close to zero to have numerical instability issue as mentioned by the work [Estimation of space–time branching process models in seismology using an EM–type algorithm, 2008] and [Robust identification of controlled Hawkes processes, 2020] since the sigmoid function can decay fast to be near zero.
2. Although the experiments show competitive performance across four real datasets, it will be more convincing to use a small synthetic dataset showing that NP-HP-C is able to better capture the clustering structures compared to other stochastic blocking based baselines.
3. On overflow and ubuntu, it shows that the proposed method gives the worst results compared to other baselines. More elaborations are needed. In particular, it will be very helpful to include the MAE of the time prediction as the evaluation metric since this is quite a standard metric used in the temporal point process literature.


**Time Spent Reviewing:**

1.5-2 hours

---

> ### Author Response · Authors · 2021-08-11
> **Response to Reviewer Cj5q**
>
> 1. ''**Do the conditional intensity functions in (4) and (5) need some minimum positive background intensity to make sure that they are not too small to be close to zero to have numerical instability issue as mentioned by the work ... since the sigmoid function can decay fast to be near zero.**''
>
>
> **Answer** No – the inferential procedure for the proposed model is based on the existing works [5,28], which do not suffer the numerical instability issue discussed in the papers abovementioned by the reviewer. We did not encounter such numerical instability issues as well. A possible reason may be that the Sigmoid function decay very slowly as the input approaches to negative infinity.
>
> [5] Christian Donner and Manfred Opper. Efficient Bayesian inference of sigmoidal Gaussian Cox
> processes. The Journal of Machine Learning Research, 19(1):2710–2743, 2018.
>
> [28] Feng Zhou, Zhidong Li, Xuhui Fan, Yang Wang, Arcot Sowmya, and Fang Chen. Efficient inference for nonparametric Hawkes processes using auxiliary latent variables. Journal of Machine Learning Research, 21(241):1–31, 2020.
>
> 2. ''**it will be more convincing to use a small synthetic dataset showing that NP-HP-C is able to better capture the clustering structures ...**''
>
> **Answer** Thank you. We have already provided a visualisation of the clustering structure of the NP-HP-C for the four real-world datasets, which can be found in the bottom panel of Figure 4 in the Supplementary Material. However, we agree that exploring this for simulated data has merit, given the presence of a known truth. Accordingly we will include a visualisation for the NP-HP-C on synthetic data, in a similar way to the exogenous function fitting and latent exogenous feature prediction for synthetic data in Figures 1 \& 2 of the Supplementary Material.
>
> 3. ''**On overflow and ubuntu, it shows that the proposed method gives the worst results compared to other baselines. More elaborations are needed.**''
>
> **Answer** Thank you -- we agree that the manuscript discussed this issue too briefly, and we will improve the text around this point. For the Overflow and Ubuntu datasets, our methods perform the best in terms of AUC, but poorly in terms of testing log-likelihood. A possible reason is that these two datasets have relatively simple patterns (here low-response activities) which can be well captured by simple rate functions. For instance, the exogenous functions for these slow-response datasets did not show clear periodicities (see the right two panels in the top row of Figure 2). Simple rate functions like constant functions may well describe the exogenous effects. For the endogenous effects, they are dominated by one single endogenous function in both Overflow and Ubuntu (see the right two panels in the bottom row of Figure 2). One single exponential decay function may also capture these simple patterns.
>
> 4. ''**... it will be very helpful to include the MAE of the time prediction as the evaluation metric since this is quite a standard metric used in the temporal point process literature....**''
>
>
> **Answer** Thank you. We have implemented the MAE calculation of the time prediction for the NP-HP-C and Hawkes-IRM. The results are displayed below (and will be incorporated into the revision):
>
> |    | Email | College | Overflow | Ubuntu |
> | ------- | ------- | ------- | ------- | ------- |
> | Hawkes-IRM  | $0.0437$  | $0.0381$  | $0.0072$  | $0.0027$  |
> | NP-HP-C  |  $0.0279$  | $0.0244$  | $0.0057$  | $0.0023$  |
>
> Our NP-HP-C performs better than Hawkes-IRM in Mean Absolute Error (MAE), indicating that our NP-HP-C predicts the number of interactions in the future time interval better than Hawkes-IRM.

---

### Decision · Program_Chairs · 2021-09-27

**Decision:**

Accept (Poster)

**Comment:**

Overall, the reviewers were positive about your work, though there are a number of suggestions that you should make to improve the manuscript. I remain a little perplexed by the performance on Overflow and Ubuntu--isn't it odd that AUC is highest but log-likelihood is lowest? Could this be due to the use of the mean-field variational Bayes approximation which can give very wrong posteriors?